# The Poisson Midpoint Method for Langevin Dynamics: Provably Efficient Discretization for Diffusion Models

**Saravanan Kandasamy**[*]
Department of Computer Science
Cornell University
sk3277@cornell.edu

**Dheeraj Nagaraj**
Google DeepMind
dheerajnagaraj@google.com

## Abstract

Langevin Dynamics is a Stochastic Differential Equation (SDE) central to sampling and generative modeling and is implemented via time discretization. Langevin Monte Carlo (LMC), based on the Euler-Maruyama discretization, is the simplest and most studied algorithm. LMC can suffer from slow convergence - requiring a large number of steps of small step-size to obtain good quality samples. This becomes stark in the case of diffusion models where a large number of steps gives the best samples, but the quality degrades rapidly with smaller number of steps. Randomized Midpoint Method has been recently proposed as a better discretization of Langevin dynamics for sampling from strongly log-concave distributions. However, important applications such as diffusion models involve non-log concave densities and contain time varying drift. We propose its variant, the Poisson Midpoint Method, which approximates a small step-size LMC with large step-sizes. We prove that this can obtain a quadratic speed up of LMC under very weak assumptions. We apply our method to diffusion models for image generation and show that it maintains the quality of DDPM with 1000 neural network calls with just 50-80 neural network calls and outperforms ODE based methods with similar compute.

## 1 Introduction

The task of sampling from a target distribution is central to Bayesian inference, generative modeling, differential privacy and theoretical computer science [48, 20, 17, 25]. Sampling algorithms, based on the discretization of a stochastic differential equation (SDE) called the Langevin Dynamics, are widely used. The straightforward time discretization (i.e., Euler Maruyama discretization) of Langevin dynamics, called Langevin Monte Carlo (LMC), is popular due to its simplicity. The convergence properties of LMC have been studied extensively in the literature under various conditions on the target distribution [8, 12, 11, 45, 15, 33, 7, 1, 6, 9, 16, 30, 53, 5]. LMC can suffer from slow convergence to the target distribution, and often requires a large number of steps with a very fine time discretization (i.e., small step-size), making it prohibitively expensive.

The Poisson Midpoint Method introduced in this paper approximates multiple steps of small step-size Euler-Maruyama discretization with one step of larger step-size via stochastic approximation. In the case of LMC, we show that our method (called PLMC) converges to the target as fast as LMC with a much smaller step-size without any additional assumptions such as isoperimetry or strong log concavity (up to a small additional error term). This is a variant of the Randomized Midpoint Method (RLMC) studied in the literature [40, 18, 50] (see Section 1.1 for comparison).

---

[*]This work was done when SK was a student researcher at Google Research.

38th Conference on Neural Information Processing Systems (NeurIPS 2024).

Diffusion models are state-of-the-art in generating new samples of images and videos given samples [20, 43, 34]. These start with a Gaussian noise vector and evolve it through the time-reversal of the SDE called the Ornstein-Uhlenbeck process. The time reversed process can be written as an SDE (Langevin Dynamics with a time dependent drift) or as an ODE (see Section 2). The DDPM scheduler [20], which discretizes the SDE, obtains the best quality images with a small step-size and a large number of steps (usually 1000 steps). However, its quality degrades with larger step-sizes and a small number of steps (say 100 steps). Schedulers such as DDIM ([42]), DPM-Solver ([28, 29]), PNDM ([27]) which solve the ODE via numerical methods perform much better than DDPM with a small number of steps. However, it is noted that they do not match the performance of DDPM with 1000 steps over many datasets ([42, 28, 41]). Poisson Midpoint Method gives a scheduler for the time-reversed SDE which maintains the quality of DDPM with 1000 steps, with just 50-80 steps.

## 1.1 Prior Work

Euler Maruyama discretization of SDEs is known to be inefficient and many powerful numerical integration techniques have been studied extensively ( [23, 32, 3, 26]). However, higher order methods such as the Runge-Kutta method require the existence and boundedness of higher order derivatives of the drift. The Randomized Midpoint Method for LMC (RLMC) was introduced for strongly log-concave sampling [40] and was further explored in [50, 18]. It was shown that RLMC, under certain conditions, can sample with a larger step size for fewer steps compared to Euler Maruyama and yet obtain the same accuracy. RLMC is popular due to its simplicity, and ease of implementation and does not require higher order bounded derivatives of the drift function. However, the current theoretical results are restricted to the case of strongly log-concave sampling, whereas non-log-concave sampling is of immense practical interest.

## 1.2 Our Contributions

**(1)** We design the Poisson Midpoint Method which discretizes SDEs by approximating $K$-steps of Euler-Maruyama discretization with step-size $\frac{\alpha}{K}$ by just one step with step-size $\alpha$. We show a strong error bound between these two processes under general conditions in Theorem 1 (no assumption on mixing, smoothness etc). This is based on a Central Limit Theorem (CLT) based method in [10] to analyze stochastic approximations of LMC.

**(2)** We apply our method to LMC to obtain PLMC. We show that it achieves a speed-up in sampling for both Overdamped LMC (OLMC) and Underdamped LMC (ULMC) whenever LMC mixes, without additional assumptions such as isoperimetry or strong-log concavity.

**(3)** When the target obeys the Logarithmic Sobolev Inequalities (LSI), we show that PLMC achieves a quadratic speed up for both OLMC and ULMC. Prior works on midpoint methods [40, 18, 50] only considered strongly log-concave distributions. We also show an improvement in computational complexity for ULMC from $\frac{1}{\epsilon^{2/3}}$ to $\frac{1}{\sqrt{\epsilon}}$ to achieve $\epsilon$ error[2].

**(4)** Empirically, we show that our technique can match the quality of DDPM Scheduler with 1000 steps with fewer steps, achieving up to 4x gains in compute. Over multiple datasets, our method outperforms ODE based schedulers such as DPM-Solver and DDIM in terms of quality.

## 1.3 Notation:

$X_{0:T}$ denotes $(X_t)_{0 \le t \le T}$ and $X_{K(0:T)}$ denotes $(X_{tK})_{0 \le t \le T}$. $\mathbf{I}$ denotes identity matrix over $\mathbb{R}^{k \times k}$ whenever $k$ is clear from context. For any vector $\mathbf{x} \in \mathbb{R}^k$, $\|\mathbf{x}\|$ denotes its Euclidean norm. For any $a, b \in \mathbb{Z}$ and $a > b$, we take the $\sum_{t=a}^{b}$ to be 0, and the product $\prod_{t=a}^{b}$ to be 1. Underdamped Langevin Dynamics happens in $\mathbb{R}^{2d}$. Here, we take vectors named $X$ (along with subscripts and superscripts) as $X = [U \quad V]^{\mathsf{T}}$ where $U, V \in \mathbb{R}^d$ also carry the same subscripts and superscripts (e.g: $\tilde{X}_4$ corresponds to $\tilde{U}_4, \tilde{V}_4$). In this case $\mathbf{I}$ represents identity matrix in $\mathbb{R}^{2d \times 2d}$ and $\mathbf{I}_d$ denotes the identity matrix in $\mathbb{R}^{d \times d}$. For any random variable $X$, we let $\text{Law}(X)$ denote its probability measure. By $\text{TV}(\mu, \nu)$ and $\text{KL}(\mu \| \nu)$ we denote the total variation distance and KL divergence (respectively) between two probability measure $\mu, \nu$. $O, \Omega, \Theta$ are standard Kolmogorov complexity notations

---

[2]The prior works considered the compexity for Wasserstein distance $\mathcal{W}_2(\text{output}, \text{target}) \lesssim \epsilon$ whereas we consider $\text{TV} \le \epsilon$. This is a standard comparison in the sampling literature [53].

whereas $\tilde{O}, \tilde{\Omega}, \tilde{\Theta}$ are same as $O, \Omega, \Theta$ up to poly-logarithmic factors in the problem parameters such as $\frac{1}{\epsilon}, \frac{1}{\alpha}, K, T, d$.

## 2 Problem Setup

Given a random vector $X_0 \in \mathbb{R}^d$, consider an iterative, discrete time process $(X_t)_{t \in \mathbb{N} \cup \{0\}}$ over $\mathbb{R}^d$, with step-size $\alpha > 0$ given by:

$$X_{t+1} = A_\alpha X_t + G_\alpha b(X_t, t\alpha) + \Gamma_\alpha Z_t \tag{1}$$

Where $A_\alpha, G_\alpha, \Gamma_\alpha$ are $d \times d$ matrix valued functions of the step-size $\alpha$ and $(Z_t)_{t \geq 0} \overset{\text{i.i.d.}}{\sim} \mathcal{N}(0, \mathbf{I}_d)$. $b : \mathbb{R}^d \to \mathbb{R}^d$ is the drift. Call this process $\mathsf{S}(A, G, \Gamma, b, \alpha)$. We consider Overdamped Langevin Monte Carlo (OLMC), Underdamped Langevin Monte Carlo (ULMC) and DDPMs as key examples.

**Overdamped Langevin Monte Carlo:** Consider Overdamped Langevin Dynamics for some $F : \mathbb{R}^d \to \mathbb{R}$:

$$d\bar{X}_\tau = -\nabla F(\bar{X}_\tau)d\tau + \sqrt{2}dB_\tau \tag{2}$$

Here $B_\tau$ is the standard Brownian motion in $\mathbb{R}^d$. Under mild conditions on $F$ and $\bar{X}_0$, $\mathsf{Law}(\bar{X}_\tau) \overset{\tau \to \infty}{\to} \pi^*$ where $\pi^\star(X) \propto \exp(-F(X))$ is the stationary distribution.

Picking $A_\alpha = \mathbf{I}, G_\alpha = \alpha\mathbf{I}, \Gamma_\alpha = \sqrt{2\alpha}\mathbf{I}$ and $b(\mathbf{x}, t\alpha) = -\nabla F(\mathbf{x})$ in Equation (1) gives us Euler-Maruyama discretization of Overdamped Langevin Dynamics: $X_t$ in (1) approximates $\bar{X}_{\alpha t}$ [38, 35]. OLMC is the canonical algorithm for sampling and has been studied under assumptions such as log-concavity of $\pi^\star$ [8, 12, 11] or that $\pi^\star$ satisfies isoperimetric inequalities [45, 15, 33, 7, 1].

**Underdamped Langevin Monte Carlo** occurs in $2d$ dimensions. We write $X_t = [U_t \quad V_t]^\mathsf{T} \in \mathbb{R}^{2d}$ where $U_t \in \mathbb{R}^d$ is the position and $V_t \in \mathbb{R}^d$ is the velocity. Fix a damping factor $\gamma > 0$. We take:

$$A_h := \begin{bmatrix} \mathbf{I}_d & \frac{1}{\gamma}(1 - e^{-\gamma h})\mathbf{I}_d \\ 0 & e^{-\gamma h}\mathbf{I}_d \end{bmatrix}, \quad G_h := \begin{bmatrix} \frac{1}{\gamma}(h - \frac{1}{\gamma}(1 - e^{-\gamma h}))\mathbf{I}_d & 0 \\ \frac{1}{\gamma}(1 - e^{-\gamma h})\mathbf{I}_d & 0 \end{bmatrix} \quad b(X_t, t\alpha) := \begin{bmatrix} -\nabla F(U_t) \\ 0 \end{bmatrix}$$

$$\Gamma_h^2 := \begin{bmatrix} \frac{2}{\gamma}\left(h - \frac{2}{\gamma}(1 - e^{-\gamma h}) + \frac{1}{2\gamma}(1 - e^{-2\gamma h})\right)\mathbf{I}_d & \frac{1}{\gamma}(1 - 2e^{-\gamma h} + e^{-2\gamma h})\mathbf{I}_d \\ \frac{1}{\gamma}(1 - 2e^{-\gamma h} + e^{-2\gamma h})\mathbf{I}_d & (1 - e^{-2\gamma h})\mathbf{I}_d \end{bmatrix}$$

This choice of $A_h, \Gamma_h, G_h, b(,)$ in Equation (1) gives the Euler-Maruyama discretization of Underdamped Langevin Dynamics (a.k.a. the Kinetic Langevin Dynamics) studied extensively in Physics[14]):

$$d\bar{U}_\tau = \bar{V}_\tau d\tau; \qquad d\bar{V}_\tau = -\gamma\bar{V}_\tau - \nabla F(\bar{U}_\tau) + \sqrt{2\gamma}dB_\tau \tag{3}$$

The stationary distribution of the SDE is given by $\pi^\star(U, V) \propto \exp(-F(U) - \frac{\|V\|^2}{2})$ [13, 9]. ULMC is popular in the literature and has been analyzed in the strongly log-concave setting [6, 9, 16] and under isoperimetry conditions [30, 53]. We refer to [53] for a complete literature review.

**Denoising Diffusion Models:** In this case the stochastic differential equation is given by:

$$d\bar{X}_\tau = (\bar{X}_\tau + 2\nabla \log p_\tau(\bar{X}_\tau))d\tau + \sqrt{2}dB_\tau \tag{4}$$

This also admits an equivalent characteristic ODE given below:

$$\frac{d\bar{X}_\tau}{d\tau} = \bar{X}_\tau + \nabla \log p_\tau(\bar{X}_\tau) \tag{5}$$

See [43, 5, 4] for further details. Here $p_\tau$ is the probability density of $e^{-t}X^* + \sqrt{1 - e^{-2t}}Z$ where $X^*$ is drawn from the target and $Z$ is drawn from $\mathcal{N}(0, \mathbf{I})$ independently. The drift $\nabla \log p_\tau$ is learned via neural networks for discrete time instants $\tau_0, \ldots, \tau_{n-1}$ (usually $n = 1000$). In practice, the iterations are written in the form [3]: $X_{t+1} = a_t X_t + b_t \nabla \log p_{\tau_t}(X_t) + \sigma_t Z_t$ where $a_t, b_t, \sigma_t$ are chosen for best performance. Aside from the original choice in [20], many others have been proposed ([2, 42]). Since $A_\alpha, G_\alpha, \Gamma_\alpha$ in Equation (1) are time independent, we provide a variant of the Poisson Midpoint Method to suit DDPMs in Section A.1, along with a few other optimizations.

---

[3]The time convention in the DDPM literature is reverse: $X_{t-1} = a_t X_t + b_t \nabla \log p_{\tau_{n-1-t}}(X_t) + \sigma_t Z_t$.

## 2.1 Technical Notes

**Scaling Relations:** We impose the following scaling relations on the matrices $A_h, G_h, \Gamma_h$ for every $h \in \mathbb{R}^+, n \in \mathbb{N}$, which are satisfied by both OLMC and ULMC. Whenever $b(\cdot)$ is a constant function, these ensure that $K$ steps of Equation (1) with step-size $\alpha/K$ is the same as 1 step with step-size $\alpha$ in distribution.

$$(A_h)^n = A_{hn}; \quad \left(\sum_{i=0}^{n-1}(A_h)^i\right)G_h = G_{hn}; \quad \sum_{i=0}^{n-1}(A_h)^i\Gamma_h^2(A_h^{\mathsf{T}})^i = \Gamma_{hn}^2$$

**Randomized Midpoint Method and Stochastic Approximation** We first illustrate the Randomized Midpoint Method [40, 18, 50] by applying it to OLMC (to obtain RLMC) to motivate our method (the Poisson Midpoint Method) and explain why we expect a quadratic speed up shown in Section 3. Overdamped Langevin Dynamics (2) satisfies:

$$\bar{X}_{(t+1)\alpha} = \bar{X}_{t\alpha} - \int_{t\alpha}^{(t+1)\alpha} \nabla F(\bar{X}_s)ds + \sqrt{2}(B_{(t+1)\alpha} - B_{t\alpha}) \tag{6}$$

Taking $X_t$ as the approximation to $\bar{X}_{t\alpha}$, LMC approximates the integral $\int_{t\alpha}^{(t+1)\alpha} \nabla F(\bar{X}_s)ds$ with $\alpha\nabla F(X_t)$, giving a 'biased' estimator to the integral (conditioned on $X_t = \bar{X}_{t\alpha}$). This gives the LMC updates $X_{t+1} = X_t - \alpha\nabla F(X_t) + \sqrt{2\alpha}Z_t; \quad Z_t \sim \mathcal{N}(0, \mathbf{I})$. RLMC chooses a uniformly random point in the interval $[t\alpha, (t+1)\alpha]$ instead of initial point $t\alpha$ as described below:

Let $u \sim \mathsf{Unif}([0,1]), Z_{t,1}, Z_{t,2} \sim \mathcal{N}(0, \mathbf{I})$ be independent and define the midpoint $X_{t+u} := X_t - u\alpha\nabla F(X_t) + \sqrt{2u\alpha}Z_{t,1}$ (notice $X_{t+u}$ is an approximation for $\bar{X}_{(t+u)\alpha}$). The RLMC update is:

$$X_{t+1} = X_t - \alpha\nabla F(X_{t+u}) + \sqrt{2u\alpha}Z_{t,1} + \sqrt{2(1-u)\alpha}Z_{t,2} \,.$$

Notice that $\sqrt{2u\alpha}Z_{t,1} + \sqrt{2(1-u)\alpha}Z_{t,2}|u, X_t \sim \mathcal{N}(0, \mathbf{I})$, and $\mathbb{E}[\alpha\nabla F(X_{t+u})|X_t, Z_{t,1}, Z_{t,2}] = \alpha\int_0^1 F(X_{t+s})ds$ which is a better approximation of the integral than $\alpha\nabla F(X_t)$. Therefore RLMC provides a nearly unbiased approximation to the updates in Equation (6).

Intuitively, we expect that reducing the bias leads to a quadratic speed-up. Let $Z, Z' \sim \mathcal{N}(0,1)$ and independent. For $\epsilon$ small enough it is easy to show that, $\mathsf{KL}\left(\mathsf{Law}(Z+\epsilon)\middle\|\mathsf{Law}(Z)\right) = \Theta(\epsilon^2)$ whereas $\mathsf{KL}\left(\mathsf{Law}(Z+\epsilon Z')\middle\|\mathsf{Law}(Z)\right) = \Theta(\epsilon^4)$. We hypothesize that $Z_t + \text{error in integral}$ is closer to $\mathcal{N}(0, \mathbf{I})$ when the error term has a small mean and a large variance (as in RLMC) than when it has a large mean but $0$ variance (as in LMC). However, rigorous analysis of RLMC has only been done under assumptions like strong log-concavity of the target distribution. This is due to the fact that $Z_t$ is dependent on the error in the integral, disallowing the argument above.

Our method, PLMC, circumvents these issues by considering a discrete set of midpoints $\{0, \frac{1}{K}, \ldots, \frac{K-1}{K}\}$ instead of $[0,1]$. It picks each midpoint with probability $\frac{1}{K}$ independently, allowing us to prove results under more general conditions using the intuitive ideas described above. Thus, our method is a variant of RLMC which is amenable to more general mathematical analysis. The **OPTION 2** of our method (see below) makes this connection clearer. PLMC is naturally suited to DDPMs since the drift function is trained only for a discrete number of timesteps (see Section 2).

## 2.2 The Poisson Midpoint Method

We introduce the Poisson Midpoint Method (PLMC) which approximates $K$ steps of $\mathsf{S}(A, G, \Gamma, b, \frac{\alpha}{K})$ with step-size $\frac{\alpha}{K}$ with one step of which has a step-size $\alpha$. We denote this by $\mathsf{PS}(A, G, \Gamma, b, \alpha, K)$ and let its iterates be denoted by $(\tilde{X}_{tK})_{t\geq 0}$ or $(X_t^{\mathsf{P}})_{t\geq 0}$. Suppose $H_{t,i} \in \{0,1\}$ be any binary sequence and $Z_{tK+i}$ be a sequence of i.i.d. $\mathcal{N}(0, \mathbf{I})$ for $t, i \in \mathbb{N} \cup \{0\}, 0 \leq i \leq K-1$. Given $\tilde{X}_{tK}$, we define the interpolation:

$$\hat{\tilde{X}}_{tK+i} := A_{\frac{\alpha i}{K}}\tilde{X}_{tK} + G_{\frac{\alpha i}{K}}b(\tilde{X}_{tK}, t\alpha) + \sum_{j=0}^{i-1} A_{\frac{\alpha(i-j-1)}{K}}\Gamma_{\frac{\alpha}{K}}Z_{tK+j} \tag{7}$$

Note that this interpolation is cheap since every one of $\hat{\tilde{X}}_{tK+i}$ can be computed with just one evaluation of the function $b()$. We then define the refined iterates for $0 \leq i \leq K-1$ for a given $t$ as:

$$\tilde{X}_{tK+i+1} = A_{\frac{\alpha}{K}}\tilde{X}_{tK+i} + G_{\frac{\alpha}{K}}\left[b(\tilde{X}_{tK}, t\alpha) + KH_{t,i}(b(\hat{\tilde{X}}_{tK+i}, \frac{(tK+i)\alpha}{K}) - b(\tilde{X}_{tK}, t\alpha))\right] + \Gamma_{\frac{\alpha}{K}}Z_{tK+i} \tag{8}$$

We pick $H_{t,i}$ based on the following two options, independent of $Z_{tK+i}, \tilde{X}_0$:

**OPTION 1:** $H_{t,i}$ are i.i.d. $\mathsf{Ber}(\frac{1}{K})$.

**OPTION 2:** Let $u_t \sim \mathsf{Unif}(\{0, \ldots, K-1\})$ i.i.d. and $H_{t,i} := \mathbb{1}(u_t = i)$.

**Remark 1.** *We call our method Poisson Midpoint method since in **OPTION 1** the set of midpoints* $\{\alpha t + \frac{\alpha i}{K} : H_{t,i} = 1\}$ *converges to a Poisson process over* $[\alpha t, \alpha(t+1)]$ *as* $K \to \infty$.

**The Algorithm and Computational Complexity**   The algorithm $\mathsf{PS}(A, G, \Gamma, b, \alpha, K)$ computes $\tilde{X}_{K(t+1)}$ given $\tilde{X}_{Kt}$ in one step by unrolling the recursion given in Equation (8). For the sake of clarity, we will relabel $\tilde{X}_{tK}$ to be $X_t^\mathsf{P}$ to stress the fact that it is the $t$-th iteration of $\mathsf{PS}(A, G, \Gamma, b, \alpha, K)$.

**Step 1:** Generate $I_t = \{i_1, \ldots, i_N\} \subseteq \{0, \ldots, K-1\}$ such that $H_{t,i} = 1$ iff $i \in I_t$. Let $i_1 < i_2 \cdots < i_N$ hold. When $N = 0$, we take this to be the empty set.
**Step 2:** Let $M_0 := 0$ and let $W_{t,k}$ be a sequence of i.i.d. $\mathcal{N}(0, \mathbf{I})$ random vectors. For $k = 1, \ldots, N, N+1$, we take:

$$M_k = A_{\frac{\alpha(i_k - i_{k-1})}{K}}M_{k-1} + \Gamma_{\frac{\alpha(i_k - i_{k-1})}{K}}W_{t,k}$$

We use the convention that $i_0 = 0$, $i_{N+1} = K-1$, $A_0 = \mathbf{I}$ and $\Gamma_0 = 0$.
**Step 3:** For $k = 1, \ldots, N$, compute $\hat{\tilde{X}}_{tK+i_k} := A_{\frac{\alpha i_k}{K}}X_t^\mathsf{P} + G_{\frac{\alpha i_k}{K}}b(X_t^\mathsf{P}, \alpha t) + M_k$

**Step 4:** $\mathsf{Corr} := K\sum_{k=1}^{N} G_{\frac{(K-1-i_k)\alpha}{K}}(b(\hat{\tilde{X}}_{tK+i}, \frac{(tK+i)\alpha}{K}) - b(X_t^\mathsf{P}, \alpha t))$

**Step 5:** $X_{t+1}^\mathsf{P} = A_\alpha X_t^\mathsf{P} + G_\alpha b(X_t^\mathsf{P}, \alpha t) + M_{N+1} + \mathsf{Corr}$ \hfill (9)

That is, the algorithm first generates the random mid-points $H_{t,i}$, computes the interpolation $\hat{\tilde{X}}_{tK+i}$ only when $H_{t,i} = 1$ and then computes $b(\hat{\tilde{X}}_{tK+i}, \frac{(tK+i)\alpha}{K})$ for these points. These computations are then combined to compute $\tilde{X}_{(t+1)K}$. In most applications, it is computationally easy to generate Gaussian random vectors and perform vector operations such as summation. However, the evaluation of the drift function $b()$ is expensive. Therefore, in this work, we consider the number of evaluations of the drift function as the measure of computational complexity. The following proposition establishes that each iteration of PLMC requires 2 evaluations of $b()$ in expectation.

**Proposition 1.** *When the scaling relations hold (Section 2.1), the trajectory* $(X_t^\mathsf{P})_{t \geq 0}$ *in Equation* (9) *has the same joint distribution as the trajectory* $(\tilde{X}_{tK})_{t \geq 0}$ *given in Equation* (8). *In expectation, one step of* $\mathsf{PS}(A, G, \Gamma, b, \alpha, K)$ *requires two evaluations of the function* $b(\cdot)$.

We call $\mathsf{PS}(A, G, \Gamma, b, \alpha, K)$ as PLMC whenever $\mathsf{S}(A, G, \Gamma, b, \frac{\alpha}{K})$ is either OLMC or ULMC.

## 3   Main Results

Theorem 1 gives an upper bound for the KL divergence of the trajectory generated by $\mathsf{PS}(A, G, \Gamma, b, \alpha, K)$ to the one generated by $\mathsf{S}(A, G, \Gamma, b, \frac{\alpha}{K})$. We refer to Section C for its proof. We note that Theorem 1 does not make any mixing or smoothness assumptions on $b(\cdot)$ and that it can handle time dependent drifts. We refer to Section 4 for a proof sketch and discussion.

**Theorem 1.** *Let* $X_t$ *be the iterates of* $\mathsf{S}(A, G, \Gamma, b, \frac{\alpha}{K})$ *and* $X_t^\mathsf{P}$ *be the iterates of* $\mathsf{PS}(A, G, \Gamma, b, \alpha, K)$ *with **OPTION 1**. Suppose that* $X_0^\mathsf{P} = X_0$. *Let* $\tilde{X}_{tK+i}$ *be the iterates in Equation* (8). *Define random variables:*

$$B_{tK+i} := \Gamma_{\frac{\alpha}{K}}^{-1} G_{\frac{\alpha}{K}}[b(\hat{\tilde{X}}_{tK+i}, t\alpha + \frac{i\alpha}{K}) - b(\tilde{X}_{tK+i}, t\alpha + \frac{i\alpha}{K})]$$

$$\beta_{tK+i} := \|K\Gamma_{\frac{\alpha}{K}}^{-1} G_{\frac{\alpha}{K}}[b(\hat{\tilde{X}}_{tK+i}, t\alpha + \frac{\alpha i}{K}) - b(\tilde{X}_{tK}, t\alpha)]\|$$

*Then, for some universal constant C and any $r > 1$:*

$$\mathsf{KL}\left(\mathsf{Law}(X_{0:T}^{\mathsf{P}})\middle\|\mathsf{Law}((X_{Kt})_{0 \le t \le T})\right)$$

$$\le \sum_{s=0}^{T-1}\sum_{i=0}^{K-1}\mathbb{E}[\|B_{sK+i}\|^2] + C\mathbb{E}\left[\frac{\beta_{sK+i}^4}{K^2} + \frac{\beta_{sK+i}^{10}}{K^3} + \frac{\beta_{sK+i}^6}{K^2} + \frac{\beta_{sK+i}^{2r}}{K}\right] \tag{10}$$

We now apply Theorem 1 to the case of OLMC and ULMC under additional assumptions, with the proofs in Sections E and F respectively.

**Assumption 1.** $F : \mathbb{R}^d \to \mathbb{R}^d$ *is L-smooth (i.e., $\nabla F$ is L-Lipschitz). $\mathbf{x}^*$ is its global minimizer.*

**Assumption 2.** *The initialization $X_0$ is such that $\mathbb{E}\|X_0 - \mathbf{x}^*\|^{14} < C_{\mathsf{init}}^{14}d^7$.*

The assumptions above are very mild and standard in the literature. Specifically, Assumption 2 shows that the initialization is close to global optimum by $O(\sqrt{d})$ up to 14th moments. For instance, this is satisfied when the initialization is a standard Gaussian variable with mean $\mu$ satisfying $\|\mu - \mathbf{x}^*\| = O(\sqrt{d})$. Specifically this is true when $\mu = 0$ and $\|\mathbf{x}^*\| = O(\sqrt{d})$. This is a weak assumption which is implied from common initialization assumptions in the literature as listed below. It can be replaced with the assumptions in [53, Appendix D and Lemma 27] which considers Gaussian initializations with the right variance and mean. The original randomized midpoint method work [40] considers initializing at $\mathbf{x}^*$ whereas [45] considers a Gaussian initialization with the right variance at a local minimum of $F$.

We do not make any assumptions regarding isoperimetry of the target distribution $\pi^\star(\mathbf{x}) \propto \exp(-F(\mathbf{x}))$.

**Theorem 2** (OLMC)**.** *Consider the setting of Theorem 1 with OLMC under Assumptions 1 and 2. There exists constants $c_1, c_2 > 0$ such that whenever $\alpha L < c_1$ and $\alpha^3 L^3 T < c_2$ then:*

$$\mathsf{KL}\left(\mathsf{Law}(X_{0:T}^{\mathsf{P}})\middle\|\mathsf{Law}(X_{K(0:T)})\right) \le CL^4\alpha^4(\mathbb{E}[F(X_0) - F(\mathbf{x}^*)] + 1)$$
$$+ O(CL^4\alpha^4 K d^2 T) \tag{11}$$

**Remark 2.** *There are lower order terms hidden in the $O()$ notation. These are explicated in Equation (40) in the appendix. The next theorem gives a similar guarantee for ULMC and the lower order terms are explicated in Equation 60 in the appendix.*

**Theorem 3** (ULMC)**.** *Consider the setting of Theorem 1 with ULMC under Assumptions 1 and 2. Suppose that $\mathbf{x}^*$ is the global minimizer of $F$. There exist constants $C_1, c_1, c_2$ such that whenever $\gamma > C_1\sqrt{L}$, $\alpha\gamma < c_1$, $T < \frac{c_2\gamma}{L^2\alpha^3}$.*

$$\mathsf{KL}\left(\mathsf{Law}(X_{(0:T)}^{\mathsf{P}})\middle\|\mathsf{Law}(X_{K(0:T)})\right) \le \frac{C\alpha^6 L^4}{\gamma^2}\left[\mathbb{E}F(U_0 + \tfrac{V_0}{\gamma}) - F(\mathbf{x}^*) + \mathbb{E}\|V_0\|^2 + 1\right]$$
$$+ O\left(\frac{K\alpha^7 L^4 T^2}{\gamma}(d + \log K)^2\right) \tag{12}$$

OLMC and ULMC are sampling algorithms which output approximate samples ($X_T$ and $U_T$ respectively) from the distribution with density $\pi^\star \propto e^{-F}$. Given $\epsilon > 0$, prior works give upper bounds on $T$ and the corresponding step-size $\alpha$ as a function of $\epsilon$ to achieve guarantees such as $\mathsf{KL}\left(\mathsf{Law}(X_T)\middle\|\pi^\star\right) \le \epsilon^2$ or $\mathsf{TV}(\mathsf{Law}(X_T), \pi^\star) \le \epsilon$. By Pinsker's inequality $\mathsf{TV}^2 \le 2\mathsf{KL}$ therefore we guarantees for $\mathsf{KL} \le \epsilon^2$ to those for $\mathsf{TV} \le \epsilon$ as is common in the literature.

**Quadratic Speedup**  Let $T = \tilde{\Theta}(1/\alpha)$ as is standard. Choosing $K = \Theta(1/\alpha)$, our method applied to OLMC achieves a KL divergence of $O(\alpha^2)$ to OLMC with step-size $\alpha^2$. Similarly, our method applied to ULMC achieves a KL divergence of $O(\alpha^4)$ to ULMC with step-size $\alpha^2$. Whenever the KL divergence of OLMC (resp. ULMC) output to $\pi^\star$, with step-size $\eta$ is $\tilde{\Omega}(\eta)$ (resp $\tilde{\Omega}(\eta^2)$) Theorem 2 (resp. Theorem 3) demonstrates a quadratic speed up.

To show the generality of our results, we combine Theorems 2 and 3 with convergence results for OLMC /ULMC in the literature ([45, 53]) when $\pi^\star$ satisfies the Logarithmic Sobolev Inequality with constant $\lambda$ ($\lambda$-LSI). We obtain convergence bounds for the last iterate of PLMC to $\pi^\star$ under the same conditions. $\lambda$-LSI is more general than strong log-concavity ($\lambda$-strongly log-concave $\pi^\star$ satisfies $\lambda$-LSI). It is stable under bounded multiplicative perturbations of the density [21] and Lipschitz mappings. $\lambda$-LSI condition has been widely used to study sampling algorithms beyond log-concavity. We present our results in Table 1 and refer to Section G for the exact conditions and results.

Table 1: Comparison of LMC and PLMC guarantees. LMC complexity is the upper bound on the number of drift ($b()$) evaluations to achieve the error guarantee in the referenced work. PLMC complexity is the corresponding upper bound for PLMC. PLMC obtains a quadratic improvement in $\epsilon$, and improved dependence on $\frac{L}{\lambda}, d$. The bounds hold up to poly-log factors.

| Algorithm | Reference | Conditions | LMC complexity | PLMC complexity |
|---|---|---|---|---|
| ULMC | [53] | $\lambda$-LSI, Assumptions 1, 2 | $\frac{L^{\frac{3}{2}} d^{\frac{1}{2}}}{\lambda^{\frac{3}{2}} \epsilon}$ for TV $\leq \epsilon$ | $(\frac{L}{\lambda})^{\frac{17}{12}} \frac{d^{\frac{5}{12}}}{\sqrt{\epsilon}}$ for TV $\leq \epsilon$ |
| OLMC | [45] | $\lambda$-LSI, Assumptions 1, 2 | $\frac{L^2 d}{\lambda^2 \epsilon^2}$ for KL $\leq \epsilon^2$ | $\frac{L^{\frac{3}{2}} d^{\frac{3}{4}}}{\lambda^{\frac{3}{2}} \epsilon}$ for TV $\leq \epsilon$ |

## 4 Proof Sketch

**Sketch for Theorem 1**   For the proof of Theorem 1, we follow the recipe given in [10] in order to analyze the stochastic approximations of LMC - where only an unbiased estimator for the drift function is known. The bias variance decomposition in Lemma 1, shows that the iterations of $\mathsf{PS}(A, G, \Gamma, \alpha, K)$ can be written in the same form of as the iterations of $\mathsf{S}(A, G, \Gamma, \frac{\alpha}{K})$:

$$\tilde{X}_{tK+i+1} = A_{\frac{\alpha}{K}} \tilde{X}_{tK+i} + G_{\frac{\alpha}{K}} \left[ b(\tilde{X}_{tK+i}) \right] + \Gamma_{\frac{\alpha}{K}} \tilde{Z}_{tK+i}$$

Where $\tilde{Z}_{tK+i} := Z_{tK+i} + B_{tK+i} + S_{tK+i}$, $B_{tK+i}$ is the 'bias' with a non-zero conditional mean, and $S_{tK+i}$ is the variance with 0 conditional mean (conditioned on $\tilde{X}_{tK+i}$). They are independent of $Z_{tK+i}$ conditioned on $\tilde{X}_{tK+i}$. Note that the sequence $(\tilde{Z}_{tK+i})_{t,i}$ is neither i.i.d. nor Gaussian. If it was a sequence of i.i.d. $\mathcal{N}(0, \mathbf{I})$, then this is exactly same as $\mathsf{S}(A, G, \Gamma, \frac{\alpha}{K})$.

The main idea behind the proof of Theorem 1 is that due to data-processing inequality, it is sufficient to show that $(\tilde{Z}_{tK+i})_{t,i}$ is close to a sequence of i.i.d. Gaussian random vectors in KL-divergence. The bias term can be shown to lead to an error bounded by $\sum_{t,i} \mathbb{E} \|B_{tK+i}\|^2$, which roughly corresponds to the KL divergence between $\mathcal{N}(B_{tK+i}, \mathbf{I})$ and $\mathcal{N}(0, \mathbf{I})$. We then show that $Z_{tK+i} + S_{tK+i} | \tilde{Z}_{0:tK+i-1}$ is close in distribution to $\mathcal{N}(0, \mathbf{I})$. In order to achieve this, we first modify the Wasserstein CLT established in [51] to show that $Z_{tK+i} + S_{tK+i}$ is close in distribution to $\mathcal{N}(0, \mathbf{I} + \Sigma_{t,i})$ when conditioned on $\tilde{Z}_{0:tK+i-1}$ where $\Sigma_{t,i}$ is the conditional covariance of $S_{tK+i}$. This CLT step gives us the error of the form $\sum_{s=0}^{T-1} \sum_{i=0}^{K-1} C \mathbb{E} \left[ \frac{\beta_{sK+i}^{10}}{K^3} + \frac{\beta_{sK+i}^6}{K^2} + \frac{\beta_{sK+i}^{2r}}{K} \right]$ in Theorem 1.

We then use the standard formula for KL divergence between Gaussians to bound the distance between $\mathcal{N}(0, \mathbf{I} + \Sigma_{t,i})$ to $\mathcal{N}(0, \mathbf{I})$. This accounts for the fact that the Gaussian noise considered has a slightly higher variance than $\mathbf{I}$. This leads to the leading term $C \mathbb{E} \left[ \frac{\beta_{sK+i}^4}{K^2} \right]$.

**Sketch for Theorem 2**   Applying Theorem 1 to OLMC , note that the term $\beta_{tK_i}$ depends on how far the coarse estimate $\hat{\tilde{X}}_{tK+i}$ is from the true value $\tilde{X}_{tK+i}$. Indeed, under the smoothness assumption on $F$ we show that: $\beta_{tK+i}^{2p} \leq (\frac{L^2 \alpha K}{2})^p \sup_{0 \leq j \leq K-1} \|\hat{\tilde{X}}_{tK+j} - \tilde{X}_{tK}\|^{2p}$. Thus:

$$\mathbb{E} \|\beta_{tK+i}\|^{2p} \lesssim L^{2p} \alpha^{3p} K^p \mathbb{E} \|\nabla F(\tilde{X}_{tK})\|^{2p} + L^{2p} \alpha^{2p} K^p d^p .$$

Therefore, the proof reduces to bounding $\sum_{t=0}^{T-1} \mathbb{E} \|\nabla F(\tilde{X}_{tK})\|^{2p}$. We observe that $\tilde{X}_{(t+1)K} = \tilde{X}_{tK} - \alpha(\nabla F(\tilde{X}_{tK}) + \Delta_t) + \sqrt{2\alpha} Z_t$ where $\Delta_t$ is a small error term appearing due to Poisson Midpoint Method. Notice that this is approximately stochastic gradient descent on $F$ with a large noise $\sqrt{2\alpha}$. Therefore, using the taylor approximation of $F$, we can show that:

$$\mathbb{E} F(\tilde{X}_{(t+1)K}) - \mathbb{E} F(\tilde{X}_{tK}) \lesssim -\alpha \|\nabla F(\tilde{X}_{tK})\|^2 + \alpha d + o(\alpha d)$$

$$\implies \sum_{t=0}^{T-1} \mathbb{E} \|\nabla F(\tilde{X}_{tK})\|^2 \lesssim \frac{F(X_0) - \inf_{\mathbf{x}} F(\mathbf{x})}{\alpha} + LTd + o(LTd)$$

The following sophisticated bound derived in this work is novel to the best of our knowledge:

$$\sum_{t=0}^{T-1} \mathbb{E} \|\nabla F(\tilde{X}_{tK})\|^{2p} \lesssim L^{p-1} \mathbb{E} \frac{(F(X_0) - \inf_{\mathbf{x}} F(\mathbf{x}))^p}{\alpha} + TL^p d^p (1 + (\alpha LT)^{p-1})$$

**Sketch for Theorem 3** This is similar Theorem 2, but requires us to bound $\mathbb{E}\sum_t \|\tilde{V}_{tK}\|^{2p}$ and $\mathbb{E}\sum_t \|\nabla F(\tilde{U}_{tK})\|^{2p}$. We track the decay of two different entities across time: (1) $\|\tilde{V}_{tK}\|^2$ and (2) $F(\tilde{U}_{tK} + \frac{\tilde{V}_{tK}}{\gamma})$. Our proof shows via a similar taylor series based argument that PLMC does not allow either $\|\tilde{V}_{tK}\|^2$ or $F(\tilde{U}_{tK} + \frac{\tilde{V}_{tK}}{\gamma})$ to grow too large. Letting $\Psi_t := \tilde{U}_{tK} + \frac{\tilde{V}_{tK}}{\gamma}$ we show (roughly):

$$\sum_{t=0}^{T-1} \mathbb{E}\|\nabla F(\tilde{U}_{tK})\|^{2p} \lesssim \frac{\gamma^{2p}}{\gamma\alpha}\left[\mathbb{E}\|\tilde{V}_0\|^{2p} + \mathbb{E}|F(\Psi_0) - F(\mathbf{x}^*)|^p\right] + T\left[\frac{\gamma^{4p}}{L^p} + (\gamma\alpha T)^{p-1}\gamma^{2p}\right]d^p$$

$$\sum_{t=0}^{T-1} \mathbb{E}\|\tilde{V}_{tK}\|^{2p} \lesssim \frac{1}{\gamma\alpha}\left[\mathbb{E}\|\tilde{V}_0\|^{2p} + \mathbb{E}|(F(\Psi_0) - F(\mathbf{x}^*)|^p\right] + T\left[\frac{\gamma^{2p}}{L^p} + (\gamma\alpha T)^{p-1}\right]d^p$$

## 5 Experiments

We now present experiments to evaluate Poisson Midpoint Method as a training-free scheduler for diffusion models. We consider the Latent Diffusion Model (LDM) [39] for CelebAHQ 256, LSUN Churches, LSUN Bedrooms and FFHQ datasets using the official (PyTorch) codebase and checkpoints. We compare the sample quality of the Poisson Midpoint Method against established methods such as DDPM, DDIM and DPM-Solver, varying the number of neural network calls (corresponding to the drift $b(\mathbf{x}, t)$) used to generate a single image.

To evaluate the quality, we generate 50k images for each method and number of neural network calls and compare it with the training dataset. We use Fréchet Inception Distance (FID) [19] metric for LSUN Churches and LSUN Bedrooms. For CelebAHQ 256 and FFHQ, we use Clip-FID, a more suitable metric as it is known that FID may exhibit inconsistencies with human evaluations datasets outside of Imagenet [24]. We refer to Section A.3 in the appendix for further details.

We refer to the ODE based sampler with $\eta = 0$ (see [42]) setting as DDIM and use the implementation in [39]. We generate images for number of neural network calls ranging from 20 to 500. For DPM-Solver, we port the official codebase of [28] to generate images for different numbers of neural network calls ranging from 10 to 100 using MultistepDPMSolver and tune the hyperparameter 'order' over $\{2, 3\}$ and 'skip_type' over $\{$'logSNR', 'time_uniform', 'time_quadratic'$\}$ for each instance to obtain the best possible FID score. This ensures that the baseline is competitive.

For the sake of clarity, we will call all SDE based methods, including DDIM with $\eta > 0$ (see [42]) as DDPM. The DDPM scheduler has many different proposals for coefficients $a_t, b_t, c_t$ (see Section 2,[42, 2]), apart from the original proposal in the work of [20] . Based on these proposals, we consider three different variants of DDPM in our experiments (see Section A.5 for exact details). This choice can have a significant impact on the performance (See Figure 1 in the Appendix) for a given number of denoising diffusion steps. For the Poisson Midpoint Method, we implement the algorithm shown in Section A.1 for number of diffusion steps ranging from 20 to 500, corresponding to 40 to 750 neural network calls (see Section A.2). This approximates $K$ steps of the 1000 step DDPM with a single step. For both Poisson Midpoint Method and DDPM, we plot the results from the best variant in Table 2 for a given number of neural network calls and refer to Section A.5 for the numbers of all variants. Poisson Midpoint Method incurs additional noise in each iteration due to the randomness introduced by $H_i$. This can lead to a large error when $K$ is large. When $K$ is large, we reduce the variance of the Gaussian noise to compensate as suggested in the literature (see Covariance correction in [10] and [31, Equation 9]). We refer to Section A.5 for full details.

### 5.1 Results

We refer to the outcome of our empirical evaluations in Table 2. The first column compares the performance against DDPM. We see that for all the datasets considered, Poisson Midpoint Method can match the quality of the DDPM sampler with 1000 neural network calls with just 40-80 neural network calls. Observe that for CelebA, LSUN-Church and FFHQ datasets, the performance of DDPM degrades rapidly with lower number of steps, showing the advantage our method in this regime. However, a limitation of our work is that the quality of our method degrades rapidly at around 40-50 neural network calls. We believe this is because our stochastic approximation breaks down with larger step-sizes and further research is needed to mitigate this.

The second column compares the performance of our method against ODE based methods. It is known in the literature that DDPM with 1000 steps outperforms DDIM and DPM-Solver in terms of the quality for a large number of models and datasets [41, 42]. Thus, in terms of quality, Poisson midpoint method with just 50-80 neural network calls outperforms ODE based methods with a similar amount of compute. Note that we optimize the performance of DPM-Solver over 6 different variants as mentioned above to maintain a fair comparison. However, in the very low compute regime (~10 steps), DPM-Solver remains the best choice.

## 6    Conclusion

We introduce the Poisson Midpoint Method, which efficiently discretizes Langevin Dynamics and theoretically demonstrates quadratic speed up over Euler-Maruyama discretization under general conditions. We apply our method to diffusion models for image generation, and show that our method maintains the quality of 1000 step DDPM with just 50-80 neural network calls. This outperforms ODE based methods such as DPM-Solver in terms of quality, with a similar amount of compute. Future work can explore variants of Poisson midpoint method with better performance when fewer than 50 neural network calls are used. An interesting theoretical direction would be to derive convergence bounds for algorithms such as DDPM which have a time dependent drift function. Future work can also consider convergence rates of PLMC under conditions such as the Poincare Inequality and whenever $\nabla F$ is Hölder continuous instead of Lipschitz continuous.

## 7    Societal Impact

Our work considers an efficient numerical discretization schemes for making diffusion model inference more efficient. Publicly available, pre-trained diffusion models are very impactful and have significant risk of abuse. In addition to theoretical guarantees, our work considers empirical experiments to evaluate the inference efficiency on publicly available, widely used diffusion models over curated datasets. We do not foresee any significant positive or negative social impact of our work.

Table 2: Empirical Results for the Latent Diffusion Model [39], comparing the Poisson midpoint method with various SDE and ODE based methods.

| Dataset | vs. SDE Based Methods | vs. ODE Based Methods |
|---|---|---|

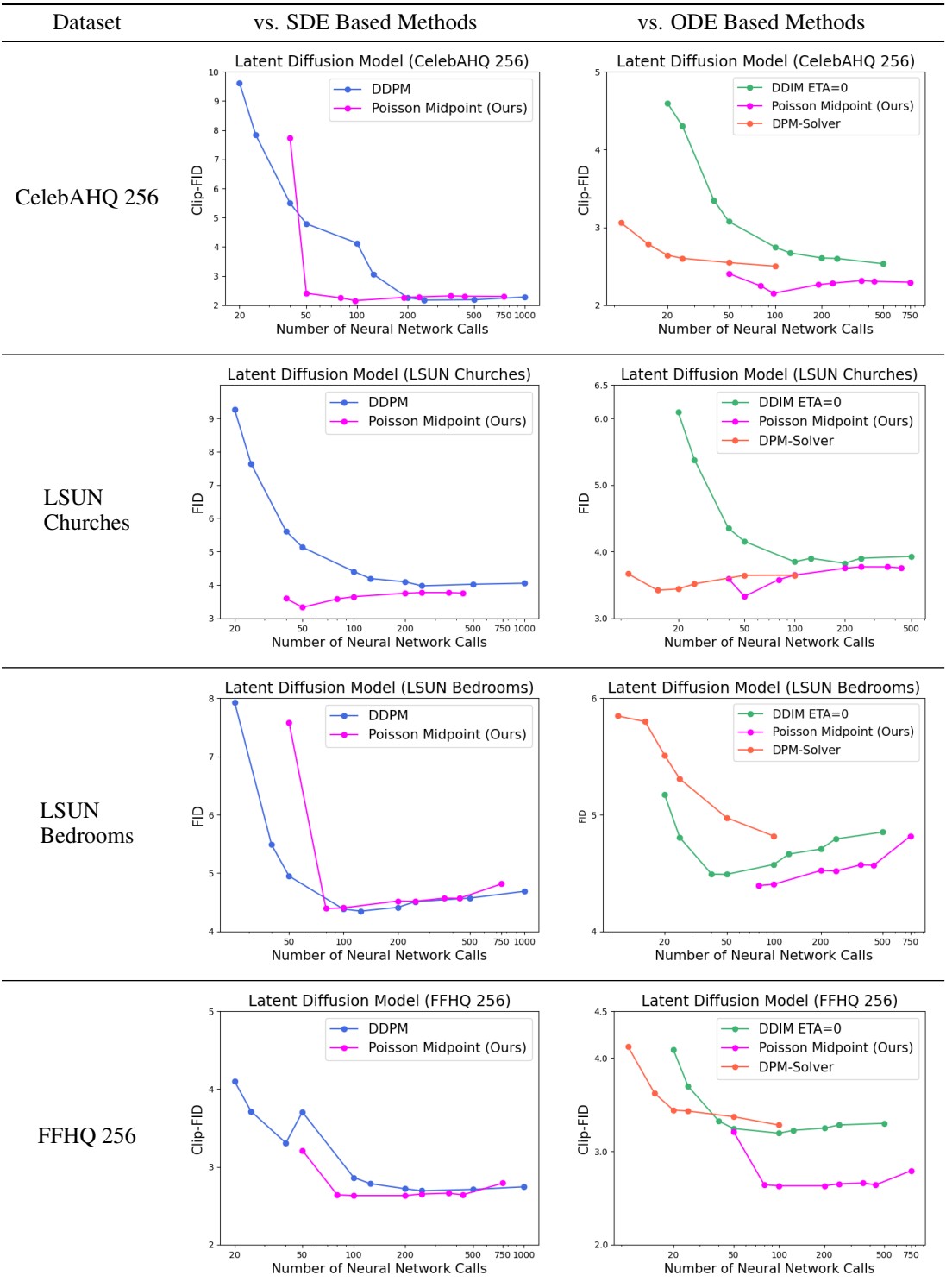

## Acknowledgments and Disclosure of Funding

We thank Prateek Jain for helpful comments during the course of this research work. This work was done when SK was a student researcher at Google Research.

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

# A  Details of Empirical Evaluations

## A.1  Pseudocode

**The Algorithm**   Let $a_t$, $b_t$ and $\sigma_t$ be the coefficients of the DDPM denoising step at time $t$ as per Section 2. Let $N$ denote the number of train steps. The main paper used notations and conventions based on the LMC literature. Here we follow different conventions and notations to connect with the diffusion model literature.

1. We take the dynamics to go backward in time (i.e., compute $X_{t-1}$ from $X_t$)
2. We use $b(X_t, t)$ instead of $b(X_t, \alpha t)$ to denote the neural network estimate of $\nabla \log p_{\tau_{N-t-1}}(X_t)$.

DDPM scheduler samples $X_{N-1} \sim \mathcal{N}(0, \mathbf{I})$ and iteratively computes $X_0$ as:
$$X_{t-1} = a_t X_t + b_t b(X_t, t) + \sigma_t Z_t; \quad Z_t \sim \mathcal{N}(0, \mathbf{I})$$

We take $N = 1000$ as is standard in the literature. The Poisson Midpoint Method approximates $K$ steps of the iteration above, with a single step. That is, it obtains an approximation for $X_{t-K}$ directly from $X_t$. The number of iterations deployed by Poisson Midpoint Method to sample $X_0$ is $N/K$. We compute the number of neural network calls required in Section A.2. The exact description of Poisson Midpoint Method is given below.

**PoissonMidpointMethod**($\tilde{X}_t, t, K, b(,), \text{OPTION}$):

1. Let $(A_k, B_k, C_k) \leftarrow$ **InterpolationConstants**$(t, k)$, for every $k \in [K]$.
2. If OPTION = 1:

    Define $H_i \sim \mathsf{Ber}(1/K)$ i.i.d for every $i \in [K-1]$.
    Define $p = K$.

    If OPTION = 2:

    Define $H_i \leftarrow \mathbb{1}(u = i)$ where $u \sim \mathsf{Unif}(\{1, \ldots, K-1\})$ i.i.d.
    Define $p = K - 1$.
3. For every $i \in [K]$, $Z_i \sim \mathcal{N}(0, \mathbf{I})$.
4. Perform the update:

$$\tilde{X}_{t-K} = A_K \tilde{X}_t + \left( \sum_{i=1}^{K} B_{K,i} \right) b(\tilde{X}_t, t) + \left( \sum_{i=1}^{K} C_{K,i} Z_i \right)$$
$$+ \sum_{i=1}^{K} p H_i B_{K,i} \big( b(\widehat{X}_{t-i+1}, t-i+1) - b(\tilde{X}_t, t) \big)$$

where

$$\widehat{X}_{t-\tau} = A_\tau \tilde{X}_t + \left( \sum_{i=1}^{\tau} B_{\tau,i} \right) b(\tilde{X}_t, t) + \sum_{j=1}^{\tau} C_{\tau, K-j+1} Z[j], \quad \forall \tau \in [K-1].$$

5. **Return** $\tilde{X}_{t-K}$.

**InterpolationConstants**($t, k$):

1. $A_k \leftarrow a_t \cdot a_{t-1} \cdot a_{t-k+1}$.
2. $B_{k,i} = \begin{cases} a_{t-k+1} \cdots a_{t-i} \cdot b_{t-i+1}, & \text{if } i < k \\ b_{t-k+1}, & \text{if } i = k \end{cases} \qquad$ for each $i \in [k]$.
3. $C_{k,i} = \begin{cases} a_{t-k+1} \cdots a_{t-i} \cdot \sigma_{t-i+1}, & \text{if } i < k \\ \sigma_{t-k+1}, & \text{if } i = k \end{cases} \qquad$ for each $i \in [k]$.
4. **Return** $(A_k, B_k, C_k)$.

**Remark 3.** *In the theoretically analyzed algorithm given in Section 2.2, we take the midpoints to be* $\{0, \frac{1}{K}, \ldots, \frac{K-1}{K}\}$. *But here, we take the midpoints to be* $\{\frac{1}{K}, \ldots, \frac{K-1}{K}\}$ *since the* $\widehat{X}_t = \tilde{X}_t$, *making the correction term* $p H_i B_{K,i} \big( b(\widehat{X}_{t-i+1}, t-i+1) - b(\tilde{X}_t, t) \big) = 0$ *when* $i = 0$.

## A.2 Number of Neural Network Calls

While each step of Poisson Midpoint Method approximates $K$ steps of DDPM, it can use more than one neural network calls. Since neural networks calls are the most computationally expensive parts of diffusion models (4 orders of magnitude larger FLOPs compared to other computations), we compute the number of neural network calls for the Poisson Midpoint Method given $K, N$ for both **OPTION 1** and **OPTION 2**.

For **OPTION 1**, $b(\tilde{X}_t, t)$ is always computed. $b(\hat{X}_{t-i+1}, t-i+1)$ needs to be computed iff $H_i = 1$. It is easy to show that $\mathbb{E}|\{i \in [K-1] : H_i = 1\}| = \frac{K-1}{K}$. Thus, the expected number of neural network calls per step is $2 - \frac{1}{K}$. The total number number of neural network calls is $\frac{N}{K}(2 - \frac{1}{K})$.

**OPTION 2** always computes evaluates $b()$ twice per step. Therefore, the number of neural network calls is $\frac{2N}{K}$.

## A.3 FID Evaluation Details

FID scores between two image datasets can vary greatly depending on the image processing details which do not have much bearing on visual characteristics (such as png vs jpeg) [37]. Thus, we utilize the standardized Clean-FID codebase [37] to compute the scores for all the datasets. As is standard, we generate 50k images for all our evaluations. For CelebAHQ 256, we evaluate the CLIP-FID scores against the combined 30k training and validation samples of the CelebA 1024x1024 dataset [22], where the CLIP-FID is invoked by using the flags *model_name = clip_vit_b_32* and *mode = clean* [37]. For FFHQ 256, we utilize the precomputed statistics of [37] on the 1024 resolution trainval70k split with the flags *dataset_split = trainfull* and *mode=clean*. As for LSUN Churches [49], we utilize the precomputed statistics of [37] with the flags *dataset_split = trainfull* and *mode=clean* to calculate the FID. And for LSUN Bedrooms [49], we downloaded the training split of [39] which consists of 3,033,042 images and used the codebase of [37] to compute the FID between two folders.

## A.4 Hardware Description and Execution Time

For our experiments, we utilized a high-performance computational setup consisting of 8 NVIDIA A100-SXM4 GPUs, each with 40 GB of VRAM, an Intel Xeon CPU with 96 cores operating at 2.2 GHz, and 1.3 TiB of RAM. The experiments take about 16 hours to generate 50k images with 1000 neural network calls per image and batch size of 50. The time is proportionally lower when fewer neural network calls are used.

## A.5 DDPM Variant Details

We describe 3 different variants for DDPM, based on 3 different choices of coefficients $a_t, b_t, \sigma_t$ used in Section A.1. Our observations indicate that each variant performs optimally for different datasets and for different ranges of the diffusion steps. For example, consider the DDPM variants of LSUN Churches in Figure 1b. Note that, Variant 3 outperforms Variant 2 for all steps less than or equal to 125 and vice versa for larger steps. Similar phenomenon can be observed for CelebAHQ 256 as shown in Figure 1a, Variant 2 outperforms the rest of the variants for steps smaller than 125, while Variant 1 achieves the best performance for larger steps. For FFHQ 256, similar phase transitions can be observed in Figure 1c at around 50-100 steps. For LSUN Bedrooms, we observed that Variant 2 outperforms 1 and 3 on all the regimes. Our experiments were carried out on all the variants and on all the steps, and the best scores are considered for comparison with Poisson Midpoint Method in Table 2. Let $\alpha_t, \beta_t, \bar{\alpha}_t$ be as defined in [20].

**Variant 1:** The first variant uses the following closed form expression to perform the update.

$$X_{t+1} = \frac{1}{\sqrt{\alpha_t}} X_t - 2 \cdot \frac{(1 - \sqrt{\alpha_t})}{\sqrt{\alpha_t}(1 - \overline{\alpha_t})} \cdot b(X_t, t) + \sigma_t \cdot Z_t \quad \text{where } \sigma_t^2 = \frac{\beta_t}{1 - \beta_t}. \quad (13)$$

**Variant 2:** The second variant of DDPM uses the default coefficients of DDIM as provided in the codebase of [39], with $\eta = 1$ and $\sigma_t^2 = \frac{(1-\overline{\alpha}_t)\cdot\beta_{t+1}}{(1-\overline{\alpha}_{t+1})}$.

**Variant 3:** The third variant of DDPM also uses the default coefficients of DDIM with $\eta = 1$, but uses a modified lower variance $\sigma_t^2 = (1 - \overline{\alpha}_t) \cdot \beta_{t+1}$.

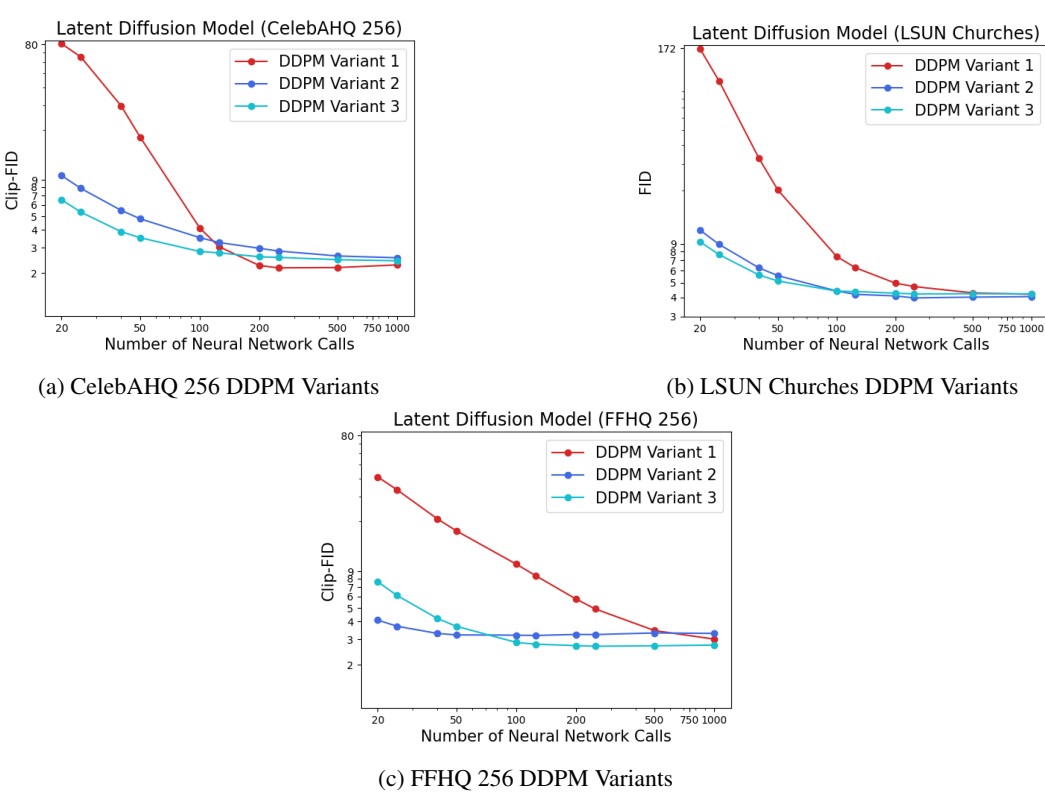

(a) CelebAHQ 256 DDPM Variants  (b) LSUN Churches DDPM Variants

(c) FFHQ 256 DDPM Variants

Figure 1: Comparison of different variants of DDPM

| Poisson Midpoint Variant | Number of Diffusion Steps / Number of Neural Network Calls | | | | | | | | |
|---|---|---|---|---|---|---|---|---|---|
| | OPTION 2 | | | OPTION 1 | | | | | |
| | 20 / 40 | 25 / 50 | 40 / 80 | 50 / 97.5 | 100 / 190 | 125 / 234.375 | 200 / 360 | 250 / 437.5 | 500 / 750 |
| Variant 1a | 14.57 | 3.29 | **2.24** | 2.21 | **2.26** | **2.28** | **2.31** | **2.30** | **2.29** |
| Variant 1b | 11.85 | 2.75 | 2.47 | **2.15** | 2.48 | 2.54 | 2.52 | 2.50 | 2.44 |
| Variant 1c | 9.65 | 2.44 | 2.74 | 2.27 | 2.76 | 2.80 | 2.85 | 2.86 | 2.77 |
| Variant 1d | **7.72** | **2.40** | 3.13 | 2.45 | 3.11 | 3.15 | 3.21 | 3.19 | 3.12 |
| Variant 2 | 12.34 | 2.90 | 2.53 | 2.23 | 2.52 | 2.55 | 2.61 | 2.59 | 2.47 |
| Variant 3 | 12.99 | 3.27 | 2.56 | 2.60 | 2.55 | 2.51 | 2.49 | 2.47 | 2.38 |

Table 3: CelebAHQ 256 – Comparison of Poisson Midpoint Variants

## A.6 Poisson Midpoint Variant Details

Similar to DDPM, we perform our experiments for different choice of coefficients $a_t, b_t, \sigma_t$. Variants 2 and 3 of Poisson Midpoint are defined with the same coefficients of the corresponding DDPM variants 2 and 3. Poisson Midpoint Method introduces additional noise due to the randomness present in $H_i$. This becomes very large whenever $K$ is large. Hence we reduce this randomness by reducing the variance of the Gaussian as suggested by [31, 10]. Specifically, for Variant 1, we consider four sub-variants 1a, 1b, 1c, 1d that respectively correspond to variances $\sigma_t^2 = \frac{\beta_t}{1+i\cdot\beta_t}$ for $i = \{-1, 0, 1, 2\}$, with the rest of the coefficients $a_t, b_t$ defined as in (13). Note that the variance is inversely proportional to $i$.

| Poisson Midpoint Variant | Number of Diffusion Steps / Number of Neural Network Calls | | | | | | | | |
|---|---|---|---|---|---|---|---|---|---|
| | OPTION 2 | | | | | | OPTION 1 | | |
| | 20 / 40 | 25 / 50 | 40 / 80 | 50 / 100 | 100 / 200 | 125 / 250 | 200 / 360 | 250 / 437.5 | 500 / 750 |
| Variant 1a | 12.44 | 9.18 | 6.04 | 5.30 | 4.12 | 4.14 | 4.07 | 4.05 | 4.41 |
| Variant 1b | 12.60 | 9.52 | 6.38 | 5.50 | 4.47 | 4.31 | 4.24 | 4.27 | 4.69 |
| Variant 1c | 12.66 | 9.92 | 6.76 | 5.99 | 4.79 | 4.61 | 4.61 | 4.58 | 4.94 |
| Variant 1d | 13.04 | 10.39 | 7.22 | 6.45 | 5.09 | 4.91 | 4.93 | 4.85 | 5.27 |
| Variant 2 | 11.18 | 8.21 | 5.45 | 4.75 | 3.81 | 3.77 | 3.77 | **3.75** | **4.22** |
| Variant 3 | **3.59** | **3.33** | **3.58** | **3.65** | **3.75** | **3.71** | **3.77** | 3.84 | 4.45 |

Table 4: LSUN Churches – Comparison of Poisson Midpoint Variants

| Poisson Midpoint Variant | Number of Diffusion Steps / Number of Neural Network Calls | | | | | | | | |
|---|---|---|---|---|---|---|---|---|---|
| | OPTION 2 | | | | | | OPTION 1 | | |
| | 20 / 40 | 25 / 50 | 40 / 80 | 50 / 100 | 100 / 200 | 125 / 250 | 200 / 360 | 250 / 437.5 | 500 / 750 |
| Variant 1a | 54.92 | 16.87 | 6.01 | 5.31 | 5.14 | 5.21 | 5.36 | 5.33 | 5.61 |
| Variant 1b | 47.03 | 13.46 | 4.98 | 4.78 | 4.69 | 4.79 | 4.72 | 4.81 | 4.97 |
| Variant 1c | 38.58 | 9.67 | 4.73 | 4.73 | 4.87 | 4.91 | 4.85 | 4.87 | 4.96 |
| Variant 1d | **31.33** | **7.57** | 5.07 | 5.47 | 5.58 | 5.65 | 5.50 | 5.58 | 5.57 |
| Variant 2 | 46.52 | 12.47 | 4.69 | 4.46 | **4.52** | **4.51** | **4.56** | **4.56** | **4.81** |
| Variant 3 | 39.82 | 9.17 | **4.39** | **4.40** | 4.55 | 4.60 | 4.66 | 4.65 | 4.98 |

Table 5: LSUN Bedrooms – Comparison of Poisson Midpoint Variants

We observe that lower variance leads to faster convergence when the number of diffusion steps is small. However, higher variances yield slightly better scores at higher steps, better quality at the expense of increased neural network evaluations. This phenomenon can be observed in Tables 3,4, 5 and 6 where the optimal score for each step-size/neural network calls is highlighted in bold. Therefore, we evaluate samples on all of the above mentioned variants and choose the best variant for every number of steps/neural network calls. We see that whenever $K$ is small (2-4), then **OPTION 1** uses fewer neural network calls than **OPTION 2**. However, the without replacement sampling technique in **OPTION 2** incurs lower variance error in the updates, allowing better convergence with lower number of steps (See Section A.2). Thus, we use **OPTION 2** for smaller number of steps and **OPTION 1** for larger number of steps.

## B  DPM-Solver Variant Details

For unconditional image generation of high-resolution images, third order (order=3) Multistep DPM-Solvers with uniform time steps are recommended. To ensure a competitive benchmark, we evaluate the DPM-Solver on six different settings by tuning the parameters 'order' and 'skip_type'. Our evaluations are shown in Table 7, where the best score for every setting, highlighted in bold, are chosen in our plots for comparison. We use the official code of [37] with parameters 'algorithm_type = dpmsolver' 'method = multistep' and vary the parameters 'order' and 'skip_type' accordingly.

| Poisson Midpoint Variant | Number of Diffusion Steps / Number of Neural Network Calls | | | | | | | | |
|---|---|---|---|---|---|---|---|---|---|
| | OPTION 2 | | | | | | OPTION 1 | | |
| | 20 / 40 | 25 / 50 | 40 / 80 | 50 / 100 | 100 / 200 | 125 / 250 | 200 / 360 | 250 / 437.5 | 500 / 750 |
| Variant 1a | 16.08 | 5.27 | 2.96 | 2.82 | 2.83 | 2.82 | 2.89 | 2.90 | 2.97 |
| Variant 1b | 13.26 | 4.31 | 2.79 | 2.77 | 2.81 | 2.80 | 2.82 | 2.85 | 2.91 |
| Variant 1c | 11.07 | 3.53 | 2.80 | 2.88 | 2.90 | 2.95 | 2.94 | 2.95 | 3.01 |
| Variant 1d | **9.39** | **3.21** | 3.04 | 3.15 | 3.22 | 3.24 | 3.21 | 3.18 | 3.24 |
| Variant 2 | 13.79 | 4.44 | 2.66 | **2.63** | **2.63** | **2.65** | **2.66** | **2.64** | **2.79** |
| Variant 3 | 13.65 | 4.11 | **2.64** | 2.65 | 2.76 | 2.78 | 2.80 | 2.82 | 2.98 |

Table 6: FFHQ 256 – Comparison of Poisson Midpoint Variants

## C   Proof of Theorem 1

The proof relies on the Wasserstein CLT based approach introduced in [10] to compare two discrete time stochastic processes. For the sake of clarity we consider the drift function $b(x, t)$ to be $b(x)$ (i.e., time invariant). However, the proof goes through for time varying drifts as well.

### C.1   The Bias-Variance Decomposition:

In the Lemma below, we will rewrite the update equations for $\tilde{X}_{tK+i}$ given in Equation (8) in the same form as the update equations for $\mathsf{S}(A, G, \Gamma, \frac{\alpha}{K})$ given in Equation (1). The lemma follows by re-arranging the terms in Equation (8).

**Lemma 1.** *We can write*

$$\tilde{X}_{tK+i+1} = A_{\frac{\alpha}{K}} \tilde{X}_{tK+i} + G_{\frac{\alpha}{K}} \left[ b(\tilde{X}_{tK+i}) \right] + \Gamma_{\frac{\alpha}{K}} \tilde{Z}_{tK+i}$$

*Where $\tilde{Z}_{tK+i} := Z_{tK+i} + B_{tK+i} + S_{tK+i}$ such that 'bias' $B_{tK+i}$ is defined as*

$$B_{tK+i} := \Gamma_{\frac{\alpha}{K}}^{-1} G_{\frac{\alpha}{K}} [b(\hat{\tilde{X}}_{tK+i}) - b(\tilde{X}_{tK+i})]$$

*and the variance $S_{tK+i}$ is defined as:*

$$S_{tK+i} := K(H_{t,i} - \tfrac{1}{K})\Gamma_{\frac{\alpha}{K}}^{-1} G_{\frac{\alpha}{K}} [b(\hat{\tilde{X}}_{tK+i}) - b(\tilde{X}_{tK})]$$

### C.2   Random Function Representation

By Equation (1), the iterates of $\mathsf{S}(A, G, \Gamma, \frac{\alpha}{K})$ satisfy:

$$X_{tK+i+1} = A_{\frac{\alpha}{K}} X_{tK+i} + G_{\frac{\alpha}{K}} \left[ b(X_{tK+i}) \right] + \Gamma_{\frac{\alpha}{K}} Z_{tK+i} \tag{14}$$

By Lemma 1, we can write down the interpolating process as:

$$\tilde{X}_{tK+i+1} = A_{\frac{\alpha}{K}} \tilde{X}_{tK+i} + G_{\frac{\alpha}{K}} \left[ b(\tilde{X}_{tK+i}) \right] + \Gamma_{\frac{\alpha}{K}} \tilde{Z}_{tK+i} \tag{15}$$

Note that while $Z_{tK+i}$ are i.i.d. $\mathcal{N}(0, \mathbf{I})$ vectors, $\tilde{Z}_{tK+i}$ can be non-Gaussian and non-i.i.d. Below we will show that the KL divergence between the joint laws of $(X_{tK+i})_{t,i}$ and $(\tilde{X}_{tK+i})_{t,i}$ can be bounded by bounding the KL divergence between $(Z_{tK+i})_{t,i}$ and $(\tilde{Z}_{tK+i})_{t,i}$. We now state the following standard results from information theory.

**Lemma 2** (Chain Rule for KL divergence). *Let $p, q$ be two probability distributions over $\mathcal{X} \times \mathcal{Y}$ where $\mathcal{X}$ and $\mathcal{Y}$ are polish spaces. Let $p_x, q_x$ denote their respective marginals over $\mathcal{X}$ and let $p_{y|x}, q_{y|x}$ denote the conditional distribution over $\mathcal{Y}$ conditioned on $x \in \mathcal{X}$. Then,*

$$\mathsf{KL}\left(p \middle\| q\right) = \mathsf{KL}\left(p_x \middle\| q_x\right) + \mathbb{E}_{x \sim p_x} \mathsf{KL}\left(p_{y|x} \middle\| q_{y|x}\right)$$

Table 7: Comparison of DPM Solver Variants

CelebAHQ 256

| Neural Network Calls | Order = 2 | | | Order = 3 | | |
| --- | --- | --- | --- | --- | --- | --- |
| | Uniform | logSNR | Quadratic | Uniform | logSNR | Quadratic |
| 10 | **3.06** | 4.05 | 4.58 | 3.07 | 3.44 | 4.31 |
| 15 | 2.90 | 3.13 | 3.35 | **2.79** | 2.82 | 3.16 |
| 20 | 2.78 | 2.83 | 2.99 | **2.64** | 2.66 | 2.84 |
| 25 | 2.72 | 2.70 | 2.83 | **2.60** | 2.62 | 2.72 |
| 50 | 2.57 | 2.56 | 2.60 | 2.57 | **2.55** | 2.56 |
| 100 | 2.53 | 2.53 | 2.55 | 2.55 | **2.50** | 2.55 |

LSUN Churches 256

| Neural Network Calls | Order = 2 | | | Order = 3 | | |
| --- | --- | --- | --- | --- | --- | --- |
| | Uniform | logSNR | Quadratic | Uniform | logSNR | Quadratic |
| 10 | 3.72 | 8.92 | 7.62 | **3.67** | 7.31 | 7.60 |
| 15 | **3.42** | 5.99 | 5.60 | 3.48 | 5.24 | 5.33 |
| 20 | **3.44** | 5.09 | 4.84 | 3.55 | 4.56 | 4.64 |
| 25 | **3.52** | 4.64 | 4.62 | 3.63 | 4.29 | 4.33 |
| 50 | 3.88 | 4.16 | 4.19 | **3.64** | 4.06 | 3.97 |
| 100 | 4.03 | 4.07 | 4.06 | **3.65** | 3.99 | 4.04 |

LSUN Bedrooms 256

| Neural Network Calls | Order = 2 | | | Order = 3 | | |
| --- | --- | --- | --- | --- | --- | --- |
| | Uniform | logSNR | Quadratic | Uniform | logSNR | Quadratic |
| 10 | **5.85** | 10.90 | 9.58 | 6.40 | 9.50 | 9.36 |
| 15 | **5.80** | 7.07 | 6.82 | 5.91 | 6.44 | 6.50 |
| 20 | 5.78 | 6.07 | 5.90 | **5.51** | 5.63 | 5.64 |
| 25 | 5.65 | 5.68 | 5.55 | 5.35 | **5.31** | 5.33 |
| 50 | 5.40 | 5.07 | 5.08 | 5.02 | **4.95** | 5.01 |
| 100 | 5.13 | 4.98 | 4.98 | **4.82** | 4.93 | 4.93 |

FFHQ 256

| Neural Network Calls | Order = 2 | | | Order = 3 | | |
| --- | --- | --- | --- | --- | --- | --- |
| | Uniform | logSNR | Quadratic | Uniform | logSNR | Quadratic |
| 10 | 4.41 | 4.62 | 6.05 | 4.19 | **4.12** | 5.73 |
| 15 | 4.23 | 3.77 | 4.48 | 3.80 | **3.62** | 4.26 |
| 20 | 4.03 | 3.55 | 3.98 | 3.61 | **3.44** | 3.78 |
| 25 | 3.93 | 3.49 | 3.77 | 3.53 | **3.43** | 3.61 |
| 50 | 3.68 | 3.40 | 3.47 | **3.37** | 3.38 | 3.43 |
| 100 | 3.50 | 3.37 | 3.40 | **3.28** | 3.37 | 3.37 |

We refer to [44, Lemma 4.18] for a proof.

**Lemma 3** (Data Processing Inequality). *Let $F : \mathbb{R}^{k_1} \to \mathbb{R}^{k_2}$ be any measurable function. Let $P, Q$ be any probability distributions over $\mathbb{R}^{k_1}$. Then, the following inequality holds:*

$$\mathsf{KL}\left(F_{\#}P \middle\| F_{\#}Q\right) \leq \mathsf{KL}\left(P \middle\| Q\right)$$

The lemma below follows from the fact that Equation (14) and Equation (15) have the same functional form.

**Lemma 4.** *Suppose* $\mathsf{S}(A, G, \Gamma, \frac{\alpha}{K})$ *and* $\mathsf{PS}(A, G, \Gamma, \alpha, K)$ *are initialized at the same point* $X_0$. *That is,* $X_0 = \tilde{X}_0$. *Then, there exists a measurable function* $F_T$ *such that the following hold almost surely.*

$$(X_\tau)_{0 \leq \tau \leq T} = F_T(X_0, Z_0, \ldots, Z_{T-1})$$

*and*

$$(\tilde{X}_\tau)_{0 \leq \tau \leq T} = F_T(X_0, \tilde{Z}_0, \ldots, \tilde{Z}_{T-1})$$

By Lemma 3 and Lemma 4, we have:

$$\mathsf{KL}\left(\mathsf{Law}((\tilde{X}_\tau)_{0 \leq \tau \leq T})\big|\big|\mathsf{Law}((X_\tau)_{0 \leq \tau \leq T})\right)$$

$$\leq \mathsf{KL}\left(\mathsf{Law}(X_0, \tilde{Z}_{0:T-1})\big|\big|\mathsf{Law}(X_0, Z_{0:T-1})\right)$$

$$= \mathbb{E}_{X_0}\mathsf{KL}\left(\mathsf{Law}(\tilde{Z}_0|X_0)\big|\big|\mathsf{Law}(Z_0)\right) +$$

$$\sum_{\tau=1}^{T-1} \mathbb{E}_{(X_0, \tilde{Z}_{0:\tau-1})}\mathsf{KL}\left(\mathsf{Law}(\tilde{Z}_\tau|\tilde{Z}_{0:\tau-1}, X_0)\big|\big|\mathsf{Law}(Z_\tau)\right) \tag{16}$$

In the last step, we have used the chain rule (Lemma 2).

### C.3 Controlling KL Divergence via Wasserstein Distances

We now seek to control the individual KL-divergences in the RHS of Equation (16) via Wasserstein distances. We re-state [10, Lemma 26]:

**Lemma 5.** *Suppose* $\mathbf{Z} \sim \mathcal{N}(0, \sigma^2 \mathbf{I})$. *Let* $\mathbf{A}$ *and* $\mathbf{B}$ *be random variables independent of* $\mathbf{Z}$. *Then,*

$$\mathsf{KL}\left(\mathsf{Law}(\mathbf{Z} + \mathbf{A})\big|\big|\mathsf{Law}(\mathbf{Z} + \mathbf{B})\right) \leq \frac{1}{2\sigma^2}\mathcal{W}_2^2(\mathbf{A}, \mathbf{B}).$$

The Lemma below follows from the triangle inequality for Wasserstein distance.

**Lemma 6.** *Let* $\mathbf{A}, \mathbf{B}, \mathbf{C}$ *be random vectors over* $\mathbb{R}^k$ *with finite second moments. Then,*

$$\mathcal{W}_2(\mathsf{Law}(\mathbf{A}), \mathsf{Law}(\mathbf{B} + \mathbf{C})) \leq \mathcal{W}_2(\mathsf{Law}(\mathbf{A}), \mathsf{Law}(\mathbf{B})) + \sqrt{\mathbb{E}\|\mathbf{C}\|^2}$$

From Lemma 1, we show that $\tilde{Z}_\tau = Z_\tau + B_\tau + S_\tau$. Conditioned on $\tilde{Z}_{0:\tau-1}, X_0$, $Z_\tau \sim \mathcal{N}(0, \mathbf{I})$ and is independent of $B_\tau, S_\tau$. We now use Gaussian splitting: if $A_1, A_2 \sim \mathcal{N}(0, \mathbf{I})$ i.i.d, then $\frac{A_1 + A_2}{\sqrt{2}} \sim \mathcal{N}(0, \mathbf{I})$. Therefore, letting $Z_{\tau,1}, Z_{\tau,2}$ be i.i.d from $\mathcal{N}(0, \mathbf{I})$ independent of $\tilde{Z}_{0:\tau-1}, X_0, S_\tau, B_\tau$, we can write:

$$\mathsf{KL}\left(\mathsf{Law}(\tilde{Z}_\tau|\tilde{Z}_{0:\tau-1}, X_0)\big|\big|\mathsf{Law}(Z_\tau)\right)$$

$$\leq \mathcal{W}_2^2\left(\mathsf{Law}(\tfrac{Z_{\tau,2}}{\sqrt{2}}), \mathsf{Law}(\tfrac{Z_{\tau,2}}{\sqrt{2}} + S_\tau + B_\tau|\tilde{Z}_{0:\tau-1}, X_0)\right)$$

$$= \frac{1}{2}\mathcal{W}_2^2\left(\mathsf{Law}(Z_{\tau,2}), \mathsf{Law}(Z_{\tau,2} + \sqrt{2}(S_\tau + B_\tau)|\tilde{Z}_{0:\tau-1}, X_0)\right)$$

$$\leq \mathcal{W}_2^2\left(\mathsf{Law}(Z_{\tau,2}), \mathsf{Law}(Z_{\tau,2} + \sqrt{2}S_\tau|\tilde{Z}_{0:\tau-1}, X_0)\right) + 2\mathbb{E}[\|B_\tau\|^2|\tilde{Z}_{0:\tau-1}, X_0] \tag{17}$$

In the second step, we have used the inequality in Lemma 5. In the third step we have used Lemma 6.

### C.4 Bounding the Error Along the Trajectory:

In Equation (17), we note that $Z_{\tau,2}$ is Gaussian but $S_\tau$ need not be Gaussian (but has zero mean). In the lemma stated below, we show that $Z_{\tau,2} + \sqrt{2}S_\tau|\tilde{Z}_{0:\tau-1}, X_0$ is close to a Gaussian of the same variance by adapting the arguments given in the proof of [51, Lemma 1.6]. The proof is defered to Section D

**Lemma 7.** *Suppose* $\mathbf{N}$ *is random vector such that the following conditions are satisfied:*

1. $\|\mathbf{N}\| \leq \beta$ almost surely, $\mathbb{E}\mathbf{N} = 0$ and $\mathbb{E}\mathbf{N}\mathbf{N}^{\mathsf{T}} = \Sigma$.

2. $\mathbf{N}$ takes its value in a one dimensional sub-space almost surely.

Suppose $\mathbf{Z} \sim \mathcal{N}(0, \mathbf{I})$ is independent of $\mathbf{N}$. Let $P = \mathsf{Law}(\sqrt{\mathbf{I} + \Sigma}\mathbf{Z})$ and $Q = \mathsf{Law}(\mathbf{Z} + \mathbf{N})$. Denoting $\nu := \mathsf{Tr}(\Sigma)$. Then,

$$\left(\mathcal{W}_2\left(P, Q\right)\right)^2 \leq 3(1 + \nu)\left[\beta^4\nu^3 + \beta^2\nu^2\right]\exp(\tfrac{3\beta^2}{2}) \tag{18}$$

We also have the crude bound:

$$\left(\mathcal{W}_2\left(P, Q\right)\right)^2 \leq 2\nu \tag{19}$$

The next lemma demonstrates the convexity of the Wasserstein distance, which is a straightforward consequence of the Kantorovich duality [47].

**Lemma 8** (Convexity of Wasserstein Distance). *Suppose $\mu$ is a measure over $\mathbb{R}^d$ and $Q(\cdot, \cdot)$ be a kernel over $\mathbb{R}^d$ with respect to some arbitrary measurable space $\Omega$ and let $M$ be a probability measure over $\Omega$. That is $Q(\cdot, \omega)$ is a probability distribution over $\mathbb{R}^d$ for every $\omega \in \Omega$.*

$$\mathcal{W}_2\left(\mu, \int Q(\cdot, \omega)dM(\omega)\right) \leq \int \mathcal{W}_2\left(\mu, Q(\cdot, \omega)\right)dM(\omega)$$

**Lemma 9** (Wasserstein Distance Between Gaussians, [36]).

$$\mathcal{W}_2^2(\mathcal{N}(0, \Sigma_1), \mathcal{N}(0, \Sigma_2)) = \mathsf{Tr}(\Sigma_1 + \Sigma_2 - 2(\Sigma_2^{\frac{1}{2}}\Sigma_1\Sigma_2^{\frac{1}{2}})^{\frac{1}{2}})$$

**Lemma 10.** *Let*

$$\bar{P} := \mathsf{Law}(Z_{Kt+i,2}) = \mathcal{N}(0, \mathbf{I}),$$
$$\bar{Q} := \mathsf{Law}(Z_{Kt+i,2} + \sqrt{2}(S_{Kt+i})|X_0, \tilde{Z}_{0:Kt+i-1})$$

*Define the random variable*

$$\beta = \|K\Gamma_{\frac{\alpha}{K}}^{-1}G_{\frac{\alpha}{K}}[b(\hat{\tilde{X}}_{tK+i}) - b(\tilde{X}_{tK})]\|$$

$$\mathcal{W}_2^2(\bar{P}, \bar{Q}) \leq C\mathbb{E}\left[\frac{\beta^4}{K^2} + \frac{\beta^{10}}{K^3} + \frac{\beta^6}{K^2} + \frac{\beta^{2r}}{K}\Big|\tilde{Z}_{0:Kt+i-1}, X_0\right] \tag{20}$$

*Proof.* Note that

$$\mathbb{E}[\sqrt{2}S_{tK+i}|\tilde{Z}_{0:Kt+i-1}, Y_{0:Kt+i-1}, X_0] = 0$$

Now, define the random variable $\beta$

$$\beta := \|K\Gamma_{\frac{\alpha}{K}}^{-1}G_{\frac{\alpha}{K}}[b(\hat{\tilde{X}}_{tK+i}) - b(\tilde{X}_{tK})]\| \tag{21}$$

Clearly, $\beta$ is measurable with respect to the sigma algebra of $\tilde{Z}_{0:Kt+i-1}, Z_{0:Kt+i-1}, X_0$. By the definition of $S_{tK+i}$, it is clear that $\|S_{tK+i}\| \leq \beta$ almost surely. Define the conditional covariance (only in this proof) to be:

$$\Sigma := 2\mathbb{E}[S_{tK+i}S_{tK+i}^{\mathsf{T}}|\tilde{Z}_{0:Kt+i-1}, Z_{0:Kt+i-1}X_0] \tag{22}$$

Thus, almost surely:

$$\mathsf{Tr}(\Sigma) \leq \frac{2\beta^2}{K}$$

$\sqrt{2}S_{tK+i}$ takes its values in a one-dimensional sub-space almost surely when conditioned on $\tilde{Z}_{0:Kt+i-1}, Z_{0:Kt+i-1}, X_0$. It is independent of $Z_{tK+i}$ conditioned on $\tilde{Z}_{0:Kt+i-1}, Z_{0:Kt+i-1}, X_0$.

Let us now define the following random probability distributions measurable with respect to the sigma algebra of $\tilde{Z}_{0:Kt+i-1}, X_0$

1. $Q := \mathsf{Law}(Z_{Kt+i,2} + \sqrt{2}S_{Kt+i}|\tilde{Z}_{0:Kt+i-1}, Z_{0,Kt+i-1}, X_0)$

2. $P := \mathsf{Law}(\sqrt{I+\Sigma}Z_{Kt+i,2}|\tilde{Z}_{0:Kt+i-1}, Z_{0,Kt+i-1}, X_0)$

First, note that by Lemma 8 and Jensen's inequality, we have:

$$\mathcal{W}_2^2\left(\bar{P}, \bar{Q}\right) \leq \mathbb{E}[\mathcal{W}_2^2\left(\bar{P}, Q\right)|\tilde{Z}_{0:Kt+i-1}, X_0]$$
$$\leq 2\mathbb{E}[\mathcal{W}_2^2\left(\bar{P}, P\right) + \mathcal{W}_2^2\left(P, Q\right)|\tilde{Z}_{0:Kt+i-1}, X_0] \tag{23}$$

First consider $\mathcal{W}_2^2\left(\bar{P}, P\right)$. Conditioned on $\tilde{Z}_{0:Kt+i-1}, Z_{0,Kt+i-1}, X_0, \Sigma$ has at-most one non-zero eigenvalue. Without loss of generality, we can take $\Sigma = \nu e_1 e_1^{\mathsf{T}}$ for the calculations below. From Lemma 9, we conclude that:

$$\mathcal{W}_2^2\left(\bar{P}, P\right) \leq \mathsf{Tr}(2\mathbf{I} + \Sigma - 2\sqrt{\mathbf{I}+\Sigma})$$
$$= 2 + \nu - 2\sqrt{1+\nu} \leq \frac{\nu^2}{4} \tag{24}$$

In the last step, we have used the following inequality which follows from the mean-value theorem: $\sqrt{1+\nu} \geq 1 + \frac{\nu}{2} - \frac{\nu^2}{8}$. We check that the conditions for Lemma 7 hold for $P, Q$ (almost surely conditioned on $\tilde{Z}_{0:Kt+i-1}, X_0$) and $\nu \leq \frac{\beta^2}{K}$ to conclude:

$$\mathcal{W}_2^2\left(P, Q\right) \leq C\left[\frac{\beta^{10}}{K^3} + \frac{\beta^6}{K^2}\right]\exp(\frac{3\beta^2}{2})\mathbb{1}(\beta \leq 1) + \frac{2\beta^2}{K}\mathbb{1}(\beta > 1)$$
$$\leq C\left[\left[\frac{\beta^{10}}{K^3} + \frac{\beta^6}{K^2}\right]\mathbb{1}(\beta \leq 1) + \frac{\beta^2}{K}\mathbb{1}(\beta > 1)\right] \tag{25}$$

Combining this with Equation (24), we conclude that for any $r \geq 1$:

$$\mathcal{W}_2^2(\bar{P}, \bar{Q}) \leq C\left[\frac{\beta^4}{K^2} + \left[\frac{\beta^{10}}{K^3} + \frac{\beta^6}{K^2}\right]\mathbb{1}(\beta \leq 1) + \frac{\beta^2}{K}\mathbb{1}(\beta > 1)\right]$$
$$\leq C\left[\frac{\beta^4}{K^2} + \frac{\beta^{10}}{K^3} + \frac{\beta^6}{K^2} + \frac{\beta^2}{K}\mathbb{1}(\beta > 1)\right]$$
$$\leq C\left[\frac{\beta^4}{K^2} + \frac{\beta^{10}}{K^3} + \frac{\beta^6}{K^2} + \frac{\beta^{2r}}{K}\right] \tag{26}$$

$\square$

### C.5  Finishing The Proof

We combine Equations (16) and (17) with Lemma 10 to conclude the result of Theorem 1.

## D  Proof of Lemma 7

The crude bound in Equation (19) follows via a naive coupling argument. We will now sketch a proof Equation (18) in Lemma 7 by showcasing how to modify the proof of [52, Lemma 1.6]. Our proof deals with a specialized case compared to [51, Lemma 1.6]. This allows us to derive a stronger result. In this section, by 'original proof', we refer to the proof in [52].

Since $\mathbf{N}$ is supported on a single dimensional sub-space almost surely, we take this direction to be $e_1$ almost surely without loss of generality. We thus take the covariance matrix of $\mathbf{N}$ to be $\Sigma = \nu e_1 e_1^{\mathsf{T}}$.

We generate jointly distributed random vectors $\mathbf{Z}, \mathbf{Z}' \in \mathbb{R}^d$ as follows: Let $\langle\mathbf{Z}, e_j\rangle$ are i.i.d. standard normal random variables for $j = 2, \ldots, d$ and $\langle\mathbf{Z}', e_j\rangle = \langle\mathbf{Z}, e_j\rangle$ almost surely. We generate $\langle\mathbf{Z}', e_1\rangle$ to be standard normal independent of $(\langle\mathbf{Z}, e_j\rangle)_{j\geq 2}$. Let $\langle\mathbf{N}, e_1\rangle$ be independent of $\mathbf{Z}'$. We draw $\langle\mathbf{Z}, e_1\rangle$ to be standard normally distributed and Wasserstein-2 optimally coupled to $\frac{\langle\mathbf{Z}', e_1\rangle + \langle\mathbf{N}, e_1\rangle}{\sqrt{1+\nu}}$ and independent of all other random variables mentioned above. We can easily check that $(\sqrt{\mathbf{I}+\Sigma}\mathbf{Z}, \mathbf{Z}' + \mathbf{N})$ as defined above is a coupling between $P, Q$.

Via this coupling, we conclude that:

$$\mathcal{W}_2\left(P, Q\right) \le \sqrt{\mathbb{E}(\sqrt{1+\nu}\langle \mathbf{Z}, e_1\rangle - \langle \mathbf{Z}', e_1\rangle - \langle \mathbf{N}, e_1\rangle)^2}$$

$$= \sqrt{1+\nu}\mathcal{W}_2\left(\langle \mathbf{Z}, e_1\rangle, \frac{\langle \mathbf{Z}', e_1\rangle + \langle \mathbf{N}, e_1\rangle}{\sqrt{1+\nu}}\right) \tag{27}$$

Let $Z_1 := \langle \mathbf{Z}, e_1\rangle$ and $Z_1' := \langle \mathbf{Z}', e_1\rangle$, $N_1 := \langle \mathbf{N}, e_1\rangle$. We define $m = 1+\frac{1}{\nu}$ (and note throughout the proof that a value of infinity can be easily handled). We see that $\frac{Z_1'+N_1}{\sqrt{1+\nu}} = \sqrt{1 - \frac{1}{m}}Z_1' + \sqrt{1 - \frac{1}{m}}N_1$

Note that $\sqrt{1 - \frac{1}{m}}\mathbf{N}$ is denoted by $Y$ in the original proof and the bound is $\|Y\| \le \frac{\beta}{\sqrt{n}}$ instead of $\|\mathbf{N}\| \le \beta$ in this proof.

Consider the function $f(x)$, which denotes the ratio of density function of $\frac{Z_1'+N_1}{\sqrt{1+\nu}}$ to that of $Z_1$ at the point $x$.

$$\mathbb{E}f(Z_1)^2 = \mathbb{E}\left[\exp\left(\frac{-(N_1^2 + (N_1')^2) + 2mN_1N_1'}{2(m+1)} + \frac{1}{2(m^2-1)} - r(m)\right)\right] \tag{28}$$

Where $N_1'$ is an i.i.d copy of $N_1$ and $r(n) := \frac{1}{2(n^2-1)} - \frac{1}{2}\log(1 + \frac{1}{n^2-1})$.

Define $Q_1 = \frac{-(N_1^2 + (N_1')^2) + 2mN_1N_1'}{2(m+1)} + \frac{1}{2(m^2-1)} - r(m)$. We modify the estimates in Lemma 4.4 and Lemma 4.5 in the original proof in the following:

**Lemma 11.** *1.* $|Q_1| \le \frac{m|N_1N_1'|}{m+1} + \frac{\beta^2}{m+1} + \frac{1}{2(m^2-1)}$ *almost surely*

*2.* $\mathbb{E}[Q_1] = -\frac{1}{2(m^2-1)} - r(m)$

*3.* $\mathbb{E}[Q_1^2] \le \frac{m\beta^2 + 2m^2 + 1}{2(m^2-1)^2}$

*Proof.* 1. Note that $r(m) \le \frac{1}{2(m^2-1)}$. By triangle inequality, we have:

$$|Q_1| \le \frac{m|N_1N_1'|}{m+1} + \frac{\beta^2}{m+1} + \frac{1}{2(m^2-1)} \tag{29}$$

2. A direct calculation shows the identity for $\mathbb{E}Q_1$.

3. Now, consider $\mathbb{E}Q_1^2$. We follow the proof of [52, Lemma 4.5], to conclude the following inequalities. Since $\mathbb{E}Q_1 \le 0$ and $\frac{1}{2(m^2-1)} - r(m) \ge 0$, we have:

$$\mathbb{E}Q_1^2 \le \mathbb{E}(Q_1 - \frac{1}{2(m^2-1)} + r(m))^2$$

$$= \frac{m^2}{(m+1)^2}\mathbb{E}N_1^2(N_1')^2 + \frac{1}{4(m+1)^2}\mathbb{E}(N_1^2 + (N_1')^2)^2$$

$$= \frac{m^2}{(m^2-1)^2} + \frac{1}{4(m+1)^2}\mathbb{E}(N_1^2 + (N_1')^2)^2$$

$$\le \frac{m\beta^2 + 2m^2 + 1}{2(m^2-1)^2} \tag{30}$$

$\square$

Note that the parameter $\sigma_i$ found in the original proof satisfies $\sigma_i = 1$ in our proof. We now proceed with the proof of Lemma 7. Define $R(Q) = \exp(Q) - 1 - Q - \frac{Q^2}{2}$. From the original proof, we conclude via the Talagrand transport inequality that:

$$\left[\mathcal{W}_2\left(\langle\mathbf{Z},e_1\rangle,\frac{\langle\mathbf{Z}',e_1\rangle+\langle\mathbf{N},e_1\rangle}{\sqrt{1+\nu}}\right)\right]^2 \leq 2[\mathbb{E}e^{Q_1}-1] = 2\left[\mathbb{E}[Q_1]+\frac{1}{2}\mathbb{E}[Q_1^2]+\mathbb{E}[R(Q_1)]\right] \quad (31)$$

From Lemma 11, and using the fact that $r(m) \geq 0$, we note that:

$$\mathbb{E}[Q_1]+\frac{1}{2}\mathbb{E}[Q_1^2] \leq \frac{3+m\beta^2}{4(m^2-1)^2}$$

Now, let us bound $\mathbb{E}R(Q_1)$. Via a straightforward application of the taylor series, we have almost surely:

$$R(Q_1) \leq \frac{|Q_1|^3\exp(|Q_1|)}{6} \quad (32)$$

From Lemma 11, we conclude that $|Q_1| \leq \beta^2 + \frac{1}{2(m^2-1)}$. Since $\nu \leq \beta^2$, we must have $m = 1+\frac{1}{\nu} \geq 1+\frac{1}{\beta^2}$. Thus, we have $|Q_1| \leq \frac{3\beta^2}{2}$ almost surely. Using this in Equation (32), we have:

$$\begin{aligned}
\mathbb{E}R(Q_1) &\leq \mathbb{E}\frac{|Q_1|^3}{6}\exp\left(\frac{3\beta^2}{2}\right) \\
&\leq \frac{\beta^2}{4}\exp(\frac{3\beta^2}{2})\mathbb{E}|Q_1|^2 \\
&\leq \left[\frac{m\beta^4+m^2\beta^2}{4(m^2-1)^2}\right]\exp(\frac{3\beta^2}{2}) \quad (33)
\end{aligned}$$

In the last step, we have used item 3 of Lemma 11 along with the fact that $m\beta^2 \geq 1$.

Combining these estimates with Equation (31), we conclude:

$$\left[\mathcal{W}_2\left(\langle\mathbf{Z},e_1\rangle,\frac{\langle\mathbf{Z}',e_1\rangle+\langle\mathbf{N},e_1\rangle}{\sqrt{1+\nu}}\right)\right]^2 \leq \left[\frac{m\beta^4+m^2\beta^2}{2(m^2-1)^2}\right]\exp(\frac{3\beta^2}{2}) + \frac{3+m\beta^2}{2(m^2-1)^2}$$

Now, we use the fact that $m = 1+\frac{1}{\nu}$, which implies $\frac{1}{m^2-1} \leq \frac{1}{(m-1)^2} \leq \nu^2$. Thus, we have:

$$\left[\mathcal{W}_2\left(\langle\mathbf{Z},e_1\rangle,\frac{\langle\mathbf{Z}',e_1\rangle+\langle\mathbf{N},e_1\rangle}{\sqrt{1+\nu}}\right)\right]^2 \leq \left[\frac{\beta^4\nu^3}{2}+\frac{\beta^2\nu^2}{2}\right]\exp(\frac{3\beta^2}{2}) + \frac{3\nu^4+\beta^2\nu^3}{2} \quad (34)$$

Using the fact that $\beta^2 \geq \nu$, we have: $\frac{3\nu^4+\beta^2\nu^3}{2} \leq 2\beta^2\nu^3$. Thus, $2(1+\nu)\beta^2\nu^3 \leq 2\beta^2\nu^3 + 2\beta^4\nu^3 \leq 2(1+\nu)(\beta^2\nu^2+\beta^4\nu^3)$. Plugging this into Equation (27), we conclude the result.

## E   Overdamped Langevin Dynamics

In this section, we will prove Theorem 2 after developing some key results regarding OLMC. Recall that in this case $b(\mathbf{x},\tau) = -\nabla F(\mathbf{x})$ for some $F : \mathbb{R}^d \to \mathbb{R}$, $G_{\frac{\alpha}{K}} = \frac{\alpha}{K}\mathbf{I}$ and $\Gamma_{\frac{\alpha}{K}} = \sqrt{\frac{2\alpha}{K}}\mathbf{I}$. Let $N_t := \sum_{j=0}^{K-1}H_{t,j}$. Throughout this section, we assume that $\nabla F$ is $L$-Lipschitz. We will assume that $T$ is a power of 2 in this entire section, which useful to apply Lemma 26. The results hold for any $T$ by considering the closest power of 2 above $T$ instead.

Recall $B_{tK+i}$ from Theorem 1. Instantiating this for Overdamped Langevin Dynamics, we conclude that for any $t, i \in \mathbb{N} \cup \{0\}$ and $0 \leq i \leq K-1$, we must have:

$$\|B_{tK+i}\|^2 \leq \frac{L^2\alpha}{2K}\sup_{0\leq j\leq K-1}\|\hat{\tilde{X}}_{tK+j}-\tilde{X}_{tK+j}\|^2 \quad (35)$$

Here, we have used the fact that $\nabla F$ is $L$-Lipschitz.

Now, consider $\beta_{tK+i}$ in Theorem 1. For any $p \geq 1$, $t, i \in \mathbb{N} \cup \{0\}$ and $0 \leq i \leq K-1$, we have:

$$\beta_{tK+i}^{2p} \leq (\frac{L^2 \alpha K}{2})^p \sup_{0 \leq j \leq K-1} \|\hat{\tilde{X}}_{tK+j} - \tilde{X}_{tK}\|^{2p} \tag{36}$$

In order to apply Theorem 1, we will now proceed to bound the quantities in the RHS of Equations (35) and (36)

### E.1 Bounding the Moments

The following lemma gives us almost sure control over quantities of interest. We refer to Section I.1 for the proof.

**Lemma 12.** *Suppose the stepsize $\alpha L < 1$. Let $M_{t,k} := \sqrt{\frac{2\alpha}{K}} \sup_{0 \leq i \leq k-1} \| \sum_{j=0}^{i} Z_{tK+j} \|$. Then the following hold almost surely for every $0 \leq k \leq K$*

1.

$$\sup_{0 \leq i \leq k-1} \|\hat{\tilde{X}}_{tK+i} - \tilde{X}_{tK}\| \leq \alpha \|\nabla F(\tilde{X}_{tK})\| + M_{t,k}$$

2.

$$\sup_{0 \leq i \leq k} \|\hat{\tilde{X}}_{tK+i} - \tilde{X}_{tK+i}\| \leq \alpha L N_t \sup_{i \leq k-1} \|\hat{\tilde{X}}_{tK+i} - \tilde{X}_{tK}\|$$

3.

$$\mathbb{E}[M_{t,K}^p] \leq C(p)(\alpha d)^{\frac{p}{2}}$$

We will now prove the following growth estimate for the trajectory $\tilde{X}_{tK}$. We refer to Section I.2 for its proof.

**Lemma 13.** *Let $p \geq 1$ be fixed. There exists a large enough constant $\bar{C}_p$ which depends only on $p$ such that whenever $s - t \leq \frac{1}{\alpha L \bar{C}_p}$, we have:*

$$\sup_{t \leq h \leq s} \left[ \mathbb{E}\|\tilde{X}_{hK} - \tilde{X}_{tK}\|^p | \tilde{X}_{tK} \right]^{\frac{1}{p}} \leq 3\alpha(s-t)\|\nabla F(\tilde{X}_{tK})\| + C_p \sqrt{\alpha d(s-t)}$$

We apply Lemma 26 along with Lemma 13 to conclude the following result which is proved in Section I.4.

**Lemma 14.** *There exists a constant $c_p$ such that whenever $\alpha L \leq c_p$, we have:*

$$\sum_{t=1}^{T} \mathbb{E}\|\nabla F(\tilde{X}_{tK})\|^{2p} \leq C_p L^p d^p T + C_p (\alpha L)^{p-1} \mathbb{E}(\sum_{t=1}^{T} \|\nabla F(\tilde{X}_{tK})\|^2)^p \tag{37}$$

While Lemma 14 controlled $\sum_{t=1}^{T} \mathbb{E}\|\nabla F(\tilde{X}_{tK})\|^{2p}$ in terms of $\sum_{t=1}^{T} \|\nabla F(\tilde{X}_{tK})\|^2$, in the Lemma below we control $\sum_{t=1}^{T} \|\nabla F(\tilde{X}_{tK})\|^2$. We refer to Section I.5 for its proof.

**Lemma 15.** *Suppose $p \geq 1$ be arbitrary. There exists $c_p > 0$ small enough such that whenever $\alpha L < c_p$, we have for some constant $C_p > 0$ depending only on $p$:*

$$\mathbb{E}(\sum_{t=0}^{T-1} \|\nabla F(\tilde{X}_{tK})\|^2)^p \leq \frac{C_p}{\alpha^p} \mathbb{E}|(F(X_0) - F(\tilde{X}_{KT}))^+|^p + C_p T^{p-1} L^{2p} \alpha^{2p} \mathbb{E} \sum_{t=0}^{T-1} \|\nabla F(\tilde{X}_{tK})\|^{2p}$$

$$+ C_p L^p d^p T^p + \frac{C_p}{\alpha^p} \tag{38}$$

We combine the results of Lemma 14 and Lemma 15 to conclude the following result:

**Lemma 16.** *Given $p \geq 1$ arbitrary, there exists a constant $c_p > 0$ depending only $p$ such that whenever $\alpha L < c_p$ and $\alpha^{3p-1} L^{3p-1} T^{p-1} < c_p$, we must have:*

$$\sum_{t=1}^{T} \mathbb{E}\|\nabla F(\tilde{X}_{tK})\|^{2p} \leq C_p L^p d^p T (1 + (\alpha L T)^{p-1}) + \frac{C_p L^{p-1}}{\alpha}\left[\mathbb{E}|(F(X_0) - F(\tilde{X}_{KT}))^+|^p + 1\right]$$

**Lemma 17.** *1.*

$$\mathbb{E}\|B_{tK+i}\|^2 \leq \frac{CL^4\alpha^5}{K}\mathbb{E}\|\nabla F(\tilde{X}_{tK})\|^2 + \frac{CL^4\alpha^4}{K}d$$

*2.*

$$\mathbb{E}\beta_{tK+i}^{2p} \leq C_p\left[L^{2p}\alpha^{3p}K^p\mathbb{E}\|\nabla F(\tilde{X}_{tK})\|^{2p} + L^{2p}\alpha^{2p}K^p d^p\right]$$

*Proof.* 1. Using Equation (35), we have:

$$
\begin{aligned}
\mathbb{E}\|B_{tK+i}\|^2 &\leq \frac{L^2\alpha}{2K}\mathbb{E}\sup_{0\leq j\leq K-1}\|\hat{\tilde{X}}_{tK+j} - \tilde{X}_{tK+j}\|^2 \\
&\leq \frac{L^4\alpha^3}{2K}\mathbb{E}N_t^2\sup_{0\leq j\leq K-1}\|\hat{\tilde{X}}_{tK+j} - \tilde{X}_{tK}\|^2 \\
&\leq \frac{L^4\alpha^3}{K}\mathbb{E}\sup_{0\leq j\leq K-1}\|\hat{\tilde{X}}_{tK+j} - \tilde{X}_{tK}\|^2 \\
&\leq \frac{CL^4\alpha^3}{K}\left[\alpha^2\mathbb{E}\|\nabla F(\tilde{X}_{tK})\|^2 + \alpha d\right] \qquad (39)
\end{aligned}
$$

2. We use Equation (36) and proceed as in item 1.

$\square$

### E.2 Finishing The Proof

For $p \geq 1$, we define $\Delta^{(p)} := \mathbb{E}[(F(X_0) - F(\mathbf{x}^*))^p]$. Let $\lesssim$ denote $\leq$ up to a universal positive multiplicative constant on the RHS. We now combine Lemma 17, Lemma 15 with Theorem 1 (taking $r = 7$) to conclude the following bound:

$$
\begin{aligned}
&\mathsf{KL}\left(\mathsf{Law}((X_t^\mathsf{P})_{0\leq t\leq T})\big\|\mathsf{Law}((X_{Kt})_{0\leq t\leq T})\right) \\
&\lesssim L^4\alpha^4(\Delta^{(1)} + 1) + L^5\alpha^5 K(\Delta^{(2)} + 1) + L^8\alpha^8 K^2(\Delta^{(3)} + 1) + L^{14}\alpha^{14}K^3(\Delta^{(5)} + 1) \\
&\quad + L^{20}\alpha^{20}K^7(\Delta^{(7)} + 1) + L^4\alpha^4 K d^2 T + L^7\alpha^7 K d^2 T^2 + L^6\alpha^6 K^2 d^3 T + L^{11}\alpha^{11}K^2 d^3 T^3 \\
&\quad + L^{10}\alpha^{10}K^3 d^5 T + L^{19}\alpha^{19}K^3 d^5 T^5 + L^{14}\alpha^{14}K^7 d^7 T + L^{27}\alpha^{27}K^7 d^7 T^7 \qquad (40)
\end{aligned}
$$

## F  Underdamped Langevin Dynamics

In this section, we will prove Theorem 3 after developing some key results regarding ULMC. We will assume that $T$ is a power of 2 in this entire section, which useful to apply Lemma 26. The results hold for any $T$ by considering the closest power of 2 above $T$ instead. Recall that in the case of ULMC, we have:

$$A_h := \begin{bmatrix} \mathbf{I}_d & \frac{1}{\gamma}(1 - e^{-\gamma h})\mathbf{I}_d \\ 0 & e^{-\gamma h}\mathbf{I}_d \end{bmatrix}, \quad G_h := \begin{bmatrix} \frac{1}{\gamma}(h - \frac{1}{\gamma}(1 - e^{-\gamma h}))\mathbf{I}_d & 0 \\ \frac{1}{\gamma}(1 - e^{-\gamma h})\mathbf{I}_d & 0 \end{bmatrix} \quad b(X_t) := \begin{bmatrix} -\nabla F(U_t) \\ 0 \end{bmatrix}$$

$$\Gamma_h^2 := \begin{bmatrix} \frac{2}{\gamma}\left(h - \frac{2}{\gamma}(1 - e^{-\gamma h}) + \frac{1}{2\gamma}(1 - e^{-2\gamma h})\right)\mathbf{I}_d & \frac{1}{\gamma}(1 - 2e^{-\gamma h} + e^{-2\gamma h})\mathbf{I}_d \\ \frac{1}{\gamma}(1 - 2e^{-\gamma h} + e^{-2\gamma h})\mathbf{I}_d & (1 - e^{-2\gamma h})\mathbf{I}_d \end{bmatrix}$$

Throughout this section, we assume that $\nabla F(\cdot)$ is $L$-Lipschitz. We let $N_t := \sum_{i=0}^{K-1} H_{t,i}$. We let $\Pi$ be the projector to the first $d$-dimensions in the standard basis. That is, $\Pi := \begin{bmatrix} \mathbf{I}_d & 0 \\ 0 & 0 \end{bmatrix}$.

The $2d$ dimensional space can be decomposed into position (sub-space spanned by the first $d$ standard basis vectors) and velocity (sub-space spanned by the last $d$ standard basis vectors) sub-spaces. We use the convention that $X \in \mathbb{R}^{2d}$ is such that $X = \begin{bmatrix} U \\ V \end{bmatrix}$ where $U \in \mathbb{R}^d$ is the position and $V \in \mathbb{R}^d$ is the velocity. Throughout this section, we implicity let $\tilde{X}_\tau = \begin{bmatrix} \tilde{U}_\tau \\ \tilde{V}_\tau \end{bmatrix}$. With some abuse of notation, we let $U = \Pi X$ and $V = (\mathbf{I} - \Pi)X$.

The following lemma collects some useful bounds on $A_h, G_h$ and $\Gamma_h$, and is proved in Section I.6

**Lemma 18.**

$$G_h^\mathsf{T} \Gamma_h^{-2} G_h \preceq \begin{bmatrix} C \frac{h \exp(2\gamma h)}{\gamma} \mathbf{I}_d & 0 \\ 0 & 0 \end{bmatrix}$$

*For all $0 \le j \le K - 1$,*

$$\|\Pi A_{\frac{j\alpha}{K}} G_{\frac{\alpha}{K}}\| \le \frac{3\alpha^2}{2K} \exp(\tfrac{\gamma\alpha}{K})$$

Analogous to the analysis of Overdamped Langevin Dynamics, we consider the following dynamics.

### F.1 Bounding the Moments:

Using the scaling relations in Section 2.1, we write down the iteration of $\mathsf{PS}(A, G, \Gamma, b, \alpha, K)$ for Underdamped Langevin Dynamics as:

$$\tilde{X}_{(t+1)K} = A_\alpha \tilde{X}_{tK} + G_\alpha b(\tilde{X}_{tK}) + \alpha \Delta_t + \Gamma_\alpha \bar{Y}_t \tag{41}$$

Where, $\quad \alpha \Delta_t \quad := \quad \sum_{i=0}^{K-1} K H_i A_{\frac{\alpha(K-i-1)}{K}} G_{\frac{\alpha}{K}} \left[ b(\hat{\tilde{X}}_{tK+i}) - b(\tilde{X}_{tK}) \right] \quad$ and $\quad \Gamma_\alpha \bar{Y}_t \quad := \sum_{j=0}^{K-1} A_{\frac{\alpha(K-1-j)}{K}} \Gamma_{\frac{\alpha}{K}} Z_{tK+j}$.

Applying the triangle inequality, we conclude the following lemma.

**Lemma 19.** *Suppose that $\|\cdot\|_{\mathsf{any}}$ is any semi-norm over $\mathbb{R}^d$. Then the following hold almost surely for every $0 \le k \le K$*

1.

$$\|\hat{\tilde{X}}_{tK+i} - \tilde{X}_{tK}\|_{\mathsf{any}} \le \|(A_{\frac{i\alpha}{K}} - \mathbf{I})\tilde{X}_{tK}\|_{\mathsf{any}} + \|G_{\frac{i\alpha}{K}} b(\tilde{X}_{tK})\|_{\mathsf{any}}$$

$$+ \|\sum_{j=0}^{i-1} A_{\frac{\alpha(i-1-j)}{K}} \Gamma_{\frac{\alpha}{K}} Z_{tK+j}\|_{\mathsf{any}}$$

2.

$$\|\hat{\tilde{X}}_{tK+i} - \tilde{X}_{tK+i}\|_{\mathsf{any}} \le \sum_{j=0}^{i-1} K H_j \|A_{\frac{(i-1-j)\alpha}{K}} G_{\frac{\alpha}{K}} (b(\hat{\tilde{X}}_{tK+i}) - b(\tilde{X}_{tK}))\|_{\mathsf{any}}$$

Define $\psi_t := \tilde{U}_{Kt} + \frac{\tilde{V}_{Kt}}{\gamma}$. The following Lemma, proved in Section I.7, gives the time evolution of $\psi_t$.

**Lemma 20.**

$$\psi_{t+1} - \psi_t = -\frac{\alpha}{\gamma} \nabla F(\tilde{U}_{tK}) - \sum_{i=0}^{K-1} \frac{H_i \alpha}{\gamma} \left[ \nabla F(\hat{\tilde{U}}_{tK+i}) - \nabla F(\tilde{U}_{tK}) \right] + \tilde{\Psi}_t$$

*Where $\tilde{\Psi}_t \sim \mathcal{N}(0, \frac{2\alpha}{\gamma} \mathbf{I}_d)$ is independent of $\tilde{X}_{tK}$ but not necessarily indepependent of $\hat{\tilde{U}}_{tK+i}$ for $i > 0$.*

**Lemma 21.** *Consider the seminorm given by* $\|x\|_\Pi^2 = x^\intercal \Pi x$. *Let* $M_{t,K} := \sup_{j \le K-1} \|\sum_{j=0}^{i-1} A_{\frac{\alpha(i-1-j)}{K}} \Gamma_{\frac{\alpha}{K}} Z_{tK+j}\|_\Pi$. *For any* $0 \le i \le K-1$.

1.

$$\|\hat{\tilde{U}}_{tK+i} - \tilde{U}_{tK}\|^2 \le C\alpha^2 \|\tilde{V}_{tK}\|^2 + C\alpha^4 \|\nabla F(\tilde{U}_{tK})\|^2 + CM_{t,K}^2 \tag{42}$$

2.

$$F(\psi_{t+1}) - F(\psi_t) \le -\frac{\alpha}{4\gamma}(1 - \frac{6\alpha L}{\gamma})\|\nabla F(\tilde{U}_{tK})\|^2 + \langle \nabla F(\psi_t) - \nabla F(\tilde{U}_{tK}), \tilde{\Psi}_t \rangle$$

$$+ \langle \nabla F(U_t), \tilde{\Psi}_t \rangle + \frac{3L}{2}\|\tilde{\Psi}_t\|^2 + \frac{3\alpha L^2 \|\tilde{V}_{tK}\|^2}{2\gamma^3}$$

$$+ \frac{N_t^2 \alpha L^2}{\gamma}\left(1 + \frac{3\alpha L}{2\gamma}\right) \sup_{0 \le i \le K-1} \|\hat{\tilde{U}}_{tK+i} - \tilde{U}_{tK}\|^2 \tag{43}$$

3. *For any* $p \ge 1$, *we have:*

$$\mathbb{E}M_{t,K}^p \le C\exp(\frac{p\gamma\alpha}{2})\gamma^{\frac{p}{2}}\alpha^{\frac{3p}{2}}(d + \log K)^{\frac{p}{2}}$$

We refer to Section I.8 for the proof. Analogous to Section E, for any $p \ge 1$, we define $\mathcal{S}_{2p}(V) := \sum_{t=0}^{T-1}\|\tilde{V}_{tK}\|^{2p}$, $\mathcal{S}_{2p}(\nabla F) := \sum_{t=0}^{T-1}\|\nabla F(\tilde{U}_{tK})\|^{2p}$. The following lemma, proved in Section I.9, bounds the moments of these quantities.

**Lemma 22.** *There exists a constant $c_0$ such that whenever $\gamma\alpha < 1$, $\frac{\alpha L}{\gamma} < c_0$. Then the following relationships hold:*

1.

$$[\mathbb{E}(\mathcal{S}_2(V))^p]^{\frac{1}{p}} \le \frac{C}{\gamma\alpha}[\mathbb{E}\|\tilde{V}_0\|^{2p}]^{\frac{1}{p}} + \frac{6}{\gamma^2}[\mathbb{E}(\mathcal{S}_2(\nabla F))^p]^{\frac{1}{p}} + C_p(T(d + \log K) + \frac{1}{\gamma\alpha})$$

$$+ \frac{L^2 T^{1-\frac{1}{p}}\alpha^2}{\gamma^2}[\mathbb{E}\mathcal{S}_{2p}(V)]^{\frac{1}{p}} + \frac{L^2 T^{1-\frac{1}{p}}\alpha^4}{\gamma^2}[\mathbb{E}\mathcal{S}_{2p}(\nabla F)]^{\frac{1}{p}} \tag{44}$$

2.

$$[\mathbb{E}(\mathcal{S}_2(\nabla F))^p]^{\frac{1}{p}} \le \frac{\gamma}{\alpha}[\mathbb{E}|(F(\Psi_0) - F(\Psi_T))^+|^p]^{\frac{1}{p}}$$

$$+ \frac{CL^2}{\gamma^2}[\mathbb{E}(\mathcal{S}_2(V))^p]^{\frac{1}{p}} + C_p\left[L(d + \log K)T + \frac{\gamma}{\alpha}\right]$$

$$+ C_p T^{1-\frac{1}{p}}L^2\alpha^2[\mathbb{E}\mathcal{S}_{2p}(V)]^{\frac{1}{p}} + C_p T^{1-\frac{1}{p}}L^2\alpha^4[\mathbb{E}\mathcal{S}_{2p}(\nabla F)]^{\frac{1}{p}} \tag{45}$$

*Suppose additionally that $\gamma \ge C_0\sqrt{L}$ for some large enough universal constant $C_0$. Then, the bounds above imply the following inequalities:*

1.

$$[\mathbb{E}(\mathcal{S}_2(V))^p]^{\frac{1}{p}} \le \frac{C}{\gamma\alpha}[\mathbb{E}\|\tilde{V}_0\|^{2p}]^{\frac{1}{p}} + \frac{C}{\gamma\alpha}[\mathbb{E}|(F(\Psi_0) - F(\Psi_T))^+|^p]^{\frac{1}{p}}$$

$$+ C_p(T(d + \log K) + \frac{1}{\gamma\alpha}) + \frac{L^2 T^{1-\frac{1}{p}}\alpha^2}{\gamma^2}[\mathbb{E}\mathcal{S}_{2p}(V)]^{\frac{1}{p}}$$

$$+ \frac{L^2 T^{1-\frac{1}{p}}\alpha^4}{\gamma^2}[\mathbb{E}\mathcal{S}_{2p}(\nabla F)]^{\frac{1}{p}} \tag{46}$$

*2.*

$$[\mathbb{E}(\mathcal{S}_2(\nabla F))^p]^{\frac{1}{p}} \le \frac{C\gamma}{\alpha}[\mathbb{E}\|\tilde{V}_0\|^{2p}]^{\frac{1}{p}} + \frac{C\gamma}{\alpha}[\mathbb{E}|(F(\Psi_0) - F(\Psi_T))^+|^p]^{\frac{1}{p}}$$
$$+ C_p\left[L(d + \log K)T + \frac{\gamma}{\alpha}\right]$$
$$+ C_p T^{1-\frac{1}{p}}L^2\alpha^2[\mathbb{E}\mathcal{S}_{2p}(V)]^{\frac{1}{p}} + C_p T^{1-\frac{1}{p}}L^2\alpha^4[\mathbb{E}\mathcal{S}_{2p}(\nabla F)]^{\frac{1}{p}} \quad (47)$$

The following lemma gives a growth bound for Overdamped Langevin dynamics. We give the proof in Section I.10.

**Lemma 23.** *Let $s, t \in \mathbb{N} \cup \{0\}$ such that $s > t$. There exists a constant $c_p > 0$ such that whenever $\frac{\alpha L(s-t)}{\gamma} \le c_p$, $\alpha\gamma(s - t) \le c_p$ and $\alpha^2 L(s - t) \le c_p$, the following statements hold:*

$$\sup_{t \le h \le s} \left[\mathbb{E}\|\tilde{U}_{hk} - \tilde{U}_{tK}\|^p\right]^{\frac{1}{p}} \le 8\alpha(s - t)\left[\mathbb{E}\|\tilde{V}_{tK}\|^p\right]^{\frac{1}{p}} + \frac{8\alpha(s - t)}{\gamma}\mathbb{E}\left[\|\nabla F(\tilde{U}_{tK})\|^p\right]^{\frac{1}{p}}$$
$$+ C_p\sqrt{\frac{d + \log K}{L}} \quad (48)$$

$$\sup_{t \le h \le s} \left[\mathbb{E}\|\tilde{V}_{hk} - \tilde{V}_{tK}\|^p\right]^{\frac{1}{p}} \le 8\gamma\alpha(s - t)\left[\mathbb{E}\|\tilde{V}_{tK}\|^p\right]^{\frac{1}{p}} + 8\alpha(s - t)\mathbb{E}\left[\|\nabla F(\tilde{U}_{tK})\|^p\right]^{\frac{1}{p}}$$
$$+ C_p\gamma\sqrt{\frac{d + \log K}{L}} \quad (49)$$

We combine Lemma 23 with Lemma 26 to conclude:

**Lemma 24.** *Let $p \ge 1$ be given. There exists $c_p > 0$ such that for any $N$ which satisfies the following conditions:*

1. *$N$ is an integer power of $2$ and $N \le T$.*
2. *$\frac{\alpha LN}{\gamma} \le c_p$, $\alpha\gamma N \le c_p$ and $\alpha^2 LN \le c_p$*

*The following satements hold:*

1.

$$\mathbb{E}\mathcal{S}_{2p}(\nabla F) \le C_p(L\alpha N)^{2p}\mathbb{E}\mathcal{S}_{2p}(V) + C_p TL^p (d + \log K)^p + \frac{2^p}{N^{p-1}}\mathbb{E}(\mathcal{S}_2(\nabla F))^p \quad (50)$$

2.

$$\mathbb{E}\mathcal{S}_{2p}(V) \le C_p(\alpha N)^{2p}\mathbb{E}\mathcal{S}_{2p}(\nabla F) + T\left[\frac{\gamma^2}{L}(d + \log K)\right]^p + \frac{2^p}{N^{p-1}}\mathbb{E}(\mathcal{S}_2(V))^p \quad (51)$$

*Proof.* We consider $a_t := \nabla F(\tilde{U}_{(t-1)K})$, $b_t := \tilde{V}_{(t-1)K}$. Let $N$ satisfy the condition in the Lemma. Since $T$ is a power of $2$, clearly, $N$ divides $T$. Let $\mathcal{T}_k = \{(k-1)N + 1, \ldots, kN\}$ be as defined in Lemma 26. Consider the following upper bound derived using the results of Lemma 23

$$\sum_{k=1}^{T/N} \sum_{\substack{j,j' \in \mathcal{T}_k \\ j' > j}} \mathbb{E}\|a_j - a'_j\|^{2p} \le C_p L^{2p}\alpha^{2p}N^{2p+1}\mathbb{E}\mathcal{S}_{2p}(V) + C_p\frac{L^{2p}\alpha^{2p}N^{2p+1}}{\gamma^{2p}}\mathbb{E}\mathcal{S}_{2p}(\nabla F)$$
$$+ C_p NTL^p (d + \log K)^p \quad (52)$$

Applying Lemma 26, we conclude:

$$\mathbb{E}\mathcal{S}_{2p}(\nabla F) \leq C_p (L\alpha N)^{2p}\mathbb{E}\mathcal{S}_{2p}(V) + C_p T L^p (d + \log K)^p + \frac{2^p}{N^{p-1}}\mathbb{E}(\mathcal{S}_2(\nabla F))^p \qquad (53)$$

Similarly, we have:

$$\sum_{k=1}^{T/N} \sum_{\substack{j,j' \in \mathcal{T}_k \\ j' > j}} \mathbb{E}\|b_j - b_j'\|^2 \leq C_p \gamma^{2p}\alpha^{2p}N^{2p+1}\mathbb{E}\mathcal{S}_{2p}(V) + C_p\alpha^{2p}N^{2p+1}\mathbb{E}\mathcal{S}_{2p}(\nabla F)$$

$$+ C_p NT \frac{\gamma^{2p}}{L^p}(d + \log K)^p \qquad (54)$$

Applying Lemma 26, we conclude:

$$\mathbb{E}\mathcal{S}_{2p}(V) \leq C_p(\alpha N)^{2p}\mathbb{E}\mathcal{S}_{2p}(\nabla F) + T\left[\frac{\gamma^2}{L}(d + \log K)\right]^p + \frac{2^p}{N^{p-1}}\mathbb{E}(\mathcal{S}_2(V))^p \qquad (55)$$

$\square$

Combining Lemmas 24 and 22 gives the following theorem.

**Theorem 4.** *Fix $p \geq 1$. There exist constants $C_p, c_p, \bar{c}_p > 0$ such that whenever: $\gamma \geq C_p\sqrt{L}$, $\alpha\gamma < c_p$, $\frac{\alpha^{3p-1}T^{p-1}L^{2p}}{\gamma^{p+1}} < \bar{c}_p$, the following results hold:*

$$\mathcal{S}_{2p}(\nabla F) \leq C_p \frac{\gamma^{2p-1}}{\alpha}\left[\mathbb{E}\|\tilde{V}_0\|^{2p} + \mathbb{E}|(F(\Psi_0) - F(\Psi_T))^+|^p + 1\right] +$$
$$C_p T\left[\frac{\gamma^{4p}}{L^p} + (\gamma\alpha T)^{p-1}\gamma^{2p}\right](d + \log K)^p$$

$$\mathcal{S}_{2p}(V) \leq C_p \frac{1}{\gamma\alpha}\left[\mathbb{E}\|\tilde{V}_0\|^{2p} + \mathbb{E}|(F(\Psi_0) - F(\Psi_T))^+|^p + 1\right]$$
$$+ C_p T\left[\frac{\gamma^{2p}}{L^p} + (\gamma\alpha T)^{p-1}\right](d + \log K)^p$$

### F.2 Bounding the Bias:

Now, consider the following term in the statement of Theorem 1:

$$B_{tK+i} := \Gamma_{\frac{\alpha}{K}}^{-1}G_{\frac{\alpha}{K}}[b(\hat{\tilde{X}}_{tK+i}) - b(\tilde{X}_{tK+i})]$$

Using Lemma 18, and under the conditions of Theorem 4 holding with $p = 1$, we have:

$$\|B_{tK+i}\|^2 \leq C\frac{\alpha L^2}{K\gamma}\sup_{0 \leq i \leq K-1}\|\hat{\tilde{U}}_{tK+i} - \tilde{U}_{tK+i}\|^2$$
$$\leq C\frac{\alpha^5 L^4 N_t^2}{K\gamma}\sup_{0 \leq i \leq K-1}\|\hat{\tilde{U}}_{tK+i} - \tilde{U}_{tK}\|^2 \qquad (56)$$

In the second step, we have used item 2 of Lemma 19 along with the bounds in Lemma 18.

Applying item 1 from Lemma 21 and Theorem 4 we conclude:

$$\sum_{t=0}^{T-1}\sum_{i=0}^{K-1}\mathbb{E}\|B_{tK+i}\|^2 \leq \frac{C\alpha^5 L^4}{\gamma}\sum_{t=0}^{T-1}\mathbb{E}N_t^2\mathbb{E}\sup_{0\leq i\leq K-1}\|\hat{\tilde{U}}_{tK+i}-\tilde{U}_{tK}\|^2$$

$$\leq \frac{C\alpha^5 L^4}{\gamma}\sum_{t=0}^{T-1}\mathbb{E}\sup_{0\leq i\leq K-1}\|\hat{\tilde{U}}_{tK+i}-\tilde{U}_{tK}\|^2$$

$$\leq \frac{C\alpha^5 L^4}{\gamma}\sum_{t=0}^{T-1}\left[\alpha^2\mathbb{E}\|\tilde{V}_{tK}\|^2+\alpha^4\mathbb{E}\|\nabla F(\tilde{U}_{tK})\|^2+\mathbb{E}M_{t,K}^2\right]$$

$$\leq \frac{C\alpha^5 L^4}{\gamma}\left[\alpha^2\mathbb{E}\mathcal{S}_2(V)+\alpha^4\mathbb{E}\mathcal{S}_2(\nabla F)+T\gamma\alpha^3(d+\log K)\right]$$

$$\leq C\alpha^7 L^3\gamma T(d+\log K)+\frac{C\alpha^6 L^4}{\gamma^2}\left[\mathbb{E}\|\tilde{V}_0\|^2+\mathbb{E}(F(\Psi_0)-F(\Psi_T))^++1\right] \quad (57)$$

In the first step we have used the fact that $N_t$ is independent of $\sup_{0\leq i\leq K-1}\|\hat{\tilde{U}}_{tK+i}-\tilde{U}_{tK}\|^2$ and the second step follows from the fact that $\mathbb{E}N_t^2\leq 2$. In the third step we have used item 1 from Lemma 21. In the fourth step, we have used item 3 of Lemma 21. In the final step we have used Theorem 4.

## F.3 Bounding the Variance:

Now consider $\beta_{tK+i}=\|K\Gamma_{\frac{\alpha}{K}}^{-1}G_{\frac{\alpha}{K}}[b(\hat{\tilde{X}}_{tK+i})-b(\tilde{X}_{tK})]\|$. Using Lemma 18 and item 1 of Lemma 21, we have:

$$\beta_{tK+i}^2 \leq \frac{CK\alpha L^2}{\gamma}\sup_{0\leq i\leq K-1}\|\hat{\tilde{U}}_{tK+i}-\tilde{U}_{tK}\|^2$$

$$\leq \frac{CK\alpha L^2}{\gamma}\left[\alpha^2\|\tilde{V}_{tK}\|^2+\alpha^4\|\nabla F(\tilde{U}_{tK})\|^2+M_{t,K}^2\right] \quad (58)$$

Therefore, we note that for any $p\geq 1$, under the assumptions of Theorem 4, we must have:

$$\sum_{t=0}^{T-1}\sum_{i=0}^{K-1}\mathbb{E}\beta_{tK+i}^{2p} \leq C_p\frac{K^{p+1}\alpha^p L^{2p}}{\gamma^p}\sum_{t=0}^{T-1}\left[\alpha^{2p}\mathbb{E}\|\tilde{V}_{tK}\|^{2p}+\alpha^{4p}\mathbb{E}\|\nabla F(\tilde{U}_{tK})\|^{2p}+\mathbb{E}M_{t,K}^{2p}\right]$$

$$\leq C_p\frac{K^{p+1}\alpha^p L^{2p}}{\gamma^p}\left[\alpha^{2p}\mathbb{E}\mathcal{S}_{2p}(V)+\alpha^{4p}\mathbb{E}\mathcal{S}_{2p}(\nabla F)+T\gamma^p\alpha^{3p}(d+\log K)^p\right]$$

$$\leq C_p\frac{K^{p+1}\alpha^{3p-1} L^{2p}}{\gamma^{p+1}}\left[\mathbb{E}\|\tilde{V}_0\|^{2p}+\mathbb{E}|(F(\Psi_0)-F(\Psi_T))^+|^p+1\right]$$

$$+ C_p\frac{K^{p+1}\alpha^{3p}L^{2p}T}{\gamma^p}\left(\frac{\gamma^{2p}}{L^p}+(\gamma\alpha T)^{p-1}\right)(d+\log K)^p \quad (59)$$

## F.4 Finishing the Proof

For $p\geq 1$, we define $\Delta^{(p)}=\mathbb{E}(F(U_0+\frac{V_0}{\gamma})-F(\mathbf{x}^*))^p+\mathbb{E}\|V_0\|^{2p}+1$. By LHS $\lesssim$ RHS we denote LHS $\leq C.$RHS for some universal positive constant $C$. We now apply Theorem 1 (with $r=7$) along with the bounds in Equations (57) and (59):

$$\mathsf{KL}\left(\mathsf{Law}((X_t^\mathsf{P})_{0\leq t\leq T})\big|\big|\mathsf{Law}((X_{Kt})_{0\leq t\leq T})\right)$$

$$\leq \mathbb{E}[\|B_{sK+i}\|^2] + C\mathbb{E}\left[\frac{\beta_{sK+i}^4}{K^2} + \frac{\beta_{sK+i}^{10}}{K^3} + \frac{\beta_{sK+i}^6}{K^2} + \frac{\beta_{sK+i}^{2r}}{K}\right]$$

$$\lesssim \frac{C\alpha^6 L^4}{\gamma^2}\Delta^{(1)} + \frac{K\alpha^5 L^4}{\gamma^3}\Delta^{(2)} + \frac{K^2\alpha^8 L^6}{\gamma^4}\Delta^{(3)} + \frac{K^3\alpha^{14}L^{10}}{\gamma^6}\Delta^{(5)} + \frac{K^7\alpha^{20}L^{14}}{\gamma^8}\Delta^{(7)}$$

$$+ C\alpha^7 L^3 \gamma T(d + \log K) + K\alpha^6\gamma^2 L^2 T(d + \log K)^2 + \frac{K\alpha^7 L^4 T^2}{\gamma}(d + \log K)^2$$

$$+ K^2\alpha^9 L^3 \gamma^3 T(d + \log K)^3 + \frac{K^2\alpha^{11}L^6 T^3}{\gamma}(d + \log K)^3 + K^3\alpha^{15}L^5\gamma^5 T(d + \log K)^5$$

$$+ \frac{K^3\alpha^{19}L^{10}T^5}{\gamma}(d + \log K)^5 + K^7\alpha^{21}L^7\gamma^7 T(d + \log K)^7$$

$$+ \frac{K^7\alpha^{27}L^{14}T^7}{\gamma}(d + \log K)^7 \tag{60}$$

## G    Convergence Under Isoperimetry Assumptions

We now prove the results concerning convergence under the assumption that $\pi^\star$ satisfies the Logarithmic Sobolev Inequalities (LSI). We first define the LSI by following the discussion in [46] and refer to this work for further details.

**Definition 1.** *We say that a measure $\pi^\star$ over $\mathbb{R}^d$ satsifies $\lambda$-LSI for some $\lambda > 0$, if for every smooth $g : \mathbb{R}^d \to \mathbb{R}$ such that $\mathbb{E}_{X\sim\pi^\star}g^2(X) < \infty$, the following inequality is satisfied:*

$$\mathbb{E}_{\pi^\star}g^2\log g^2 - \mathbb{E}_{\pi^\star}g^2\log\mathbb{E}_{\pi^\star}g^2 \leq \frac{2}{\lambda}\mathbb{E}_{\pi^\star}\|\nabla g\|^2$$

We note that whenever the density $\pi^\star(\mathbf{x}) = e^{-F(\mathbf{x})}$ and $F$ is $\lambda$ strongly convex, then $\pi^\star$ satisfies the $\lambda$-LSI. It is well known that $\lambda$-LSI implies the Poincare inequality (PI) with parameter $\lambda$ defined below:

**Definition 2.** *We say that a measure $\pi^\star$ over $\mathbb{R}^d$ satsifies $\lambda$-PI for some $\lambda > 0$, if for every smooth $g : \mathbb{R}^d \to \mathbb{R}$ such that $\mathbb{E}_{X\sim\pi^\star}g^2(X) < \infty$, the following inequality is satisfied:*

$$\mathbb{E}_{\pi^\star}g^2 - (\mathbb{E}_{\pi^\star}g)^2 \leq \frac{1}{\lambda}\mathbb{E}_{\pi^\star}\|\nabla g\|^2$$

We note the following useful lemma:

**Lemma 25.** *Let $\pi^\star(\mathbf{x}) \propto \exp(-F(\mathbf{x}))$ satisfy $\lambda$-PI or $\lambda$-LSI and let $F$ be $L$-smooth. Then, $L \geq \lambda$.*

*Proof.* Let $X, X' \sim \pi^\star$ be i.i.d. Applying integration by parts, we have: $\mathbb{E}\nabla F(X) = 0$ and $\mathbb{E}\langle X, \nabla F(X)\rangle = d$. Note that LSI implies PI and under $\lambda$-PI, we must have for any unit norm vector $v \in \mathbb{R}^d$:

$$\mathsf{var}(\langle X, v\rangle) \leq \frac{1}{\lambda}$$

Let $\Sigma$ be the covariance of $X$ (it exists due to the assumption of PI). Therefore,

$$\frac{1}{L} = \frac{1}{2dL}\mathbb{E}\langle X - X', \nabla F(X) - \nabla F(X')\rangle$$

$$\leq \frac{1}{2d}\mathbb{E}\|X - X'\|^2 = \frac{\mathsf{Tr}(\Sigma)}{d} \leq \frac{1}{\lambda} \tag{61}$$

In the second step we have used the fact that whenever $F$ is $L$-smooth, then $F(x) - F(y) \leq \langle\nabla F(y), x - y\rangle + \frac{L}{2}\|x - y\|^2$ for all $x, y \in \mathbb{R}^d$. $\qquad\square$

We now state [45, Theorem 1] (adapting to our notation) which gives convergence guarantees for OLMC under the assumption of $\lambda$-LSI.

**Theorem 5.** *Suppose $\pi^*(\mathbf{x}) \propto \exp(-F(\mathbf{x}))$ satisfies the $\lambda$-LSI and that $F$ is $L$-smooth. Consider OLMC with set size $\eta$ which satsifies $0 < \eta \leq \frac{\lambda}{2L^2}$. Then, LMC satisfies:*

$$\mathsf{KL}\left(\mathsf{Law}(X_N)\big\|\pi^\star\right) \leq e^{-\lambda\eta N}\mathsf{KL}\left(\mathsf{Law}(X_0)\big\|\pi^\star\right) + \frac{8dL^2\eta}{\lambda}\,.$$

*Where $X_0, \ldots, X_N$ are the iterates of OLMC.*

Whenever $\epsilon \leq 1$, we take $\frac{\alpha}{K} = \frac{\lambda\epsilon^2}{16dL^2}$. $T = \frac{1}{\alpha\lambda}\log\left(\frac{2\mathsf{KL}\left(\mathsf{Law}(X_0)\big\|\pi^\star\right)}{\epsilon^2}\right)$. With the suggested initialization in [45] i.e $X_0 \sim \mathcal{N}(\mathbf{x}, \frac{\mathbf{I}}{L})$ for any stationary point $\mathbf{x}$ of $F$ we have $\mathsf{KL}\left(\mathsf{Law}(X_0)\big\|\pi^\star\right) \leq \tilde{O}(d)$. Now, take $\alpha = c_0 \min\left(\frac{\epsilon\sqrt{\lambda}}{L^{\frac{3}{2}}d^{\frac{3}{4}}}, \frac{\lambda^{\frac{7}{27}}\epsilon^{\frac{2}{27}}}{L^{\frac{20}{27}}d^{\frac{7}{54}}}\right)$, $T = \tilde{O}(\frac{C_1}{\lambda\alpha})$ and $K = \frac{C_2}{\alpha L\sqrt{d}}$.

Let $X_0, \ldots, X_{TK}$ be the iterates of OLMC with step-size $\frac{\alpha}{K}$ with the parameters as chosen above. Applying Theorem 5, we conclude: $\mathsf{KL}\left(\mathsf{Law}(X_{TK})\big\|\pi^\star\right) \leq \frac{\epsilon^2}{8}$.

Let $X_0^P, \ldots, X_T^P$ be the iterates of $\mathsf{PS}(A, G, \Gamma, \alpha, K)$ with $A, G, \Gamma$ corresponding to OLMC. Applying Theorem 3 (with the explicit lower order terms given in Equation (40)), we conclude that:

$\mathsf{KL}\left(\mathsf{Law}(X_T^P)\big\|\mathsf{Law}(X_{TK})\right) \leq \frac{\epsilon^2}{8}$.

Now, by Pinsker's inequality and the triangle inequality for TV distance, we have:

$$\mathsf{TV}(\mathsf{Law}(X_T^P), \pi^\star) \leq \sqrt{2\mathsf{KL}\left(\mathsf{Law}(X_T^P)\big\|\mathsf{Law}(X_{TK})\right)} + \sqrt{2\mathsf{KL}\left(\mathsf{Law}(X_{TK})\big\|\pi^\star\right)} \leq \epsilon \quad (62)$$

This proves the result in Table 1 for OLMC.

We now consider the analogous result for ULMC below. The following result is a re-statement of [53, Theorem 7]. We note that $C_{\mathsf{LSI}}$ in this work is $\frac{1}{\lambda}$ in our work.

**Theorem 6.** *Let $X_0, \ldots, X_N$ be the iterates of ULMC with step-size $\eta$. We make the following assumptions:*

1. *$F$ is $L$-smooth and that $\pi^\star$ satisfies $\lambda$-LSI.*

2. *Suppose $\mathbf{x}$ be any stationary point of $F$. Initialize $X_0$ such that $U_0, V_0$ are independent with $U_0 \sim \mathcal{N}(\mathbf{x}, \zeta\mathbf{I})$ ($\zeta$ as given in [53]) and $V_0 \sim \mathcal{N}(0, \mathbf{I})$.*

3. *$F(\mathbf{x}) - F(\mathbf{x}^*) = \tilde{O}(d)$ and $\mathbb{E}_{U,V\sim\pi^\star}\|U\| \leq \tilde{O}(d)$.*

*Let $\epsilon \leq 1$. Then, we let $\eta = \tilde{\Theta}(\frac{\epsilon\sqrt{\lambda}}{L\sqrt{d}})$, $c\sqrt{L} \leq \gamma \leq C\sqrt{L}$, $T = \tilde{\Theta}(\frac{L^{\frac{3}{2}}\sqrt{d}}{\lambda^{\frac{3}{2}}\epsilon})$ then,*

$$\mathsf{TV}(\mathsf{Law}(U_N), \pi^\star) \leq \epsilon$$

Let $X_0, \ldots, X_{TK}$ be the iterates of ULMC with step-size $\frac{\alpha}{K}$. We take: $\alpha = \tilde{\Theta}(\sqrt{\frac{\epsilon}{L}}(\frac{\lambda}{Ld})^{\frac{5}{12}})$, $K = \tilde{\Theta}(\frac{1}{\alpha}\frac{\lambda^{\frac{1}{3}}}{d^{\frac{1}{3}}L^{\frac{5}{6}}})$ and $N = \tilde{\Theta}(\frac{\sqrt{L}}{\lambda\alpha})$. We now consider $X_0^P, \ldots, X_T^P$ to be the iterates of $\mathsf{PS}(A, G, \Gamma, \alpha, K)$ with $A, G, \Gamma$ corresponding to ULMC, such that $X_0^P$ and $F$ satisfy the same assumptions in Theorem 6. Applying Theorem 3 along with Pinsker's inequality, we conclude that:

$$\mathsf{TV}(\mathsf{Law}(X_T^{(P)}), \mathsf{Law}(X_{TK})) \leq \frac{\epsilon}{2}\,.$$

Now, using Theorem 6, along with the triangle inequality for TV, we conclude:

$$\mathsf{TV}(\mathsf{Law}(X_T^{(P)}), \pi^\star) \leq \epsilon$$

This proves the bounds for ULMC in Table 1.

# H   Some Technical Results

We state the following technical lemma which will be proved in Section I.3.

**Lemma 26.** *Let $a_1, \ldots, a_T$ be random vectors in $\mathbb{R}^d$. Consider the partitioning $\{1, \ldots, T\} = \cup_{k=1}^{\lceil T/N \rceil} \mathcal{T}_k$ where $\mathcal{T}_k := \{(k-1)N + 1, \ldots, \min(kN, T)\}$. Assume that $N$ divides $T$ and $T \geq N$. Then, we have:*

$$\sum_{j=1}^{T} \mathbb{E}\|a_j\|^{2p} \leq \frac{2^p}{N} \sum_{k=1}^{\frac{T}{N}} \sum_{j \in \mathcal{T}_k} \sum_{\substack{j' \in \mathcal{T}_k \\ j' > j}} \mathbb{E}\|a_j - a_{j'}\|^{2p} + \left(\frac{2}{N}\right)^{p-1} \mathbb{E}(\sum_{j=1}^{T} \|a_j\|^2)^p$$

**Lemma 27.** *Let $Z_1, \ldots, Z_T$ be i.i.d. standard Gaussian random variables, and a sequence of random vectors $g_1, \ldots, g_T$ such that $g_t$ is independent of $Z_t, \ldots, Z_T$. Then,*

$$\mathbb{E}|\sum_{s=1}^{T}\langle g_s, Z_s\rangle|^p \leq C_p \sqrt{\mathbb{E}(\sum_{s=1}^{T} \|g_s\|^2)^p}$$

*Proof.* Let $\lambda \in \mathbb{R}$. Consider the random variable $M_t = \exp(\sum_{s=1}^{t} \lambda \langle g_s, Z_s\rangle - \frac{\lambda^2 \|g_s\|^2}{2})$. $M_t$ is a super martingale and hence, we have: $\mathbb{E}[M_T] \leq 1$. Applying the Chernoff bound, we conclude that for anly $\lambda, \lambda > 0$ we must have:

$$\mathbb{P}(|\sum_{s=1}^{T}\langle g_s, Z_s\rangle| > \frac{\lambda}{2}\sum_{s=1}^{T}\|g_s\|^2 + \frac{h}{\lambda}) \leq 2\exp(-h).$$

$$\implies \mathbb{P}(|\sum_{s=1}^{T}\langle g_s, Z_s\rangle|^p > C_p\lambda^p(\sum_{s=1}^{T}\|g_s\|^2)^p + h) \leq 2\exp(-C_p\lambda h^{\frac{1}{p}})$$

Now, for any variable $X$, $\mathbb{P}(X^+ > h) = \mathbb{P}(X > h)$ whenever $h \geq 0$. Now using the fact that: $\mathbb{E}X \leq \mathbb{E}X^+ = \int_0^\infty \mathbb{P}(X^+ > h)dh$, and taking $X = |\sum_{s=1}^{T}\langle g_s, Z_s\rangle|^p - C_p\lambda^p(\sum_{s=1}^{T}\|g_s\|^2)^p$:

$$\mathbb{E}|\sum_{s=1}^{T}\langle g_s, Z_s\rangle|^p \leq C_p\lambda^p\mathbb{E}(\sum_{s=1}^{T}\|g_s\|^2)^p + \int_0^\infty 2\exp(-c_p\lambda h^{\frac{1}{p}})dh \leq C_p\lambda^p\mathbb{E}(\sum_{s=1}^{T}\|g_s\|^2)^p + \frac{C_p}{\lambda^p}$$

We conclude the result by choosing $\lambda^{2p} = \frac{1}{\mathbb{E}(\sum_{s=1}^{T}\|g_s\|^2)^p}$. □

**Lemma 28.** *Let $K$ be any positive integer. Consider $G \sim \mathsf{Bin}(K, \frac{1}{K})$. Then for any $p \geq 1$, we must have: $\mathbb{E}G^p \leq C_p$ for some constant depending only on $p$.*

*Proof.* The proof is simple for $K = 1$. Assume $K \geq 2$. For any $0 \leq n \leq K$, let $p_n := \mathbb{P}(G = n)$. Then,

$$n!p_n = (1 - \tfrac{1}{K})^{K-n}\frac{K!}{K^n(K-n)!}$$

$$\leq (1 - \tfrac{1}{K})^{K-1} \leq \frac{2}{e}$$

In the second step we have used the inequality $(1 - \frac{1}{n})^n \leq \frac{1}{e}$ and the fact that $K \geq 2$.

Note that $\frac{1}{en!}$ is the probability that the standard poisson random variable is $n$ (denote this by $p_n^*$). Therefore, we must have: $p_n \leq 2p_n^*$. Thus, we must have:

$$\mathbb{E}G^p \leq \sum_{n \in \mathbb{N}} 2n^p p_n^* \leq C_p$$

□

# I  Proofs of Technical Lemmas

## I.1  Proof of Lemma 12

*Proof.*        1. This follows by applying the triangle inequality to the definition of $\hat{\tilde{X}}_{tK+i}$.

2. Consider:

$$\|\hat{\tilde{X}}_{tK+i} - \tilde{X}_{tK+i}\| \leq \alpha \sum_{j=0}^{i-1} H_{t,j} \|\nabla F(\hat{\tilde{X}}_{tK+j}) - \nabla F(\tilde{X}_{tK})\|$$

$$\leq \sum_{j=0}^{i-1} H_{t,j} L \|\hat{\tilde{X}}_{tK+j} - \tilde{X}_{tK}\|$$

$$\leq \alpha N_t L \sup_{0 \leq j \leq K-1} \|\hat{\tilde{X}}_{tK+j} - \tilde{X}_{tK}\| \tag{63}$$

3. This follows by a direct application of Doob's inequality.

$\square$

## I.2  Proof of Lemma 13

*Proof.* We will use $b()$ and $\nabla F()$ interchangeably in the proof. Consider the update

$$\tilde{X}_{(t+1)K} = \tilde{X}_{tK} + \alpha b(\tilde{X}_{tK}) + \alpha \Delta_t + \sqrt{2\alpha} \bar{Y}_t \,.$$

Where $\bar{Y}_t = \frac{1}{\sqrt{K}} \sum_{i=0}^{K-1} Z_{tK+i}$ and $\Delta_t = \sum_{i=0}^{K-1} H_i (b(\hat{\tilde{X}}_{tK+i}) - b(\tilde{X}_{tK}))$

Therefore, we have for any $s > t$:

$$\tilde{X}_{sK} - \tilde{X}_{tK} = \sum_{h=t}^{s-1} \alpha b(\tilde{X}_{hK}) + \alpha \Delta_h + \sqrt{2\alpha} \tilde{Y}_h$$

$$= \alpha(s-t) b(\tilde{X}_{tK}) + \sum_{h=t}^{s-1} \alpha(b(\tilde{X}_{hK}) - b(\tilde{X}_{tK})) + \alpha \Delta_h + \sqrt{2\alpha} \tilde{Y}_h \tag{64}$$

In this proof only, for any random variable $U$, we let $\mathcal{M}_p(U) := (\mathbb{E}[\|U\|^p | \tilde{X}_{tK}])^{\frac{1}{p}}$. Using the triangle inequality for $\mathcal{M}_p$, we conclude:

$$\mathcal{M}_p(\tilde{X}_{sK} - \tilde{X}_{tK}) \leq \alpha(s-t) \mathcal{M}_p(b(\tilde{X}_{tK})) + \sum_{h=t}^{s-1} [\alpha \mathcal{M}_p(b(\tilde{X}_{hK}) - b(\tilde{X}_{tK})) + \alpha \mathcal{M}_p(\Delta_h)]$$

$$+ \sqrt{2\alpha} \mathcal{M}_p\left(\sum_{h=t}^{s-1} \tilde{Y}_h\right)$$

$$\leq \alpha(s-t) \|b(\tilde{X}_{tK})\| + \alpha \sum_{h=t}^{s-1} \left[ L \mathcal{M}_p(\tilde{X}_{hK} - \tilde{X}_{tK}) + \mathcal{M}_p(\Delta_h) \right]$$

$$+ C_p \sqrt{2\alpha d(s-t)} \tag{65}$$

Now, consider $\Delta_h = \sum_{i=0}^{K-1} H_i(b(\hat{\tilde{X}}_{hK+i}) - b(\tilde{X}_{hK}))$. We must have

$$
\begin{aligned}
\mathcal{M}_p(\Delta_h) &\leq \mathcal{M}_p(\sum_{i=0}^{K-1} LH_i\|\hat{\tilde{X}}_{hK+i} - \tilde{X}_{hK}\|) \leq \mathcal{M}_p(LN_h \sup_{0\leq i\leq K-1}\|\hat{\tilde{X}}_{hK+i} - \tilde{X}_{hK}\|) \\
&= LM_p(N_h)\mathcal{M}_p(\sup_{0\leq i\leq K-1}\|\hat{\tilde{X}}_{hK+i} - \tilde{X}_{hK}\|) \\
&\leq LC_p\alpha\mathcal{M}_p(\|b(\tilde{X}_{hK})\|) + LC_p\mathcal{M}_p(M_{h,K}) \\
&\leq LC_p\alpha\|b(\tilde{X}_{tK})\| + L^2C_p\alpha\mathcal{M}_p(\tilde{X}_{hK} - \tilde{X}_{tK}) + LC_p\sqrt{\alpha d}
\end{aligned}
\tag{66}
$$

In the second line, we have used the fact that $N_h$ is independent of $\sup_{0\leq i\leq K-1}\|\hat{\tilde{X}}_{hK+i} - \tilde{X}_{hK}\|$ and the fact that $\mathcal{M}_p(N_t) \leq C_p$ for some constant $C_p$ (Lemma 28 ). In the third step we have applied item 1 from Lemma 12.

Putting this back in Equation 65, we conclude:

$$
\begin{aligned}
\mathcal{M}_p(\tilde{X}_{sK} - \tilde{X}_{tK}) \leq{}& \alpha(s-t)(1+\alpha LC_p)\|b(\tilde{X}_{tK})\| + \alpha L(1+\alpha LC_p)\sum_{h=t}^{s-1}\left[\mathcal{M}_p(\tilde{X}_{hK} - \tilde{X}_{tK})\right] \\
&+ C_p(L\alpha^{\frac{3}{2}}\sqrt{d}(s-t) + \sqrt{2\alpha d(s-t)})
\end{aligned}
\tag{67}
$$

Therefore, we must have:

$$
\begin{aligned}
\sup_{t\leq h\leq s}\mathcal{M}_p(\tilde{X}_{hK} - \tilde{X}_{tK}) \leq{}& \alpha L(s-t)(1+\alpha LC_p)\left[\frac{\|b(\tilde{X}_{tK})\|}{L} + \sup_{t\leq h\leq s}\mathcal{M}_p(\tilde{X}_{hK} - \tilde{X}_{tK})\right] \\
&+ C_p(L\sqrt{d}\alpha^{\frac{3}{2}}(s-t) + \sqrt{2\alpha d(s-t)})
\end{aligned}
\tag{68}
$$

Now, supposing $s - t \leq \frac{1}{\bar{C}_p\alpha L}$ for some large enough constant $\bar{C}_p$ which depends only on $p$. This ensures that $\alpha L(s-t)(1+\alpha LC_p) \leq \frac{1}{2}$ in the equation above. Thus, we have:

$$
\sup_{t\leq h\leq s}\mathcal{M}_p(\tilde{X}_{hK} - \tilde{X}_{tK}) \leq 3\alpha L(s-t)\frac{\|b(\tilde{X}_{tK})\|}{L} + C_p\sqrt{d\alpha(s-t)}
$$

$\square$

### I.3 Proof of Lemma 26

*Proof.* $\bar{a}_k := \frac{1}{|\mathcal{T}_k|}\sum_{j\in\mathcal{T}_k} a_j$ Consider the following quantity:

$$
\begin{aligned}
\sum_{t=1}^{T}\|a_t\|^{2p} &= \sum_k\sum_{j\in\mathcal{T}_k}\|a_j\|^{2p} \\
&\leq \sum_{k=1}^{\frac{T}{N}}\sum_{j\in\mathcal{T}_k}(2^{p-1}\|a_j - \bar{a}_k\|^{2p} + 2^{p-1}\|\bar{a}_k\|^{2p}) \\
&= 2^{p-1}\sum_k\sum_{j\in\mathcal{T}_k}(\|a_j - \bar{a}_k\|^{2p}) + 2^{p-1}N\sum_k\|\bar{a}_k\|^{2p} \\
&\leq 2^{p-1}\sum_k\sum_{j\in\mathcal{T}_k}(\|a_j - \bar{a}_k\|^{2p}) + 2^{p-1}N(\sum_k\|\bar{a}_k\|^2)^p
\end{aligned}
\tag{69}
$$

Now, by Jensen's inequality, we must have: $\|\bar{a}_k\|^2 \leq \frac{1}{|\mathcal{T}_k|}\sum_{j\in\mathcal{T}_k}\|a_j\|^2$. Therefore, we conclude:

$$\sum_{t=1}^{T} \|a_t\|^{2p} \le 2^{p-1} \sum_k \sum_{j \in \mathcal{T}_k} (\|a_j - \bar{a}_k\|^{2p}) + 2^{p-1} N^{1-p} (\sum_{j=1}^{T} \|a_j\|^2)^p \tag{70}$$

Now, consider

$$\sum_{j \in \mathcal{T}_k} \mathbb{E}\|a_j - \bar{a}_k\|^{2p} \le \frac{1}{N} \sum_{j \in \mathcal{T}_k} \sum_{j' \in \mathcal{T}_k} \mathbb{E}\|a_j - a_{j'}\|^{2p}$$

$$= \frac{2}{N} \sum_{j \in \mathcal{T}_k} \sum_{j' > j} \mathbb{E}\|a_j - a_{j'}\|^{2p} \tag{71}$$

Using this with Equation (70), we conclude the statement of the lemma.

$\square$

### I.4 Proof of Lemma 14

*Proof.* We now apply Lemma 26 with $a_t := b(\tilde{X}_{(t-1)K})$. Whenever $\alpha L \le c_p$ for some small enough constant $c_p$, we can take $N$ to be the largest integer power of 2 lower than $\lceil \frac{1}{\alpha L \bar{C}_{2p}} \rceil$ for a large enough constant $\bar{C}_{2p}$ such that the result of Lemma 13 with $s - t \le N$. Applying Lemma 13, we conclude:

$$\frac{2}{N} \sum_{j \in \mathcal{T}_k} \sum_{j' > j} \mathbb{E}\|a_j - a_j'\|^{2p} \le 2 \sum_{j \in \mathcal{T}_k} \mathbb{E}(3\alpha LN\|a_j\| + C_p L\sqrt{\alpha dN})^{2p}$$

$$\le C_p L^{2p} (\alpha d)^p N^{p+1} + C_p (\alpha LN)^{2p} \sum_{j \in \mathcal{T}_k} \mathbb{E}\|a_j\|^{2p} \tag{72}$$

Applying Lemma 26

$$\sum_{t=1}^{T} \mathbb{E}\|a_t\|^{2p} \le 2^{p-1} C_p L^{2p} (\alpha dN)^p T + C_p (\alpha LN)^{2p} \sum_k \sum_{j \in \mathcal{T}_k} \mathbb{E}\|a_j\|^{2p} + 2^{p-1} N^{1-p} (\sum_{j=1}^{T} \|a_j\|^2)^p \tag{73}$$

Note that whenever the constant $\bar{C}_{2p}$ defining $N$ above is large enough, this implies

$$\sum_{t=1}^{T} \mathbb{E}\|a_t\|^{2p} \le C_p L^p d^p T + C_p (\alpha L)^{p-1} (\sum_{j=1}^{T} \|a_j\|^2)^p \tag{74}$$

$\square$

### I.5 Proof of Lemma 15

*Proof.* Since $\nabla F$ is $L$ Lipschitz, we have:

$$F(\tilde{X}_{(t+1)K}) - F(\tilde{X}_{tK}) \le \langle \nabla F(\tilde{X}_{tK}), \tilde{X}_{(t+1)K} - \tilde{X}_{tK} \rangle + \frac{L}{2}\|\tilde{X}_{(t+1)K} - \tilde{X}_{tK}\|^2 \tag{75}$$

Also note that:
$$\tilde{X}_{(t+1)K} = \tilde{X}_{tK} - \alpha \nabla F(\tilde{X}_{tK}) + \alpha \Delta_t + \sqrt{2\alpha}\bar{Y}_t.$$

Where $\bar{Y}_t = \frac{1}{\sqrt{K}} \sum_{i=0}^{K-1} Z_{tK+i}$ and $\Delta_t = -\sum_{i=0}^{K-1} H_{t,i}(\nabla F(\hat{\tilde{X}}_{tK+i}) - \nabla F(\tilde{X}_{tK}))$

Thus, summing Equation (75) from $t = 0$ to $t = T - 1$, we conclude:

$$F(\tilde{X}_{TK}) - F(\tilde{X}_0) \leq \sum_{t=0}^{T-1} \langle \nabla F(\tilde{X}_{tK}), \tilde{X}_{(t+1)K} - \tilde{X}_{tK} \rangle + \sum_{t=0}^{T-1} \frac{L}{2} \|\tilde{X}_{(t+1)K} - \tilde{X}_{tK}\|^2$$

$$\leq \sum_{t=0}^{T-1} \langle \nabla F(\tilde{X}_{tK}), \tilde{X}_{(t+1)K} - \tilde{X}_{tK} \rangle + \sum_{t=0}^{T-1} \frac{3L\alpha^2}{2}[\|\nabla F(\tilde{X}_{tK})\|^2 + \|\Delta_t\|^2] + 3L\alpha\|\bar{Y}_t\|^2$$

$$\leq \sum_{t=0}^{T-1} -(\alpha - \frac{3\alpha^2 L}{2})\|\nabla F(\tilde{X}_{tK})\|^2 + \alpha\langle \nabla F(\tilde{X}_{tK}), \Delta_t \rangle + \frac{3L\alpha^2}{2}\|\Delta_t\|^2 + 3L\alpha\|\bar{Y}_t\|^2$$

$$+ \sqrt{2\alpha}\langle \nabla F(\tilde{X}_{tK}), \bar{Y}_t \rangle$$

$$\leq \sum_{t=0}^{T-1} -\frac{\alpha}{4}\|\nabla F(\tilde{X}_{tK})\|^2 + 3\alpha\|\Delta_t\|^2 + 3L\alpha\|\bar{Y}_t\|^2 + \sqrt{2\alpha}\langle \nabla F(\tilde{X}_{tK}), \bar{Y}_t \rangle \tag{76}$$

In the last step, we have used the fact that $2|\langle \nabla F(\tilde{X}_{tK}), \Delta_t \rangle| \leq \|\nabla F(\tilde{X}_{tK})\|^2 + \|\Delta_t\|^2$ and the assumption that $\alpha L \leq c_p$ for some small enough constant $c_p$.

Therefore, we conclude:

$$(\sum_{t=0}^{T-1} \|\nabla F(\tilde{X}_{tK})\|^2)^p \leq \frac{C_p}{\alpha^p}|(F(X_0) - F(\tilde{X}_{KT}))^+|^p + C_p(\sum_{t=0}^{T-1} \|\Delta_t\|^2)^p + C_p L^p (\sum_{t=0}^{T-1} \|\bar{Y}_t\|^2)^p$$

$$+ \frac{C_p}{\alpha^{\frac{p}{2}}}|\sum_{t=0}^{T-1} \langle \nabla F(\tilde{X}_{tK}), \bar{Y}_t \rangle|^p \tag{77}$$

The properties of Gaussians show that: $\mathbb{E}(\sum_{t=0}^{T-1} \|\bar{Y}_t\|^2)^p \leq C_p T^p d^p$.

By Lemma 27 and the AM-GM inequality, we conclude that for any $\kappa > 0$ arbitrary, we have:

$$\frac{1}{\alpha^{p/2}}\mathbb{E}|\sum_{t=0}^{T-1} \langle \nabla F(\tilde{X}_{tK}), \bar{Y}_t \rangle|^p \leq \frac{C_p}{\alpha^{p/2}}\sqrt{\mathbb{E}(\sum_{t=0}^{T-1} \|\nabla F(\tilde{X}_{tK})\|^2)^p}$$

$$\leq \frac{\kappa C_p}{2\alpha^p} + \frac{C_p}{2\kappa}\mathbb{E}(\sum_{t=0}^{T-1} \|\nabla F(\tilde{X}_{tK})\|^2)^p \tag{78}$$

$$\mathbb{E}(\sum_{t=0}^{T-1} \|\Delta_t\|^2)^p \leq T^{p-1}\sum_{t=0}^{T-1} \mathbb{E}\|\Delta_t\|^{2p}$$

$$\leq T^{p-1}\sum_{t=0}^{T-1} \mathbb{E}(N_t)^{2p}L^{2p}\mathbb{E}(\sup_{0 \leq i \leq K-1}\|\hat{\tilde{X}}_{tK+i} - \tilde{X}_{tK}\|)^{2p}$$

$$\leq C_p T^{p-1}\sum_{t=0}^{T-1} L^{2p}\mathbb{E}(\sup_{0 \leq i \leq K-1}\|\hat{\tilde{X}}_{tK+i} - \tilde{X}_{tK}\|)^{2p} \tag{79}$$

In the first step, we have used Jensen's inequality. In the second step we have used the fact that $\nabla F$ is $L$-Lipshcitz and that $N_t$ is independent of $\hat{\tilde{X}}_{tK+i}$ for $0 \leq i \leq K-1$. In the last step, we have used the fact that $\mathbb{E}(N_t)^{2p} \leq C_p$ for some constant $C_p$ (Lemma 28).

By Lemma 12, we have:

$$\mathbb{E}\sup_{0 \leq i \leq k}\|\hat{\tilde{X}}_{tK+i} - \tilde{X}_{tK}\|^{2p} \leq C_p\alpha^{2p}\mathbb{E}\|\nabla F(\tilde{X}_{tK})\|^{2p} + C_p\alpha^p d^p$$

Plugging all of these bounds, and taking $\kappa$ in Equation (78) to be large enough, we conclude:

$$\mathbb{E}(\sum_{t=0}^{T-1}\|\nabla F(\tilde{X}_{tK})\|^2)^p \leq \frac{C_p}{\alpha^p}\mathbb{E}|(F(X_0)-F(\tilde{X}_{KT}))^+|^p + C_p T^{p-1}L^{2p}\alpha^{2p}\mathbb{E}\sum_{t=0}^{T-1}\|\nabla F(\tilde{X}_{tK})\|^{2p}$$

$$+ C_p L^p d^p T^p + \frac{C_p}{\alpha^p} \tag{80}$$

$\square$

## I.6 Proof of Lemma 18

*Proof.* For the sake of clarity, define $x = 1 - \exp(-\gamma h)$. $\rho = \frac{(1-x)^2}{\gamma}\frac{1}{\sqrt{\frac{2}{\gamma}(h-\frac{2(1-x)}{\gamma}+\frac{(1-x^2)}{2\gamma})(1-x^2)}}$

Algebraic manipulations show that:

$$\rho = \sqrt{\frac{(1-x)^3}{2(1+x)\left(\gamma h - (1-x) - \frac{(1-x)^2}{2}\right)}} \tag{81}$$

Now note that $\gamma h = -\log(1-(1-x)) = \sum_{i=1}^{\infty}\frac{(1-x)^i}{i}$. Therefore, we conclude: $\gamma h - (1-x) - \frac{(1-x)^2}{2} \geq \frac{(1-x)^3}{3} + \frac{(1-x)^4}{4}$. Therefore,

$$\rho \leq \sqrt{\frac{1}{2(1+x)\left(\frac{1}{3}+\frac{1-x}{4}\right)}}$$

$$\leq \sqrt{\frac{1}{2\inf_{t\in[0,1]}(1+t)\left(\frac{1}{3}+\frac{1-t}{4}\right)}} \leq \sqrt{\frac{6}{7}} \tag{82}$$

Now, we note that for scalars $A, B > 0$ and $\rho \in [0,1]$, we have:

$$\begin{bmatrix} A^2\mathbf{I}_d & \rho AB\mathbf{I}_d \\ \rho AB\mathbf{I}_d & B^2\mathbf{I}_d \end{bmatrix} \succeq \begin{bmatrix} A^2(1-\rho)\mathbf{I}_d & 0 \\ 0 & B^2(1-\rho)\mathbf{I}_d \end{bmatrix}$$

Thus, we conclude from the above computations that:

$$\Gamma_h^2 \succeq c\begin{bmatrix} \frac{2}{\gamma}\left(h-\frac{2}{\gamma}(1-\exp(-\gamma h))+\frac{1}{2\gamma}(1-\exp(-2\gamma h))\right)\mathbf{I}_d & 0 \\ 0 & (1-\exp(-2\gamma h))\mathbf{I}_d \end{bmatrix}$$

$$\succeq c\begin{bmatrix} \frac{2}{3\gamma^2}(1-\exp(-\gamma h))^3\mathbf{I}_d & 0 \\ 0 & (1-\exp(-2\gamma h))\mathbf{I}_d \end{bmatrix} =: \Gamma_{\text{ub}}^2 \tag{83}$$

We have used the fact that:

$$\frac{2}{\gamma}\left(h-\frac{2}{\gamma}(1-\exp(-\gamma h))+\frac{1}{2\gamma}(1-\exp(-2\gamma h))\right) \geq \frac{2}{3\gamma^2}(1-\exp(-\gamma h))^3$$

Using the fact that for PSD matrices $A, B$ $A \preceq B$ implies $B^{-1} \preceq A^{-1}$, we have:

$$G_h^\mathsf{T}\Gamma_h^{-2}G_h \preceq G_h^\mathsf{T}\Gamma_{\text{ub}}^{-2}G_h \preceq \begin{bmatrix} C\frac{h\exp(2\gamma h)}{\gamma}\mathbf{I}_d & 0 \\ 0 & 0 \end{bmatrix} \tag{84}$$

The second inequality follows easily by elementary manipulations.

$\square$

## I.7 Proof of Lemma 20

*Proof.* Explicit computations show that:

$$\psi_{t+1} = \psi_t - \frac{\alpha}{\gamma}\nabla F(\tilde{U}_{tK}) + \alpha \begin{bmatrix} \mathbf{I}_d & \frac{\mathbf{I}_d}{\gamma} \end{bmatrix} \Delta_t + \begin{bmatrix} \mathbf{I}_d & \frac{\mathbf{I}_d}{\gamma} \end{bmatrix} \Gamma_\alpha \tilde{Y}_t$$

Now, note that $\begin{bmatrix} \mathbf{I}_d & \frac{\mathbf{I}_d}{\gamma} \end{bmatrix} A_h = \begin{bmatrix} \mathbf{I} & \frac{\mathbf{I}_d}{\gamma} \end{bmatrix}$ and $\begin{bmatrix} \mathbf{I}_d & \frac{\mathbf{I}_d}{\gamma} \end{bmatrix} G_h = \begin{bmatrix} \frac{h}{\gamma}\mathbf{I}_d & 0 \end{bmatrix}$. Therefore, from the definition of $\Delta_t$, we conclude that:

$$\alpha \begin{bmatrix} \mathbf{I}_d & \frac{\mathbf{I}_d}{\gamma} \end{bmatrix} \Delta_t = -\sum_{i=0}^{K-1} \frac{H_i \alpha}{\gamma} \left[ \nabla F(\hat{\tilde{U}}_{tK+i}) - \nabla F(\tilde{U}_{tK}) \right]$$

A straight forward calculation shows that: $\tilde{\Psi}_t := \begin{bmatrix} \mathbf{I}_d & \frac{\mathbf{I}_d}{\gamma} \end{bmatrix} \Gamma_\alpha \tilde{Y}_t \sim \mathcal{M}(0, \frac{2h}{\gamma}\mathbf{I}_d)$. $\qquad\square$

## I.8 Proof of Lemma 21

*Proof.* 1. Note that

$$\hat{\tilde{U}}_{tK+i} - \tilde{U}_{tK} = \Pi(A_{\frac{i\alpha}{K}} - \mathbf{I})\tilde{X}_{tK} + \Pi G_{\frac{i\alpha}{K}} b(\tilde{X}_{tK}) + \sum_{j=0}^{i-1} \Pi A_{\frac{\alpha(i-1-j)}{K}} \Gamma_{\frac{\alpha}{K}} Z_{tK+j}$$

We bound each of the terms separately.

$$\|(A_{\frac{i\alpha}{K}} - \mathbf{I})\tilde{X}_{tK}\|_\Pi \leq \frac{i\alpha}{K}\|\tilde{V}_{tK}\| \leq \alpha\|\tilde{V}_{tK}\|$$

$$\|G_{\frac{i\alpha}{K}} b(\tilde{X}_{tK})\|_\Pi \leq \frac{\alpha^2 i^2}{2K^2}\|\nabla F(\tilde{U}_{tK})\| \leq \frac{\alpha^2}{2}\|\nabla F(\tilde{U}_{tK})\|$$

Plugging these inequalities gives the result.

2. Since $\nabla F$ is $L$-Lipschitz, we have:

$$F(\psi_{t+1}) - F(\psi_t) \leq \langle \nabla F(\psi_t), \psi_{t+1} - \psi_t \rangle + \frac{L}{2}\|\psi_{t+1} - \psi_t\|^2$$

$$= \langle \nabla F(\psi_t) - \nabla F(\tilde{U}_{tK}), \psi_{t+1} - \psi_t \rangle + \langle \nabla F(\tilde{U}_{tK}), \psi_{t+1} - \psi_t \rangle + \frac{L}{2}\|\psi_{t+1} - \psi_t\|^2$$

$$\tag{85}$$

Using Lemma 20 to evaluate $\psi_{t+1} - \psi_t$, we conclude:

$$\langle \nabla F(\tilde{U}_{tK}), \psi_{t+1} - \psi_t \rangle = -\frac{\alpha}{\gamma} \|\nabla F(\tilde{U}_{tK})\|^2 + \langle \nabla F(U_t), \tilde{\Psi}_t \rangle$$

$$- \sum_{i=0}^{K-1} \frac{H_{t,i}\alpha}{\gamma} \langle \nabla F(\tilde{U}_{tK}), \nabla F(\hat{\tilde{U}}_{tK+i}) - \nabla F(\tilde{U}_{tK}) \rangle$$

$$\leq -\frac{\alpha}{\gamma} \|\nabla F(\tilde{U}_{tK})\|^2 + \langle \nabla F(U_t), \tilde{\Psi}_t \rangle$$

$$+ \frac{N_t \alpha L}{\gamma} \|\nabla F(\tilde{U}_{tK})\| \sup_{0 \leq i \leq K-1} \|\hat{\tilde{U}}_{tK+i} - \tilde{U}_{tK}\|$$

$$\leq -\frac{\alpha}{\gamma} \|\nabla F(\tilde{U}_{tK})\|^2 + \langle \nabla F(U_t), \tilde{\Psi}_t \rangle$$

$$+ \frac{\alpha}{2\gamma} \|\nabla F(\tilde{U}_{tK})\|^2 + \frac{N_t^2 \alpha L^2}{2\gamma} \sup_{0 \leq i \leq K-1} \|\hat{\tilde{U}}_{tK+i} - \tilde{U}_{tK}\|^2$$

$$= -\frac{\alpha}{2\gamma} \|\nabla F(\tilde{U}_{tK})\|^2 + \langle \nabla F(U_t), \tilde{\Psi}_t \rangle$$

$$+ \frac{N_t^2 \alpha L^2}{2\gamma} \sup_{0 \leq i \leq K-1} \|\hat{\tilde{U}}_{tK+i} - \tilde{U}_{tK}\|^2 \tag{86}$$

In the second step, we have use the fact that $\nabla F$ is $L$-Lipschitz along with the Cauchy-Schwarz inequality. In the third step, we have used the AM-GM inequality which states that for any $a, b \geq 0$, we have $2ab \leq a^2 + b^2$

Similarly, we note that:

$$\langle \nabla F(\psi_t) - \nabla F(\tilde{U}_{tK}), \psi_{t+1} - \psi_t \rangle \leq \frac{\alpha L}{\gamma^2} \|\nabla F(\tilde{U}_{tK})\| \|\tilde{V}_{tK}\| + \langle \nabla F(\psi_t) - \nabla F(\tilde{U}_{tK}), \tilde{\Psi}_t \rangle$$

$$+ \frac{N_t \alpha L^2}{\gamma^2} \|\tilde{V}_{tK}\| \sup_{0 \leq i \leq K-1} \|\hat{\tilde{U}}_{tK+i} - \tilde{U}_{tK}\|$$

$$\leq \frac{\alpha}{4\gamma} \|\nabla F(\tilde{U}_{tK})\|^2 + \frac{3\alpha L^2 \|\tilde{V}_{tK}\|^2}{2\gamma^3} + \langle \nabla F(\psi_t) - \nabla F(\tilde{U}_{tK}), \tilde{\Psi}_t \rangle$$

$$+ \frac{N_t^2 \alpha L^2}{2\gamma} \sup_{0 \leq i \leq K-1} \|\hat{\tilde{U}}_{tK+i} - \tilde{U}_{tK}\|^2 \tag{87}$$

We have again used the AM-GM inequality in the second step above. Convexity of $x \to x^2$ implies:

$$\frac{L}{2} \|\psi_{t+1} - \psi_t\|^2 = \frac{3L}{2} \left[ \frac{\alpha^2}{\gamma^2} \|\nabla F(\tilde{U}_{tK})\|^2 + \frac{N_t^2 \alpha^2 L^2}{\gamma^2} \sup_{0 \leq i \leq K-1} \|\hat{\tilde{U}}_{tK+i} - \tilde{U}_{tK}\|^2 + \|\tilde{\Psi}_t\|^2 \right] \tag{88}$$

Plugging these upper bounds into Equation (85), we have:

$$F(\psi_{t+1}) - F(\psi_t) \leq -\frac{\alpha}{4\gamma}\left(1 - \frac{6\alpha L}{\gamma}\right)\|\nabla F(\tilde{U}_{tK})\|^2 + \langle \nabla F(\psi_t) - \nabla F(\tilde{U}_{tK}), \tilde{\Psi}_t \rangle$$

$$+ \langle \nabla F(U_t), \tilde{\Psi}_t \rangle + \frac{3L}{2} \|\tilde{\Psi}_t\|^2 + \frac{3\alpha L^2 \|\tilde{V}_{tK}\|^2}{2\gamma^3}$$

$$+ \frac{N_t^2 \alpha L^2}{\gamma} \left(1 + \frac{3\alpha L}{2\gamma}\right) \sup_{0 \leq i \leq K-1} \|\hat{\tilde{U}}_{tK+i} - \tilde{U}_{tK}\|^2 \tag{89}$$

3. Using the scaling relations (Section 2.1), it is easy to show that for any $i$ such that $0 \leq i \leq K-1$, $\sum_{j=0}^{i-1} \Pi A_{\frac{\alpha(i-1-j)}{K}} \Gamma_{\frac{\alpha}{K}} Z_{tK+j}$ is a Gaussian with covariance matrix $\Sigma_i$ such that $\Sigma_i \preceq \frac{2\exp(\gamma\alpha)\gamma\alpha^3}{3} \mathbf{I}_d$.

Applying Gaussian concentration for $\sum_{j=0}^{i-1} \Pi A_{\frac{\alpha(i-1-j)}{K}} \Gamma_{\frac{\alpha}{K}} Z_{tK+j}$ along with the union bound over all $0 \le i \le K-1$, we conclude:

$$\mathbb{P}(M_{t,K} > C \exp(\tfrac{\gamma\alpha}{2})\sqrt{\gamma\alpha^3}(\sqrt{d} + \beta + \sqrt{\log K})) \le \exp(-\beta^2/2)$$

We conclude the moment bounds by integrating tail probabilities.

$\square$

## I.9   Proof of Lemma 22

*Proof.*     1. Let $g := e^{-\gamma\alpha}$. Then, we have:

$$\tilde{V}_{(t+1)K} = g\tilde{V}_{tK} - \frac{1-g}{\gamma}\nabla F(\tilde{U}_{tK}) + (\mathbf{I} - \Pi)(\alpha\Delta_t + \Gamma_\alpha \tilde{Y}_t) \tag{90}$$

Now, consider:

$$
\begin{aligned}
\|\tilde{V}_{(t+1)K}\|^2 &= g^2\|\tilde{V}_{tK}\|^2 + \frac{(1-g)^2}{\gamma^2}\|\nabla F(\tilde{U}_{tK})\|^2 + \|(\mathbf{I} - \Pi)(\alpha\Delta_t + \Gamma_\alpha \tilde{Y}_t)\|^2 \\
&\quad - 2\frac{g(1-g)}{\gamma}\langle \nabla F(\tilde{U}_{tK}), \tilde{V}_{tK}\rangle + 2g\langle \tilde{V}_{tK}, (\mathbf{I} - \Pi)(\alpha\Delta_t + \Gamma_\alpha \tilde{Y}_t)\rangle \\
&\quad - 2\frac{(1-g)}{\gamma}\langle \nabla F(\tilde{U}_{tK}), (\mathbf{I} - \Pi)(\alpha\Delta_t + \Gamma_\alpha \tilde{Y}_t)\rangle \\
&\le g\|\tilde{V}_{tK}\|^2 + \frac{(1-g)}{\gamma^2}\|\nabla F(\tilde{U}_{tK})\|^2 + \|(\mathbf{I} - \Pi)(\alpha\Delta_t + \Gamma_\alpha \tilde{Y}_t)\|^2 \\
&\quad + 2g\langle \tilde{V}_{tK}, (\mathbf{I} - \Pi)(\alpha\Delta_t + \Gamma_\alpha \tilde{Y}_t)\rangle \\
&\quad - 2\frac{(1-g)}{\gamma}\langle \nabla F(\tilde{U}_{tK}), (\mathbf{I} - \Pi)(\alpha\Delta_t + \Gamma_\alpha \tilde{Y}_t)\rangle \\
&\le \sqrt{g}\|\tilde{V}_{tK}\|^2 + \frac{2(1-g)}{\gamma^2}\|\nabla F(\tilde{U}_{tK})\|^2 + \frac{4}{1-\sqrt{g}}\|(\mathbf{I} - \Pi)(\alpha\Delta_t)\|^2 \\
&\quad + 2\|(\mathbf{I} - \Pi)\Gamma_\alpha \tilde{Y}_t)\|^2 + 2g\langle \tilde{V}_{tK}, (\mathbf{I} - \Pi)(\Gamma_\alpha \tilde{Y}_t)\rangle \\
&\quad - 2\frac{(1-g)}{\gamma}\langle \nabla F(\tilde{U}_{tK}), (\mathbf{I} - \Pi)(\Gamma_\alpha \tilde{Y}_t)\rangle \tag{91}
\end{aligned}
$$

In the second step, we have use the fact that $|\frac{2g(1-g)}{\gamma}\langle \nabla F(\tilde{U}_{tK}), \tilde{V}_{tK}\rangle| \le g(1 - g)\frac{\|\nabla F(\tilde{U}_{tK})\|^2}{\gamma^2} + g(1 - g)\|\tilde{V}_{tK}\|^2$. In the third step, we have used the fact that: $2g\langle \tilde{V}_{tK}, (\mathbf{I} - \Pi)\alpha\Delta_t\rangle \le (\sqrt{g} - g)\|\tilde{V}_{tK}\|^2 + \frac{\|(\mathbf{I}-\Pi)\alpha\Delta_t\|^2}{1-\sqrt{g}}$, $-\frac{2(1-g)}{\gamma}\langle \nabla F(\tilde{U}_{tK}), (\mathbf{I} - \Pi)\alpha\Delta_t\rangle \le \frac{1-g}{\gamma^2}\|\nabla F(\tilde{U}_{tK})\|^2 + (1-g)\|(\mathbf{I} - \Pi)\alpha\Delta\|^2$ and $\|(\mathbf{I} - \Pi)(\alpha\Delta_t + \Gamma_\alpha \tilde{Y}_t)\|^2 \le 2\|(\mathbf{I} - \Pi)(\alpha\Delta_t)\|^2 + 2\|(\mathbf{I} - \Pi)\Gamma_\alpha \tilde{Y}_t\|^2$.

With extention of notation, let $\mathcal{S}_2((\mathbf{I}-\Pi)\alpha\Delta) := \sum_{t=0}^{T-1} \|(\mathbf{I}-\Pi)\Delta_t\|^2$, $\mathcal{S}_2((\mathbf{I}-\Pi)\Gamma_\alpha \tilde{Y}) := \sum_{t=0}^{T-1} \|(\mathbf{I} - \Pi)\Gamma_\alpha \tilde{Y}_t\|^2$. Using Equation (91), we conclude:

$$
\begin{aligned}
\mathcal{S}_2(V) &\le \frac{\|\tilde{V}_0\|^2 - \|\tilde{V}_{tK}\|^2}{1 - \sqrt{g}} + \frac{4}{\gamma^2}\mathcal{S}_2(\nabla F) + \frac{4}{(1-\sqrt{g})^2}\mathcal{S}_2((\mathbf{I} - \Pi)\alpha\Delta) \\
&\quad + \frac{2\mathcal{S}_2((\mathbf{I} - \Pi)\Gamma_\alpha \tilde{Y})}{1 - \sqrt{g}} + \sum_{t=0}^{T-1} 2\frac{g}{1 - \sqrt{g}}\langle \tilde{V}_{tK}, (\mathbf{I} - \Pi)(\Gamma_\alpha \tilde{Y}_t)\rangle \\
&\quad - 2\frac{(1-g)}{\gamma(1 - \sqrt{g})}\langle \nabla F(\tilde{U}_{tK}), (\mathbf{I} - \Pi)(\Gamma_\alpha \tilde{Y}_t)\rangle \tag{92}
\end{aligned}
$$

Note that $(\mathbf{I} - \Pi)\Gamma_\alpha \tilde{Y}_t \sim \mathcal{N}(0, (1 - e^{-2\gamma\alpha})\mathbf{I}_d)$. By properties of Gaussians, it is clear that for any $p \geq 1$, we must have $\mathbb{E}\left(\mathcal{S}_2((\mathbf{I} - \Pi)\Gamma_\alpha \tilde{Y})\right)^p \leq C_p(\gamma\alpha Td)^p$. By Lemma 27, we conclude:

$$\mathbb{E}|\sum_{t=0}^{T-1}\langle \tilde{V}_{tK}, (\mathbf{I} - \Pi)\Gamma_\alpha \tilde{Y}_t\rangle|^p \leq C_p(\gamma\alpha)^{\frac{p}{2}}\sqrt{\mathbb{E}(\mathcal{S}_2(V))^p}$$

$$\mathbb{E}|\sum_{t=0}^{T-1}\langle \nabla F(\tilde{U}_{tK}), (\mathbf{I} - \Pi)\Gamma_\alpha \tilde{Y}_t\rangle|^p \leq C_p(\gamma\alpha)^{\frac{p}{2}}\sqrt{\mathbb{E}(\mathcal{S}_2(\nabla F))^p}$$

By $L$ smoothness of $F$, we have: $\|(\mathbf{I} - \Pi)(\alpha\Delta_t)\| \leq L\alpha N_t \sup_{0 \leq i \leq K}\|\hat{\tilde{U}}_{tK+i} - \tilde{U}_{tK}\|$. Using the result in Lemma 21, we conclude:

$$\mathbb{E}(\mathcal{S}_2((\mathbf{I} - \Pi)\alpha\Delta))^p \leq C_p L^{2p}\mathbb{E}(\sum_{t=0}^{T-1}\alpha^4 N_t^2\|\tilde{V}_{tK}\|^2 + \alpha^6 N_t^2\mathbb{E}\|\nabla F(\tilde{U}_{tK})\|^2 + \alpha^2 N_t^2 M_{t,K}^2)^p$$

$$\leq C_p L^{2p}T^{p-1}\mathbb{E}(\sum_{t=0}^{T-1}\alpha^{4p}N_t^{2p}\|\tilde{V}_{tK}\|^{2p} + \alpha^{6p}N_t^{2p}\mathbb{E}\|\nabla F(\tilde{U}_{tK})\|^{2p} + \alpha^{2p}N_t^{2p}M_{t,K}^{2p})$$

$$\leq C_p L^{2p}T^{p-1}\alpha^{4p}\mathbb{E}\mathcal{S}_{2p}(V) + C_p T^{p-1}L^{2p}\alpha^{6p}\mathbb{E}\mathcal{S}_{2p}(\nabla F) + C_p L^{2p}T^p\alpha^{5p}\gamma^p(d + \log K)^p \tag{93}$$

In the second step, we have used jensen's inequality to show that $(\frac{1}{T}\sum_t a_t^2)^p \leq \frac{1}{T}\sum_t a_t^{2p}$. In the third step, we have used the fact that $N_t$ is independent of $\tilde{X}_{tK}$ and that $\mathbb{E}N_t^{2p} \leq C_p$ for some constant $C_p$ (Lemma 28).

We now use the fact that for any two random variables $X, Y$ such that $\mathbb{E}\|X\|^p, \mathbb{E}\|Y\|^p < \infty$, $[\mathbb{E}\|X + Y\|^p]^{\frac{1}{p}} \leq [\mathbb{E}\|X\|^p]^{\frac{1}{p}} + [\mathbb{E}\|Y\|^p]^{\frac{1}{p}}$. Using the bounds established above and applying them to Equation (91), we conclude:

$$[\mathbb{E}(\mathcal{S}_2(V))^p]^{\frac{1}{p}} \leq \frac{[\mathbb{E}\|\tilde{V}_0\|^{2p}]^{\frac{1}{p}}}{1 - \sqrt{g}} + \frac{4}{\gamma^2}[\mathbb{E}(\mathcal{S}_2(\nabla F))^p]^{\frac{1}{p}} + \frac{C(\mathbb{E}(\mathcal{S}_2((\mathbf{I} - \Pi)\alpha\Delta))^p)^{\frac{1}{p}}}{(1 - \sqrt{g})^2}$$

$$+ C_p\frac{\gamma\alpha Td}{1 - \sqrt{g}} + C_p\frac{\sqrt{\gamma\alpha}[\mathbb{E}(\mathcal{S}_2(V))^p]^{\frac{1}{2p}}}{1 - \sqrt{g}} + C_p\sqrt{\frac{\alpha}{\gamma}}[\mathbb{E}(\mathcal{S}_2(\nabla F))^p]^{\frac{1}{2p}}$$

$$\leq \frac{C}{\gamma\alpha}[\mathbb{E}\|\tilde{V}_0\|^{2p}]^{\frac{1}{p}} + \frac{4}{\gamma^2}[\mathbb{E}(\mathcal{S}_2(\nabla F))^p]^{\frac{1}{p}} + \frac{C(\mathbb{E}(\mathcal{S}_2((\mathbf{I} - \Pi)\alpha\Delta))^p)^{\frac{1}{p}}}{\gamma^2\alpha^2}$$

$$+ C_p Td + C_p\frac{[\mathbb{E}(\mathcal{S}_2(V))^p]^{\frac{1}{2p}}}{\sqrt{\gamma\alpha}} + C_p\sqrt{\frac{\alpha}{\gamma}}[\mathbb{E}(\mathcal{S}_2(\nabla F))^p]^{\frac{1}{2p}}$$

$$\leq \frac{C}{\gamma\alpha}[\mathbb{E}\|\tilde{V}_0\|^{2p}]^{\frac{1}{p}} + \frac{4}{\gamma^2}[\mathbb{E}(\mathcal{S}_2(\nabla F))^p]^{\frac{1}{p}} + C_p Td + C_p\frac{[\mathbb{E}(\mathcal{S}_2(V))^p]^{\frac{1}{2p}}}{\sqrt{\gamma\alpha}}$$

$$+ C_p\frac{L^2 T^{1-\frac{1}{p}}\alpha^2}{\gamma^2}[\mathbb{E}\mathcal{S}_{2p}(V)]^{\frac{1}{p}} + C_p\frac{L^2 T^{1-\frac{1}{p}}\alpha^4}{\gamma^2}[\mathbb{E}\mathcal{S}_{2p}(\nabla F)]^{\frac{1}{p}}$$

$$+ C_p\frac{TL^2\alpha^3(d + \log K)}{\gamma} + C_p\sqrt{\frac{\alpha}{\gamma}}[\mathbb{E}(\mathcal{S}_2(\nabla F))^p]^{\frac{1}{2p}} \tag{94}$$

Here, we have used that fact that $1 - \sqrt{g} \geq C\gamma\alpha$ whenever $\gamma\alpha < 1$. We now use the AM-GM inequality to show that

$$C_p \frac{[\mathbb{E}(\mathcal{S}_2(V))^p]^{\frac{1}{2p}}}{\sqrt{\gamma\alpha}} \leq \frac{[\mathbb{E}(\mathcal{S}_2(V))^p]^{\frac{1}{p}}}{10} + \frac{C_p'}{\gamma\alpha}\,.$$

Similarly, we have:

$$C_p \sqrt{\frac{\alpha}{\gamma}}\,[\mathbb{E}(\mathcal{S}_2(\nabla F))^p]^{\frac{1}{2p}} \leq \frac{[\mathbb{E}(\mathcal{S}_2(\nabla F))^p]^{\frac{1}{p}}}{2\gamma^2} + C_p'\gamma\alpha\,.$$

Applying this to the RHS of Equation (94) and re-arranging, we conclude the statement of the lemma.

2. Using similar methods as in item 1 above, we apply Lemma 27 and collect the following moment bounds:

$$\mathbb{E}|\sum_{t=0}^{T-1}\langle \nabla F(\psi_t) - \nabla F(\tilde{U}_{tK}), \tilde{\Psi}_t\rangle|^p \leq C_p \frac{L^p \alpha^{\frac{p}{2}}}{\gamma^{\frac{3p}{2}}}\sqrt{\mathbb{E}(\mathcal{S}_2(V))^p}$$

$$\mathbb{E}|\sum_{t=0}^{T-1}\langle \nabla F(\tilde{U}_{tK}), \tilde{\Psi}_t\rangle|^p \leq C_p \frac{\alpha^{\frac{p}{2}}}{\gamma^{\frac{p}{2}}}\sqrt{\mathbb{E}(\mathcal{S}_2(\nabla F))^p}$$

By a calculation similar to that in item 1, we have:

$$\mathbb{E}\left(\sum_t \frac{N_t^2 \alpha L^2}{\gamma}\left(1 + \frac{3\alpha L}{2\gamma}\right)\sup_{0 \leq i \leq K-1}\|\hat{\tilde{U}}_{tK+i} - \tilde{U}_{tK}\|^2\right)^p$$

$$\leq C_p T^{p-1}\frac{L^{2p}}{\gamma^p}\left[\alpha^{3p}\mathbb{E}\mathcal{S}_{2p}(V) + \alpha^{5p}\mathcal{S}_{2p}(\nabla F) + T\alpha^{4p}\gamma^p(d + \log K)^p\right] \quad (95)$$

Summing item 2 of Lemma 21 over $t = 0$ to $t = T-1$ and applying the triangle inequality for $p$-th moments, we have:

$$[\mathbb{E}(\mathcal{S}_2(\nabla F))^p]^{\frac{1}{p}} \leq \frac{\gamma}{\alpha}[\mathbb{E}|(F(\Psi_0) - F(\Psi_T))^+|^p]^{\frac{1}{p}} + C_p\sqrt{\frac{\gamma}{\alpha}}[\mathbb{E}(\mathcal{S}_2(\nabla F))^p]^{\frac{1}{2p}}$$

$$+ C_p L\sqrt{\frac{1}{\gamma\alpha}}[\mathbb{E}(\mathcal{S}_2(V))^p]^{\frac{1}{2p}} + \frac{CL^2}{\gamma^2}[\mathbb{E}(\mathcal{S}_2(V))^p]^{\frac{1}{p}} + C_p LT(d + \log K)$$

$$+ C_p T^{1-\frac{1}{p}}L^2\alpha^2[\mathbb{E}\mathcal{S}_{2p}(V)]^{\frac{1}{p}} + C_p T^{1-\frac{1}{p}}L^2\alpha^4[\mathbb{E}\mathcal{S}_{2p}(\nabla F)]^{\frac{1}{p}}$$

$$+ C_p T\gamma L^2\alpha^3(d + \log K)$$

$$\leq \frac{\gamma}{\alpha}[\mathbb{E}|(F(\Psi_0) - F(\Psi_T))^+|^p]^{\frac{1}{p}} + C_p\sqrt{\frac{\gamma}{\alpha}}[\mathbb{E}(\mathcal{S}_2(\nabla F))^p]^{\frac{1}{2p}}$$

$$+ C_p L\sqrt{\frac{1}{\gamma\alpha}}[\mathbb{E}(\mathcal{S}_2(V))^p]^{\frac{1}{2p}} + \frac{CL^2}{\gamma^2}[\mathbb{E}(\mathcal{S}_2(V))^p]^{\frac{1}{p}} + C_p LT(d + \log K)$$

$$+ C_p T^{1-\frac{1}{p}}L^2\alpha^2[\mathbb{E}\mathcal{S}_{2p}(V)]^{\frac{1}{p}} + C_p T^{1-\frac{1}{p}}L^2\alpha^4[\mathbb{E}\mathcal{S}_{2p}(\nabla F)]^{\frac{1}{p}} \quad (96)$$

In the last step, we have use the fact that $\gamma\alpha < 1$ and $\frac{L\alpha}{\gamma} < c_0$. Applying AM-GM inequality to $C_p\sqrt{\frac{\gamma}{\alpha}}[\mathbb{E}(\mathcal{S}_2(\nabla F))^p]^{\frac{1}{2p}}$ and $C_p L\sqrt{\frac{1}{\gamma\alpha}}[\mathbb{E}(\mathcal{S}_2(V))^p]^{\frac{1}{2p}}$ similar to item 1, we conclude the result.

□

## I.10 Proof of Lemma 23

*Proof.*

$$\tilde{X}_{sK} - \tilde{X}_{tK} = (A_{\alpha(s-t)} - \mathbf{I})\tilde{X}_{tK} + \sum_{h=t}^{s-1} A_{\alpha(s-h-1)} \Big[ G_\alpha b(\tilde{X}_{hK}) + \alpha\Delta_h + \Gamma_\alpha \tilde{Y}_h \Big]$$

In this proof only, for any random variable $W$ and any projection operator $\mathcal{P}$ over $\mathbb{R}^{2d}$ we let $\mathcal{M}_p(W; \mathcal{P}) := (\mathbb{E}[\|\mathcal{P}W\|^p | \tilde{X}_{tK}])^{\frac{1}{p}}$. We use the convention that $\mathcal{M}_p(W) = (\mathbb{E}\|W\|^p)^{\frac{1}{p}}$. Using the triangle inequality for $\mathcal{M}_p$, we conclude:

$$\mathcal{M}_p(\tilde{X}_{sK} - \tilde{X}_{tK}, \Pi) \le \mathcal{M}_p((A_{\alpha(s-t)} - \mathbf{I})\tilde{X}_{tK}, \Pi) + \sum_{h=t}^{s-1} \mathcal{M}_p(A_{\alpha(s-h-1)}G_\alpha b(\tilde{X}_{hK}), \Pi)$$

$$+ \sum_{h=t}^{s-1} \mathcal{M}_p(A_{\alpha(s-h-1)}\alpha\Delta_h, \Pi) + \mathcal{M}_p(\sum_{h=t}^{s-1} A_{\alpha(s-h-1)}\Gamma_\alpha \tilde{Y}_h, \Pi) \quad (97)$$

Similarly, we have:

$$\mathcal{M}_p(\tilde{X}_{sK} - \tilde{X}_{tK}, \mathbf{I} - \Pi) \le \mathcal{M}_p((A_{\alpha(s-t)} - \mathbf{I})\tilde{X}_{tK}, \mathbf{I} - \Pi) + \sum_{h=t}^{s-1} \mathcal{M}_p(A_{\alpha(s-h-1)}\alpha\Delta_h, \mathbf{I} - \Pi)$$

$$+ \sum_{h=t}^{s-1} \mathcal{M}_p(A_{\alpha(s-h-1)}G_\alpha b(\tilde{X}_{hK}), \mathbf{I} - \Pi)$$

$$+ \mathcal{M}_p(\sum_{h=t}^{s-1} A_{\alpha(s-h-1)}\Gamma_\alpha \tilde{Y}_h, \mathbf{I} - \Pi) \quad (98)$$

By scaling relations given in Section 2.1, we have: $\sum_{h=t}^{s-1} \Pi A_{\alpha(s-h-1)}\Gamma_\alpha \tilde{Y}_h \sim \mathcal{N}(0, \Pi(\Gamma^2_{(s-t)\alpha})\Pi)$ and $\sum_{h=t}^{s-1} (\mathbf{I} - \Pi) A_{\alpha(s-h-1)}\Gamma_\alpha \tilde{Y}_h \sim \mathcal{N}(0, (\mathbf{I} - \Pi)(\Gamma^2_{(s-t)\alpha})(\mathbf{I} - \Pi))$. Notice that $\Pi(\Gamma^2_{\alpha(s-t)})\Pi \lesssim \begin{bmatrix} \frac{4\alpha(s-t)}{\gamma} & 0 \\ 0 & 0 \end{bmatrix}$ and $(\mathbf{I} - \Pi)(\Gamma^2_{\alpha(s-t)})(\mathbf{I} - \Pi) \lesssim \begin{bmatrix} 0 & 0 \\ 0 & 2\gamma\alpha(s-t) \end{bmatrix}$

Therefore, we conclude:

$$\mathcal{M}_p(\sum_{h=t}^{s-1} A_{\alpha(s-h-1)}\Gamma_\alpha \tilde{Y}_h, \Pi) \le C_p \sqrt{\frac{d\alpha(s-t)}{\gamma}}$$

$$\mathcal{M}_p(\sum_{h=t}^{s-1} A_{\alpha(s-h-1)}\Gamma_\alpha \tilde{Y}_h, \mathbf{I} - \Pi) \le C_p \sqrt{d\alpha\gamma(s-t)}$$

Notice that $\|\Pi(A_{\alpha(s-t)} - \mathbf{I})\tilde{X}_{tK}\| \le \alpha(s-t)\|\tilde{V}_{tK}\|$ and $\|(\mathbf{I} - \Pi)(A_{\alpha(s-t)} - \mathbf{I})\tilde{X}_{tK}\| \le \alpha\gamma(s-t)\|\tilde{V}_{tK}\|$. This implies:

$$\mathcal{M}_p((A_{\alpha(s-t)} - \mathbf{I})\tilde{X}_{tK}, \Pi) \le \alpha(s-t)\mathcal{M}_p(\tilde{V}_{tK})$$

$$\mathcal{M}_p((A_{\alpha(s-t)} - \mathbf{I})\tilde{X}_{tK}, \mathbf{I} - \Pi) \le \alpha\gamma(s-t)\mathcal{M}_p(\tilde{V}_{tK})$$

Notice that: $\|\Pi A_{\alpha(s-h-1)}G_\alpha b(\tilde{X}_{hK})\| \leq \frac{\alpha}{\gamma}\|\nabla F(\tilde{U}_{hK})\|$ and $\|(\mathbf{I}-\Pi)A_{\alpha(s-h-1)}G_\alpha b(\tilde{X}_{hK})\| \leq \alpha\|\nabla F(\tilde{U}_{hK})\|$. This implies:

$$\mathcal{M}_p(A_{\alpha(s-h-1)}G_\alpha b(\tilde{X}_{hK}),\Pi) \leq \frac{\alpha}{\gamma}\mathcal{M}_p(\nabla F(\tilde{U}_{hK})) \leq \frac{\alpha L}{\gamma}\mathcal{M}_p(\tilde{X}_{hK}-\tilde{X}_{tK},\Pi) + \frac{\alpha}{\gamma}\mathcal{M}_p(\nabla F(\tilde{U}_{tK}))$$

$$\mathcal{M}_p(A_{\alpha(s-h-1)}G_\alpha b(\tilde{X}_{hK}),\mathbf{I}-\Pi) \leq \alpha\mathcal{M}_p(\nabla F(\tilde{U}_{hK})) \leq \alpha L\mathcal{M}_p(\tilde{X}_{hK}-\tilde{X}_{tK},\Pi) + \alpha\mathcal{M}_p(\nabla F(\tilde{U}_{tK}))$$

Notice that: $\|\Pi A_{\alpha(s-h-1)}\alpha\Delta_h\| \leq \frac{\alpha L N_h}{\gamma}\sup_{0\leq i\leq K-1}\|\hat{\tilde{U}}_{hK+i} - \tilde{U}_{hK}\|$ and $\|(\mathbf{I}-\Pi)A_{\alpha(s-h-1)}\alpha\Delta_h\| \leq \alpha L N_h\sup_{0\leq i\leq K-1}\|\hat{\tilde{U}}_{hK+i} - \tilde{U}_{hK}\|$. Applying item 1 in Lemma 21, we conclude:

$$\|\Pi A_{\alpha(s-h-1)}\alpha\Delta_h\| \leq \frac{C\alpha^2 L N_h}{\gamma}\|\tilde{V}_{hK}\| + \frac{C\alpha^3 L N_h}{\gamma}\|\nabla F(\tilde{U}_{hK})\| + \frac{C\alpha L N_h}{\gamma}M_{hK} \qquad (99)$$

Therefore, we conclude:

$$\mathcal{M}_p(A_{\alpha(s-h-1)}\alpha\Delta_h,\Pi) \leq \frac{C_p\alpha^2 L}{\gamma}\mathcal{M}_p(\tilde{V}_{hK}) + \frac{C_p\alpha^3 L}{\gamma}\mathcal{M}_p(\nabla F(\tilde{U}_{hK})) + \frac{C_p\alpha L}{\gamma}\mathcal{M}_p(M_{hK})$$

$$\leq \frac{C_p\alpha^2 L}{\gamma}\mathcal{M}_p(\tilde{V}_{hK}) + \frac{C_p\alpha^3 L}{\gamma}\mathcal{M}_p(\nabla F(\tilde{U}_{tK})) + \frac{C_p\alpha^3 L^2}{\gamma}\mathcal{M}_p(\tilde{X}_{tK} - \tilde{X}_{hk},\Pi)$$

$$+ \frac{C_p\alpha L}{\gamma}\mathcal{M}_p(M_{hK})$$

$$\leq \frac{C_p\alpha^2 L}{\gamma}\mathcal{M}_p(\tilde{V}_{hK}) + \frac{C_p\alpha^3 L}{\gamma}\mathcal{M}_p(\nabla F(\tilde{U}_{tK})) + \frac{C_p\alpha^3 L^2}{\gamma}\mathcal{M}_p(\tilde{X}_{tK} - \tilde{X}_{hk},\Pi)$$

$$+ \frac{C_p\alpha^{\frac{5}{2}}L\sqrt{(d+\log K)}}{\sqrt{\gamma}}$$

$$\leq \frac{C_p\alpha^2 L}{\gamma}\mathcal{M}_p(\tilde{X}_{tK},\mathbf{I}-\Pi) + \frac{C_p\alpha^2 L}{\gamma}\mathcal{M}_p(\tilde{X}_{tK} - \tilde{X}_{hK},\mathbf{I}-\Pi) + \frac{C_p\alpha^3 L}{\gamma}\mathcal{M}_p(\nabla F(\tilde{U}_{tK}))$$

$$+ \frac{C_p\alpha^3 L^2}{\gamma}\mathcal{M}_p(\tilde{X}_{tK} - \tilde{X}_{hK},\Pi) + \frac{C_p\alpha^{\frac{5}{2}}L\sqrt{(d+\log K)}}{\sqrt{\gamma}} \qquad (100)$$

In the first step, we have used the triangle inequality for $\mathcal{M}_p$ along with the fact that $N_h$ is independent of $M_{h,K}$ and $\tilde{X}_{hK}$ and the fact that its $p$-th moment is bounded by a constant $C_p$. In the second step, we have controlled the norm of $\nabla F(\tilde{U}_{hK})$ with that of $\nabla F(\tilde{U}_{tK})$ using Lipschitzness. In the third step, we have invoked item 3 of Lemma 21 to bound the moments of $M_{h,K}$. In the fourth step, we have controlled the norm $\tilde{V}_{hK}$ in terms of $\tilde{X}_{tK}$ and $\tilde{X}_{tK} - \tilde{X}_{hK}$.

Similarly, we have:

$$\mathcal{M}_p(A_{\alpha(s-h-1)}\alpha\Delta_h,\mathbf{I}-\Pi) \leq C_p\alpha^2 L\mathcal{M}_p(\tilde{X}_{tK},\mathbf{I}-\Pi) + C_p\alpha^2 L\mathcal{M}_p(\tilde{X}_{tK} - \tilde{X}_{hK},\mathbf{I}-\Pi)$$

$$+ C_p\alpha^3 L\mathcal{M}_p(\nabla F(\tilde{U}_{tK})) + C_p\alpha^3 L^2\mathcal{M}_p(\tilde{X}_{tK} - \tilde{X}_{hK},\Pi)$$

$$+ C_p\alpha^{\frac{5}{2}}\sqrt{\gamma}L\sqrt{(d+\log K)} \qquad (101)$$

Applying the estimates derived above to Equation (97), along with the assumption that $\frac{\alpha L(s-t)}{\gamma} \leq c_p$, $\alpha\gamma(s-t) \leq c_p$ and $\alpha^2 L(s-t) \leq c_p$ for some small enough constant $c_p$ which depends only on $p$, we conclude:

$$\sup_{t\leq h\leq s}\mathcal{M}_p(\tilde{X}_{hK} - \tilde{X}_{tK},\Pi) \leq 2\alpha(s-t)\mathcal{M}_p(\tilde{X}_{tK},\mathbf{I}-\Pi) + \alpha\sup_{t\leq h\leq s}\mathcal{M}_p(\tilde{X}_{tK} - \tilde{X}_{hK},\mathbf{I}-\Pi)$$

$$+ \frac{2\alpha(s-t)}{\gamma}\mathcal{M}_p(\nabla F(\tilde{U}_{tK})) + C_p\sqrt{\frac{(d+\log K)}{L}} \qquad (102)$$

Similarly considering Equation (98), we have:

$$\sup_{t \le h \le s} \mathcal{M}_p(\tilde{X}_{hK} - \tilde{X}_{tK}, \mathbf{I} - \Pi) \le 2\alpha\gamma(s-t)\mathcal{M}_p(\tilde{X}_{tK}, \mathbf{I} - \Pi) + 2\alpha(s-t)\mathcal{M}_p(\nabla F(\tilde{X}_{tK}))$$

$$+ 2\alpha L(s-t)\sup_{t \le h \le s}\mathcal{M}_p(\tilde{X}_{hK} - \tilde{X}_{tK}, \Pi) + C_p\gamma\sqrt{\frac{d + \log K}{L}} \tag{103}$$

From Equations (102) and (103), we conclude:

$$\sup_{t \le h \le s}\mathcal{M}_p(\tilde{X}_{hK} - \tilde{X}_{tK}, \Pi) \le 8\alpha(s-t)\mathcal{M}_p(\tilde{X}_{tK}, \mathbf{I} - \Pi) + \frac{8\alpha(s-t)}{\gamma}\mathcal{M}_p(\nabla F(\tilde{U}_{tK}))$$

$$+ C_p\sqrt{\frac{d + \log K}{L}} \tag{104}$$

$$\sup_{t \le h \le s}\mathcal{M}_p(\tilde{X}_{hK} - \tilde{X}_{tK}, \mathbf{I} - \Pi) \le 8\gamma\alpha(s-t)\mathcal{M}_p(\tilde{X}_{tK}, \mathbf{I} - \Pi) + 8\alpha(s-t)\mathcal{M}_p(\nabla F(\tilde{U}_{tK}))$$

$$+ C_p\gamma\sqrt{\frac{d + \log K}{L}} \tag{105}$$

$\square$

