# OpenReview forum: "The Poisson Midpoint Method for Langevin Dynamics:  Provably Efficient Discretization for Diffusion Models"
_NeurIPS.cc/2024/Conference — NeurIPS 2024 poster_

### Official Review · Reviewer_p2Nt · 2024-07-12

**Soundness:** 3
**Presentation:** 3
**Contribution:** 3
**Rating:** 7
**Confidence:** 3

**Summary:**

The paper proposes a novel, multi-step discretization method for SDEs, with particular applications to the Langevin diffusions (overdamped and underdamped) and the denoising diffusion model SDE. The rates obtained are better than any other known guarantees in KL divergence.

**Strengths:**

The discretization proposed appears to be completely novel, and the analytic framework (using CLT type arguments) has not appeared before in this context.

The rates are state of the art in KL, clearly surpassing the previous best known discretizations in KL without any caveats.

The experimental evidence is thorough and appears to suggest that this discretization is indeed better than a naive one, for the diffusion model setting.

**Weaknesses:**

The algorithm as described in the paper is somewhat cumbersome to read. Perhaps some simpler exposition or visualization may be helpful.

The rates obtained are worse than those of the randomized midpoint discretizations (noting that these latter rates are only valid in Wasserstein distance).

**Questions:**

Although the notation in Section 2 is necessary to capture all the cases considered in this work, it is a bit cumbersome and makes the results difficult to parse. Perhaps for the proof sketch, only the OLMC case could be presented.

The final rates have some somewhat strange dependence on the key parameters; for instance, the dependence on dimension is $d^{5/12}$ for the underdamped PLMC. Do the authors have any intuition whether the rates obtained in their analysis are close to optimal for this discretization, and whether it would be possible to improve it with slight algorithmic modifications?

Is the method extendible to the case where the tail growth of the potential is of order $|x|^\alpha$, for $\alpha < 2$? Or the case of weak smoothness?

Minor:

59: stochastic approximations LMC -> "stochastic approximations of LMC"

70: denotes identity -> denotes the identity

108 "Motive" -> "motivate"

206: Theorem 1; I believe this is meant to refer to Theorem 3.

Many equations/inline formulas are missing punctuation. For instance Eq. 6, Steps 3-5 (ll 145-149).

**Limitations:**

The authors have adequately addressed the limitations of their work.

---

> ### Author Rebuttal · Authors · 2024-07-31
>
> We thank the reviewer for the great feedback and multiple interesting possibilities for extension. We will improve the exposition and readability of the paper in the revision based on your comments. We address the questions raised by the reviewer below.
>
>
> **The main weakness** pointed out by the reviewer is that our rates are worse than prior works on randomized midpoint discretizations. We respectfully disagree with this claim. Our rates are better than the prior works. To elaborate: The literature on sampling compares $\mathcal{W}_2 \leq  \epsilon$ to $\mathsf{TV} \leq \epsilon$ to $\mathsf{KL} \leq \epsilon^2$ (due to the information transportation inequality and the Pinsker's inequality). The number of steps required by our algorithm to achieve $\mathsf{TV} \leq \epsilon$ is $O(\frac{1}{\sqrt{\epsilon}})$ for ULMC (see table 1 in our manuscript) whereas prior works (see [1, Theorem 3]) require $O(1/\epsilon^{2/3})$ to achieve $\mathcal{W}_2 \leq  \epsilon$. Please let us know in case we misunderstood your comment.
>
> [1] Shen, Ruoqi, and Yin Tat Lee. "The randomized midpoint method for log-concave sampling." Advances in Neural Information Processing Systems 32 (2019).
>
> ### Questions:
>
> 1. **Strange problem parameter dependencies and explanation:** We use a Wasserstein CLT based analysis (based on results from Zhai et al). Based on results of Theorem 1, this requires controlling the moments $\mathbb{E}\|\nabla F(\tilde{X}_t)\|^{2k}$. Since we focus on convergence under very general conditions, we control these moments using smoothness of $F$ only as described in Section 4. This leads to a multitude of complex terms which finally leads to the said bound. It is not intuitively clear to us why factors such as $d^{5/12}$ make sense. However, we note that the dependence on $\epsilon$ has a clear explanation given in Section 2.
>
> We believe the results can be further improved with stronger assumptions such as strong log-concavity. Here numerous nice properties such as step by step contractive coupling can allow us to control the higher order terms precisely. We have not been able to design a better algorithm in this setting to overcome the said dependencies. We leave these further investigations for future work.
>
> 2. **Extensions to other tail growth conditions and weakly smooth potentials:**
>
> Based on the comments by the reviewer, we surveyed the literature to check if this is possible. We believe these results are extendible to settings where the Poincare inequality holds. This corresponds to a tail decay of the form $e^{-\|x\|}$ (since the Poincare inequality implies sub-exponential concentration).
>
> Specifically, we can combine the guarantees for LMC when the target satisfies PI (such as [2]) and obtain guarantees for our algorithm PLMC based on Theorem 2. Extension to other types of tails should be possible if results similar those in [2] are available in such settings for LMC. However, we do not know if sub-exponential (or stretched exponential type tails) are sufficient for fast convergence of LMC and additional isoperimetry type assumptions might be required.
>
> We checked that we can extend the results of Theorems 2 and 3 to consider weakly smooth potentials, with some modifications to the proof. These modifications, along with off the shelf results for LMC given in works such as [3] can allow us to consider weakly smooth potentials as well.
>
>
> [2] Balasubramanian, Krishna, et al. "Towards a theory of non-log-concave sampling: first-order stationarity guarantees for langevin monte carlo." Conference on Learning Theory. PMLR, 2022.
>
> [3] Chewi, Sinho, et al. "Analysis of langevin monte carlo from poincare to log-sobolev." Foundations of Computational Mathematics (2024): 1-51.
>
> ### Conclusion
> We believe that we have addressed all the technical questions posed by the reviewer. We will fix the exposition in our next revision. Our work is novel theoretically, and we demonstrate the superior performance of our method with large scale image generation tasks. We hope the reviewer can reconsider their score.

---

> > ### Comment · Reviewer_p2Nt · 2024-08-13
> >
> > I thank the authors for their response. In particular, given that they claim to be able to extend this analysis even to the weakly smooth and Poincare cases, I believe this framework has enough power and novelty to be accepted. Hence, I have raised my score.
> >
> > I was mistaken regarding the comparison with randomized midpoint; the dimension dependence is worse (which is still mysterious to me) but the inverse accuracy dependence is indeed better as noted by the authors. I thank the authors for their correction.

---

> > > ### Author Response · Authors · 2024-08-14
> > > **Thank you**
> > >
> > > Thank you very much!

---

### Official Review · Reviewer_HMqT · 2024-07-12

**Soundness:** 3
**Presentation:** 1
**Contribution:** 3
**Rating:** 5
**Confidence:** 3

**Summary:**

In this work, the authors draw inspiration from the randomized midpoint method for Langevin dynamics and design a new method called the poisson midpoint method. They establish guarantees for this method relative to the EM discretisation with thinning, and discuss theoretical and practical applications to sampling.

**Strengths:**

### Originality and Significance

The idea is quite interesting, and binning the unit interval to perform a randomized midpoint discretization is novel in my view. This clearly is not a straightforward innovation despite seeming like one. The theoretical statements also help characterize the benefits of this new strategy, which appear to be shown under weaker conditions than those required for sampling. Practically, this is also very appealing as the computational costs for the applications to generative modelling is lower than now-standard techniques when fixing the requirement (i.e., low FID).

### Quality and Clarity

The idea is well-motivated, but I feel like the paper could be a lot more clearly presented, especially to be accessible to broader audiences. Please see the weaknesses / questions section below.

**Weaknesses:**

The writing of the paper can be definitely improved. It appears a bit rushed in multiple key areas, and several definitions and procedures are stated without much consideration towards the readability of those portions. Since there is a new method being proposed, it would help if intuition for this method is proposed, not just for the idea, but also the design of the method itself. Additionally, when certain complex steps (relative to other methods described in the paper like LMC) are mentioned (see the Algorithm and Computational complexity paragraph on Page 5), it would help to explain each (or a subset of them) in layman's terms to connect them to method (the two equations (7) and (8) on the previous page). The theoretical result in Section 3, specifically theorem 1 has inconsistent terminology ($B$ is the Brownian motion, $\mathsf{S}(...)$ and $\mathsf{PS}$ are lacking a $b$, $B$, $b$, $\beta$, the role of $r$ and the randomness of the expectation is unclear. In the next theorems, the authors state "Lower Order Terms" instead of a small-O notation would have seemed more appropriate. The paragraph on quadratic speedup is also quite hastily presented in my opinion, because it is still not clear to me how the speedup is represented.

**Questions:**

1. What is $p_{\tau}$ in Eq. (4)?
2. As mentioned in Page 2, $A_\alpha$, $G_\alpha$, $\Gamma_\alpha$ are time independent, and so DDPM doesn't fall under the model in (1) right? How is DDPM relevant to the scheme (1) in that case?
3. The scaling relations part of Section 2.1 could perhaps be placed later on? It's difficult to understand its purpose since it seems disconnected from the rest of the main draft.
4. In Line 109, "speed up" is usually a computational term. Is this alluding to the "speed up" discussed later in Section 3 (in which case it might help to reference it here)?
5. Since RLMC is also relatively complex (in comparison to a standard forward EM discretization), it would help to remark on how this is implemented, even if briefly. As written, it is essentially a combination of two discretizations: one forward, and one backward, and the backward discretization is the harder one to implement.
6. The section of text between Lines 118--123 is very hard to follow. Since it is essential to the intuition, I would recommend that it be reworded suitably to be more formal and less colloquial. What is $\mathrm{law}(Z + \epsilon)$ and $\mathrm{law}(Z + \epsilon Z')$?
7. I feel like the claim of "replacing K steps with one step" is not quite true, since eqs (7), (8) are each performing K steps each right?
8. In eq (7), could the authors perhaps give some intuition about the summation on the RHS, and what the cheap interpolation is actually performing? A diagram would help greatly, but it is understandable if this is difficult to provide.
9. In (8), what is the purpose of H_{t,i}? Why don't the $\Gamma$, $A$, $G$ terms not have a "i" scaling like in the cheap interpolation?
10. Why does $I_t$ have $N$ elements, and what is $N$ here? Initial conditions / conventions for the recursion in step 2 would be better off being stated apriori. What is $\mathrm{Corr}$ in Step 4?
11. Where is the computational complexity discussed in the paragraph on the top of page 5?
12. How is there no dependence on the $\{X_{t}\}$ sequence in Eq. 10?
13. Why is this strange dependence on 14th moment in Assumption 2 present?

**Limitations:**

Yes.

---

> ### Author Rebuttal · Authors · 2024-08-01
>
> We thank the reviewer for the detailed comments about the paper. We will improve the typos and the inconsistent notations in the revised manuscript. Since the revised version allows for an additional page, we hope to have enough space for a detailed discussion of every aspect pointed out by the reviewer. We will also replace the “Lower Order Terms” with the O() notation.
>
>
> ### General Discussion of the Algorithm:
> Before we delve into responses to specific questions, we add some discussion which can help answer questions 7,8,9 below. Consider
>
> A1: The dynamics S with step size $\frac{\alpha}{K}$ performed for $K$ steps.
>
> A2: The dynamics S with step size $\alpha$ performed for $1$ step.
>
> O1: Our method PS with step size $\alpha$, which approximates A1 with one step.
>
> A1 outperforms A2 since it has a finer time discretization and evaluates the drift function $b()$ a total of $K$ times. The drift evaluation is the most expensive part of the computation and is essentially the bottleneck for the computations. In case of DDPMs, it involves neural networks with $O(10^9)$ parameters. Like most of the sampling literature, we consider computational complexity in terms of number of evaluations of the drift function. We will clarify this in the revised version.
>
> O1 is obtained by adding a randomized correction term to A2, so that in expectation (with respect to the randomization in the correction) K steps of A1 approximately matches one step of A2. However, such a randomized correction cannot involve $K$ evaluations of the drift $b()$ since that is expensive. Thus, we evaluate the drift at a random set of midpoints given by $I_t$ and rescale it. The construction of the random midpoint set $I_t$ is via the random variables $H_{t,i}$ which are i.i.d. $\mathsf{Ber}(1/K)$. That is, the i-th point for A1 is included in $I_t$ if and only if $H_{t,i} =1$. Note that $\mathbb{E}|I_t| = 1$. Thus, we only need to do one additional drift function evaluation in expectation.
>
> ### Responses to Specific Questions:
>
>
> 1. Here $p_\tau$ is the probability density function $X_{\tau}$. This is explained in the linked references to the DDPM, but we will add a short discussion in the revised version. Thank you for pointing this out.
> 2. We refer to lines 103-105 in the manuscript where we have discussed this. We chose $A_{\alpha},B_{\alpha},\Gamma_{\alpha}$ to be time independent for the sake of clarity, but give a similar algorithm applicable to DDPMs in Appendix A.1.
> 3. We will relocate the scaling relations to a later section in the revised version.
> 4. This indeed refers to the results in Section 3. We will clarify the computational aspects of this in the revised version. Also refer to the general discussion above.
> 5. We will add a discussion of RLMC implementation in the appendix in the revised version.
> 6. $\mathsf{law}(X)$ for a random variable $X$ refers to its probability measure. This notation is defined in Section 1.3. We will make the said part clearer. The technical statement we wanted to make was that a Gaussian with mean 0 and variance $1+\epsilon^2$ is closer to a standard Gaussian than a Gaussian with mean $\epsilon$ and variance $1$.
> 7. Eventhough (7) and (8) are mathematically $K$ steps (the equations being useful in the theoretical analysis), we refer to lines 141 to 153 where we algorithmically implement this much more efficiently. More precisely, we generate the random midpoint set $I_t$, and update the iterations only at the midpoints. Moreover, $I_t$ is such that $\mathbb{E}|I_t|  = 1$. Since lines 141-153 give our main algorithm $\mathsf{PS}$, we believe that this does approximate $K$ steps with just one step.
> 8.  Please refer to the general discussion above.
> 9.  Please refer to the general discussion above.
> 10. Please refer to the general discussion above. The term $\mathsf{Corr}$ is defined in step 4. The notation $:=$ means that the LHS is defined as RHS.
> 11. This has been discussed in Proposition 1 in page 5. This shows that one step of $\mathsf{PS}()$ requires two evaluations of the drift $b()$. As discussed above, the notion of computational complexity in the sampling literature is the number of evaluations of the drift function. We will make this clear in the revised version.
> 12. We compare the distributions of $X$ and $\tilde{X}$ via the KL divergence with the distribution of $X$ being the base measure. The expectation with respect to $\tilde{X}$ is a consequence of the chain rule for KL divergence (Lemma 2 in the appendix):  Let $p$ and $q$ be the joint distributions over $\mathbb{X}\times \mathbb{Y}$ and $p_x,q_x$ be the marginals over $\mathbb{X}$ and $p_{y|x}, q_{y|x}$ be the respective conditional distributions over $\mathbb{Y}$ given $x \in \mathbb{X}$. The chain rule states that:
>
> $$\mathsf{KL}(p||q)  = \mathsf{KL}(p_x||q_x) + \mathbb{E}_{x \sim p_x} \mathsf{KL} (p(y|x)||q(y|x))$$
>
>
> The second term is the expected KL divergence between conditionals where the conditioning variable is derived from $p_x$ rather than from $q_x$. In our case $p$ corresponds to $\tilde{X}$ and $q$ corresponds to $X$.
>
>
>
> 13. The 14th moment assumption is very mild since prior works consider Gaussian initializations which have all moments. We have explained this point in detail in the response to Reviewer F8sy. To understand why this shows up, we look at Equation (10) in Theorem 1. Here we take r = 7 and that requires us to bound $\mathbb{E} \beta_{tK+i}^{2r}$. By Lipschitzness assumption on $b()$, this reduces to bounding the 14th moment of $\tilde{X}_{tK+i}$, which in turn can be controlled via the initialization.
>
>
> ### Conclusion
> The reviewer agrees that the paper is very well motivated, and has strong theoretical and empirical results. We have answered the reviewer's technical concerns above and we will improve the presentation in the next version of our manuscript. We hope that the reviewer can reconsider their score in this light. We are happy to answer any further technical questions.

---

> > ### Comment · Reviewer_HMqT · 2024-08-10
> > **Thank you for your rebuttal**
> >
> > I have no further questions, and I feel that the contributions of the paper is very interesting, albeit poorly presented. Improvements to the presentations as suggested by multiple reviewers would certainly make this a good paper. I maintain by score on the basis of this.

---

### Official Review · Reviewer_F8sy · 2024-07-26

**Soundness:** 3
**Presentation:** 2
**Contribution:** 2
**Rating:** 5
**Confidence:** 4

**Summary:**

Based on the Randomized Midpoint Method (RMM) proposed for simulating the Langevin Dynamics numerically, the authors proposed a variant called the Poisson Midpoint Method (PMM). Theoretical analysis is performed to bound the deviation between the continuous dynamics and the discretized dynamics. Moreover, when PMM is applied to simulate the Langevin Dynamics, the authors have shown that PMM can accelerate both the Overdamped Langevin Monte Carlo (OLMC) and the Underdamped Langevin Monte Carlo (ULMC) under additional assumptions like isoperimetry. Furthermore, numerical experiments on image generation with DDPM are conducted to illustrate the effectiveness of the proposed method.

**Strengths:**

This paper is technically solid with a complete list of related work. The contributions are stated in a clear way. Also, a complete set of numerical experiments on diffusion models is included to illustrate the effectiveness of the proposed method.

**Weaknesses:**

The presentation of this paper can be potentially improved from a few aspects. Firstly, it seems a bit weird to use the phrase "lower order terms" (see for instance, line 167-168, 172-173, 205-206) in mathematical equations and derivations. One possible alternative is to replace it via the big O notation. Secondly, the authors might consider clarifying the differences between Langevin Dynamics and the backward SDE in diffusion model. One important thing that probably needs to stated clearly is that PMM is a method for simulating general time-dependent SDE (like the reversed SDE in diffusion models), but this paper only provides theoretical analysis for the case when PMM is applied to simulate Langevin Dynamics. Furthermore, it seems that the authors didn't discuss potential topics for future work.

**Questions:**

One major concern of the reviewer is that one assumption in the paper seems to be too strong. Specifically, Assumption 2 in the paper bounds the 14-th moment of the distance between the starting point and the global minima. To the best of the reviewer's knowledge, such assumption seems to be uncommon in sampling literature. Would it be possible for the authors to elaborate a bit more on why this assumption is essential here? It seems that this assumption essentially restricts the starting point to be a warm start, which makes the sampling task easier.

**Limitations:**

A few typos (the list is probably incomplete) in the current version of the manuscript are also listed below:

1. From line 179-180, the two phrases "to OLMC" and "to ULMC" appeared twice. The two sentences probably need to revised.
2. In line 206, the subtitle should be "Sketch for Theorem 3". Moreover, the following sentence should be "This is similar to Theorem 2".

---

> ### Author Rebuttal · Authors · 2024-07-31
>
> We thank the reviewer for the great review, which has helped us improve our work. Our responses to the questions and concerns raised are given below.
>
> ### The Main Concern:
> This is regarding the 14th moment assumption (i.e, Assumption 2) for the initialization. Below we discuss the validity of our assumption and demonstrate that such assumptions are common in the sampling literature (and implied by the assumptions made).
>
> Suppose $x^*$ is the global minimum of the potential function $F$ and note that $x^*$ is the mode of the target distribution. The assumption is that the initial iterate satisfies $\mathbb{E}\|x_0 - x^*\|^{14} = O(d^7)$. It is reasonable to assume $\|x^*\| = O(\sqrt{d})$ (for example this happens when each coordinate of $x^* $ is $O(1)$). In this case, a standard Gaussian initialization satisfies Assumption 2 since all its moments exist and the $2k$-th moment is of the order $d^{k}$.  Assumption 2 is used to bound $\mathbb{E} [(F(x_0)- F(x^*))^7]$, and this can be replaced with any of the alternative assumptions as in the literature (see Examples 1 and 2 below).
>
> It is common in the literature to consider well crafted Gaussian initializations with the mean being a local minimum/ global minimum of the potential function, along with an appropriate covariance, which then satisfy the $14$-th moment assumption. We discuss 4 such examples from recent works. We can provide more references with similar assumptions/elaborate the details mathematically if required.
>
>
> **Example 1:** For ULMC, see Appendix D and Lemma 27 of [1, COLT 2023 version]. This work considers $0$ to be a local minima of $F$ and the function value at $F(0)-F(x^*) = O(d)$. The initialization is a Gaussian centered at $0$ with an appropriate variance.  We note that [1] achieves the best known analysis for ULMC in the literature. This assumption can in fact replace our Assumption 2. (We are happy to elaborate more on this connection if required). Similar considerations are also used in [5, Appendix A].
>
> **Example 2:** For OLMC, see discussion below Theorem 2  in [2, NeurIPS 2019 version], where the initialization is a Gaussian centered at a local minimum of $F$.
>
> **Example 3:** The original paper on the Randomized Midpoint Method considers starting the algorithm from exactly the global minimum of F (see [3, Theorem 3]).
>
> **Example 4:** Consider the case of image generation with pixel values in the range $[0,1]$ as considered in [4], which uses Langevin dynamics to generate these samples. Here it is reasonable to assume that it is reasonable to assume that $\|x^{*}\| = O(\sqrt{d})$. The initialization is a uniform distribution which satisfies our assumption.
>
> **Regarding Warm-Start:**
>
> We also respectfully disagree that this is a warm-start. In the sampling literature see [6,7,8], warm start means an almost sure upper bound on the radon nikodym derivative of the initial distribution with respect to the target. Our condition is much weaker than a warm start. A simple counter example would be if the initial distribution is a Gaussian $\mathcal{N}(0,2\mathbf{I}_d)$ and the target distribution $\mathcal{N}(0,\mathbf{I}_d)$. This satisfies our initialization condition whereas this does not satisfy the warm-start condition.
>
> [1] Improved discretization analysis for underdamped Langevin Monte Carlo, Zhang et al, COLT 2023
>
> [2] Rapid convergence of the unadjusted langevin algorithm: Isoperimetry suffices, Vempala and Wibisono, NeurIPS 2019
>
> [3] The randomized midpoint method for log-concave sampling. Shen and Lee, NeurIPS 2019.
>
> [4] Generative modeling by estimating gradients of the data distribution., Song and Ermon , NeurIPS 2019
>
> [5] Analysis of Langevin Monte Carlo from Poincare to Log-Sobolev,  Chewi, et al. Foundations of Computational Mathematics (2024)
>
> [6] Geometric random walks: a survey.  Santosh Vempala. Combinatorial and computational geometry  (2005)
>
> [7] Nonconvex sampling with the Metropolis-adjusted Langevin algorithm. Oren and Vishnoi COLT, 2019.
>
> [8] Random walks in a convex body and an improved volume algorithm. Lovász and Simonovits. Random structures & algorithms (1993)
> ### Discussion of  Future work and  Langevin Dynamics vs Backwards SDE in DDPM:
>
> We have discussed future work very briefly in lines 255-256, where we call for investigation of numerical approximations which work below 50 steps of DDPM. An interesting theoretical direction could be to derive bounds for time dependent SDE based sampling algorithms such as DDPM and with conditions such as the Poincare inequality or weak smoothness of the $F$ instead of smoothness (that is $\nabla F$ is Holder continuous instead of Lipschitz). We will elaborate on this discussion and also discuss the differences between Langevin Dynamics and Backwards SDE in DDPM in the revision, along with the required references. Since we will have one more page for the revised version, we can add a more detailed discussion as pointed out by the reviewer in order to make the presentation of the material clearer.
>
> ### Regarding “Lower Order Terms”:
>
> Thanks for noting the confusion caused by using “Lower Order Terms” instead of the O() notation. We mainly wanted to convey the existence of other explicit terms (given in the appendix), with varied dependencies on the problem parameters. We can switch to the O() notation in the revision.
>
> ### Conclusion:
> We will address all the minor typos and other points raised by the reviewer in the revised version. We believe we have addressed the main concern of the reviewer (the 14-th moment assumption) and hope the reviewer can revise their rating.

---

> > ### Comment · Reviewer_F8sy · 2024-08-09
> > **Response to the authors' rebuttal**
> >
> > Dear authors,
> >
> > Thank you so much for your detailed response, which have addressed most of my concerns. By reading other reviewers' comments, I agree with them that the presentation of the paper can be improved from many aspects. It would help improve the quality of the manuscript if the authors could follow the suggestions to revise the paper accordingly.
> >
> > Best regards,
> >
> > Reviewer F8sy

---

### Author Rebuttal · Authors · 2024-08-07

We thank the reviewers for valuable feedback and have responded to the key technical concerns in our individual rebuttals. We are committed to enhancing the clarity and comprehensiveness of the paper by incorporating additional details, as suggested.

The reviewers unanimously recognize the novelty of the proposed algorithm, the rigor of its theoretical analysis, and the effectiveness demonstrated through empirical evaluation with diffusion models. Based on our past experience with NeurIPS, a paper exhibiting these strengths is generally regarded as a significant contribution to the field. We respectfully request the reviewers reconsider their scores in light of this.

---

### Decision · Program_Chairs · 2024-09-25

**Decision:**

Accept (poster)

**Comment:**

This paper considers Poisson Midpoint Method, which is an alternative to Randomized Midpoint Method. Authors prove that the proposed scheme can achieve quadratic speed up of LMC. Authors demonstrate the performance via numerical studies.

This paper was reviewed by three reviewers with the following Scores/Confidence: 5/3, 7/3, 5/4. I think the paper is studying an interesting topic and the results are relevant to NeurIPS community. The following concerns were brought up by the reviewers:

- Theorem statements are quite convoluted and should be improved.

- Proof sketch can be more user friendly for the broader audience.

- Poor presentation should be improved in a few places. See reviewers' comments for details.

- The condition on initialization is certainly not restrictive but should be clarified.

Authors should carefully go over reviewers' suggestions and address any remaining concerns in their final revision. Based on the reviewers' suggestion, as well as my own assessment of the paper, I recommend including this paper to the NeurIPS 2024 program.